# On the Theory of Reinforcement Learning with Once-per-Episode Feedback

**Niladri S. Chatterji**[*]
Stanford University
niladri@cs.stanford.edu

**Aldo Pacchiano**[*]
Microsoft Research
apacchiano@microsoft.com

**Peter L. Bartlett**
UC Berkeley
peter@berkeley.edu

**Michael I. Jordan**
UC Berkeley
jordan@cs.berkeley.edu

## Abstract

We study a theory of reinforcement learning (RL) in which the learner receives binary feedback only once at the end of an episode. While this is an extreme test case for theory, it is also arguably more representative of real-world applications than the traditional requirement in RL practice that the learner receive feedback at every time step. Indeed, in many real-world applications of reinforcement learning, such as self-driving cars and robotics, it is easier to evaluate whether a learner's complete trajectory was either "good" or "bad," but harder to provide a reward signal at each step. To show that learning is possible in this more challenging setting, we study the case where trajectory labels are generated by an unknown parametric model, and provide a statistically and computationally efficient algorithm that achieves sublinear regret.

## 1 Introduction

The Reinforcement Learning (RL) paradigm involves a learning agent interacting with an unknown dynamical environment over multiple time steps. The learner receives a reward signal after each step which it uses to improve its performance over time. This formulation of RL has had significant empirical success in the recent past [24, 23, 33, 32].

While this empirical success is encouraging, as RL starts to tackle a more wide-ranging class of consequential real-world problems, such as self-driving cars, supply chains, and medical care, a new set of challenges arise. Foremost among them is the lack of a well-specified reward signal associated with every state-action pair in many real-world settings. For example, consider a robot manipulation task where the robot must fold a pile of clothes. It is not clear how to design a useful reward signal that aids the robot to learn to complete this task. However, it is fairly easy to check whether the task was successfully completed (that is, whether the clothes were properly folded) and provide feedback at the end of the episode.

This is a classical challenge but it is one that is often neglected in theoretical treatments of RL. To address this challenge we introduce a framework for RL that eschews the need for a Markovian reward signal at every step and provides the learner only with binary feedback based on its complete trajectory in an episode. In our framework, the learner interacts with the environment for a fixed number of time steps $(H)$ in each episode to produce a trajectory $(\tau)$ which is the collection of all

35th Conference on Neural Information Processing Systems (NeurIPS 2021).

---

[*]Equal contribution.

states visited and actions taken in these rounds. At the end of the episode a binary reward $y_\tau \in \{0, 1\}$ is drawn from an unknown distribution $\mathbb{Q}(\cdot|\tau)$ and handed to the learner. This protocol continues for $N$ episodes and the learner's goal is to maximize the number of expected binary "successes."

One approach to deal with the lack of a reward function in the literature is Inverse Reinforcement Learning [25], which uses demonstrations of good trajectories to learn a reward function. However, this approach is difficult to use when good demonstrations are either prohibitively expensive or difficult to obtain. Another closely related line of work studies reinforcement learning with preference feedback [2, 15, 3, 5, 37, 26, 38]. Our framework provides the learner with an even weaker form of feedback than that studied in this line of work. Instead of providing preferences between trajectories, we only inform the learner whether the task was completed successfully or not at the end.

To study whether it is possible to learn under such drastically limited feedback we study the case where the conditional rewards $(y_\tau)$ are drawn from an unknown logistic model (see Assumption 2.1). Under this assumption we show that learning is possible—we provide an optimism-based algorithm that achieves sublinear regret (see Theorem 3.2). Technically our theory leverages recent results of Russac et al. [31] for the online estimation of the parameters of the underlying logistic model, and combining them with the UCBVI algorithm [4] to obtain regret bounds. Under an explorability assumption we also show that our algorithm is computationally efficient and we provide a dynamic programming algorithm to solve for the optimistic policy at every episode.

We note that Efroni et al. [11] study a similar problem to ours, such that a reward is revealed only at the end of the episode, but they assume that there exists an underlying linear model that determines the reward associated with each state-action pair, and reward revealed to the learner is the sum of rewards over the state-action pairs with added stochastic noise. This assumption ensures that the reward function is Markovian, and allows them to use an online linear bandit algorithm [1] to directly estimate the underlying reward function. This is not possible in our setting since we do not assume the existence of an underlying Markovian reward function. Cohen et al. [6] provided an algorithm that learns in this setting even when the noise is adversarially chosen. An open problem posed by Efroni et al. [11] was to find an algorithm that learns in this setting of reinforcement learning, with once per episode feedback, when the rewards are drawn from an unknown generalized linear model (GLM). In this paper we consider a specific GLM—the logistic model.

The remainder of the paper is organized as follows. In Section 2 we introduce notation and describe our setting. In Section 3 we present our algorithm and main results. Under an explorability assumption we prove that our algorithm is computationally efficient (in Appendix E). Section 4 points to other related work and we conclude with a discussion in Section 5. Other technical details, proofs and experiments are deferred to the appendix.

## 2 Preliminaries

This section presents notational conventions and a description of the setting.

### 2.1 Notation

For any $k \in \mathbb{N}$ we denote the set $\{1, \dots, k\}$ by $[k]$. Given any set $\mathcal{T}$, let $\Delta_\mathcal{T}$ denote the simplex over this set. Given a vector $\mathbf{v}$, for any $p \in \mathbb{N}$, let $\|\mathbf{v}\|_p$ denote the $\ell_p$ norm of the vector. Given a vector $\mathbf{v}$ and positive semi-definite matrix $\mathbf{M}$, define $\|\mathbf{v}\|_\mathbf{M} := \sqrt{\mathbf{v}^\top \mathbf{M} \mathbf{v}}$. Given a matrix $\mathbf{M}$ let $\|\mathbf{M}\|_{op}$ denote its operator norm. For any positive semi-definite matrix $\mathbf{M}$ we use $\lambda_{\max}(\mathbf{M})$ and $\lambda_{\min}(\mathbf{M})$ to denote its maximum and minimum eigenvalues respectively. We will use $C_1, C_2, \dots$ to denote absolute constants whose values are fixed throughout the paper, and $c, c', \dots$ to denote "local" constants, which may take different values in different contexts. We use the standard "big Oh notation" [see, e.g., 7].

### 2.2 The Setting

We study a Markov decision process (MDP) $\mathcal{M} = (\mathcal{S}, \mathcal{A}, \mathbb{P}, H)$, where $\mathcal{S}$ is the set of states, $\mathcal{A}$ is the set of actions, $\mathbb{P}(\cdot|s, a)$ is the law that governs the transition dynamics given a state and action pair

$(s, a)$, and $H \in \mathbb{N}$ is the length of an episode. Both the state space $\mathcal{S}$ and action space $\mathcal{A}$ are finite in our paper. The learner's trajectory $\tau$ is the concatenation of all states and actions visited during an episode; that is, $\tau := (s_1, a_1, \cdots, s_H, a_H)$. Given any $h \in [H]$ and trajectory $\tau$, a sub-trajectory $\tau_h := (s_1, a_1, \ldots, s_h, a_h)$ is all the states and actions taken up to step $h$. Also set $\tau_0 := \emptyset$. Let $\tau_{h:H} := (s_h, a_h, \ldots, s_H, a_H)$ denote the states and action from step $h$ until the end of the episode. Let $\Gamma$ be the set of all possible trajectories $\tau$. Analogously, for any $h \in [H]$ let $\Gamma_h$ be the set of all sub-trajectories up to step $h$. At the start of each episode the initial state $s_1$ is drawn from a fixed distribution $\rho$ that is known to the learner.

At the end of an episode the trajectory $\tau$ gets mapped to a feature map $\phi(\tau) \in \mathbb{R}^d$. We also assume that the learner has access to this feature map $\phi$. Here are two examples of feature maps:

1. **Direct parametrization:** Without loss of generality assume that $\mathcal{S} = \{1, \ldots, |\mathcal{S}|\}$ and $\mathcal{A} = \{1, \ldots, |\mathcal{A}|\}$. The feature map $\phi(\tau) = \sum_{h=1}^{H} \phi_h(s_h, a_h)$, where the per-step maps $\phi_h(s, a) \in \mathbb{R}^{|\mathcal{S}||\mathcal{A}|H}$ are defined as follows:

$$(\phi_h(s, a))_j = \begin{cases} 1 & \text{if } j = (h-1)|\mathcal{S}||\mathcal{A}| + (s-1)|\mathcal{A}| + a, \\ 0 & \text{otherwise.} \end{cases}$$

   The complete feature map $\phi(\tau) \in \mathbb{R}^{|\mathcal{S}||\mathcal{A}|H}$ is therefore an encoding of the trajectory $\tau$.

2. **Reduced parametrization:** Any trajectory $\tau$ is associated with a feature $\phi(\tau) \in \mathbb{R}^d$, where $d < |\mathcal{S}||\mathcal{A}|H$.

After the completion of an episode the learner is given a random binary reward $y_\tau \in \{0, 1\}$. Let $\mathbf{w}_\star \in \mathbb{R}^d$ be a vector that is unknown to the learner. We study the case where the rewards are drawn from a binary logistic model as described below.

**Assumption 2.1** (Logistic model). *Given any trajectory $\tau \in \Gamma$, the rewards are said to be drawn from a logistic model if the law of $y_\tau | \tau$ is*

$$y_\tau | \tau = \begin{cases} 1 & w.p. \quad \mu\left(\mathbf{w}_\star^\top \phi(\tau)\right) \\ 0 & w.p. \quad 1 - \mu\left(\mathbf{w}_\star^\top \phi(\tau)\right), \end{cases} \tag{1}$$

*where for any $z \in \mathbb{R}$, $\mu(z) = \frac{1}{1+\exp(-z)}$ is the logistic function. We shall refer to $\mathbf{w}_\star$ as the "reward parameters."*

We make the following boundedness assumptions on the features and reward parameters.

**Assumption 2.2** (Bounded features and parameters). *We assume that*

- $\|\mathbf{w}_\star\|_2 \leq B$ *for some known value $B > 0$ and*

- *for all $\tau \in \Gamma$, $\|\phi(\tau)\|_2 \leq 1$.*

We note that such boundedness assumptions are standard in the logistics bandits literature [13, 31, 14].

A policy $\pi$ is a collection of per-step policies $(\pi_1, \ldots, \pi_H)$ such that

$$\pi_h : \Gamma_{h-1} \times \mathcal{S} \to \Delta_\mathcal{A}.$$

If the agent is using the policy $\pi$ then at round $h$ of the episode the learner plays according to the policy $\pi_h$. We let $\Pi_h$ denote the set of all valid policies at step $h$ and let $\Pi$ denote the set of valid policies over the trajectory. Let $\mathbb{P}^\pi(\cdot|s_1)$ denote the joint probability distribution over the learner's trajectory $\tau$ and the reward $y_\tau$ when the learner plays according to the policy $\pi$ and the initial state is $s_1$. Often when the initial state is clear from the context we will refer to $\mathbb{P}^\pi(\cdot|s_1)$ by simply writing $\mathbb{P}^\pi$. Also with some abuse of notation we will sometimes let $\mathbb{P}^\pi$ denote the distribution of the trajectory and the reward where the initial state is drawn from the distribution $\rho$.

Given an initial state $s \in \mathcal{S}$ the value function corresponding to a policy $\pi$ is

$$V^\pi(s) := \mathbb{E}_{y_\tau, \tau \sim \mathbb{P}^\pi}[y_\tau \mid s_1 = s] = \mathbb{E}_{\tau \sim \mathbb{P}^\pi}\left[\mu\left(\mathbf{w}_\star^\top \phi(\tau)\right) \mid s_1 = s\right],$$

where the second equality follows as the mean of $y_\tau$ conditioned on $\tau$ is $\mu(\mathbf{w}_\star^\top \phi(\tau))$. With some abuse of notation we denote the average value function as $V^\pi := \mathbb{E}_{s_1 \sim \rho}[V^\pi(s_1)]$.

Define the optimal policy as $\pi_\star \in \arg\max_{\pi \in \Pi} V^\pi$. It is worth noting that in our setting the optimal policy may be *non-Markovian*. The learner plays for a total of $N$ episodes. The policy played in episode $t \in [N]$ is $\pi^{(t)}$ and its value function is $V^{(t)} := V^{\pi^{(t)}}$. Also define the value function for the optimal policy to be $V_\star := V^{\pi_\star}$. Our goal shall be to control the regret of the learner, which is defined as

$$\mathcal{R}(N) := \sum_{t=1}^{N} V_\star - V^{(t)}. \tag{2}$$

The trajectories in these $N$ episodes are denoted by $\{\tau^{(t)}\}_{t=1}^{N}$ and rewards received are denoted by $\{y^{(t)}\}_{t=1}^{N}$.

## 3   Optimistic Algorithms that Use Trajectory Labels

We now present an algorithm to learn from labeled trajectories. Throughout this section we assume that both Assumptions 2.1 and 2.2 are in force.

The derivative of the logistic function is $\mu'(z) = \frac{\exp(-z)}{(1+\exp(-z))^2}$, and therefore, $\mu$ is $1/4$-Lipschitz. The following quantity will play an important role in our bounds

$$\kappa := \max_{\tau \in \Gamma} \sup_{\mathbf{w}:\|\mathbf{w}\| \leq B} \frac{1}{\mu'(\mathbf{w}^\top \phi(\tau))}.$$

A consequence of Assumption 2.2 is that $\kappa \leq \exp(B)$. We briefly note that $\kappa$ is a measure of curvature of the logistic model. It also plays an important role in the analysis of logistic bandit algorithms [13, 31].

Since the true reward parameter $\mathbf{w}_\star$ is unknown we will estimate it using samples. At any episode $t \in [N]$, a natural way of computing an estimator of $\mathbf{w}_\star$, given past trajectories $\{\tau^{(q)}\}_{q \in [t-1]}$ and labels $\{y^{(q)}\}_{q \in [t-1]}$, is by minimizing the $\ell_2$-regularized cross-entropy loss:

$$\mathcal{L}_t(\mathbf{w}) := -\sum_{q=1}^{t-1} y^{(q)} \log\left(\mu\left(\mathbf{w}^\top \phi(\tau^{(q)})\right)\right) - (1 - y^{(q)}) \log\left(1 - \mu\left(\mathbf{w}^\top \phi(\tau^{(q)})\right)\right) + \frac{\|\mathbf{w}\|_2^2}{2}.$$

This function is strictly convex and its minimizer is defined to be

$$\widehat{\mathbf{w}}_t := \arg\min_{\mathbf{w} \in \mathbb{R}^d} \mathcal{L}_t(\mathbf{w}). \tag{3}$$

Define a design matrix at every episode

$$\boldsymbol{\Sigma}_1 := \kappa \mathbf{I}, \quad \text{and} \quad \boldsymbol{\Sigma}_t := \kappa \mathbf{I} + \sum_{q=1}^{t-1} \phi(\tau^{(q)})\phi(\tau^{(q)})^\top, \quad \text{for all } t \geq 1.$$

Further, define the confidence radius $\beta_t(\delta)$ as follows

$$\beta_t(\delta) := \left(1 + B + \rho_t(\delta)\left(\sqrt{1+B} + \rho_t(\delta)\right)\right)^{3/2} \tag{4}$$

$$\text{where,} \quad \rho_t(\delta) := d\log\left(4 + \frac{4t}{d}\right) + 2\log\left(\frac{N}{\delta}\right) + \frac{1}{2}.$$

We adapt a result due to Russac et al. [31, Proposition 7] who studied the online logistic bandits problem to establish that at every episode and every trajectory the difference between $\mu(\mathbf{w}_\star^\top \phi(\tau))$ and $\mu(\widehat{\mathbf{w}}_t^\top \phi(\tau))$ is small.

**Lemma 3.1.** *For any $\delta \in (0,1]$, define the event*

$$\mathcal{E}_\delta := \left\{ \text{for all } t \in [N], \tau \in \Gamma : \left|\mu(\mathbf{w}_\star^\top \phi(\tau)) - \mu(\widehat{\mathbf{w}}_t^\top \phi(\tau))\right| \leq \sqrt{\kappa}\beta_t(\delta)\|\phi(\tau)\|_{\boldsymbol{\Sigma}_t^{-1}} \right\}. \tag{5}$$

*Then $\mathbb{P}(\mathcal{E}_\delta) \geq 1 - \delta$.*

We provide a proof in Appendix B.2. The proof follows by simply translating [31, Proposition 7] into our setting. We note that we specifically adapt these recent results by Russac et al. [31] since they directly apply to $\widehat{\mathbf{w}}_t$, the minimizer of the $\ell_2$-regularized cross-entropy loss. In contrast, previous work on the logistic bandits problem [see, e.g., 14, 13] established confidence sets for an estimator that was obtained by performing a non-convex (and potentially computationally intractable) projection of $\widehat{\mathbf{w}}_t$ onto the ball of Euclidean radius $B$.

Our algorithm shall construct an estimate of the transition dynamics $\widehat{\mathbb{P}}_t$. Let $N_t(s, a)$ be the number of times that the state-action pair $(s, a)$ is encountered before the start of episode $t$, and let $N_t(s'; s, a)$ be the number of times the learner encountered the state $s'$ after taking action $a$ at state $s$ before the start of episode $t$. Define the estimator of the transition dynamics as follows:

$$\widehat{\mathbb{P}}_t(s'|a, s) := \frac{N_t(s'; s, a)}{N_t(s, a)}. \tag{6}$$

Also define the state-action bonus at episode $t$

$$\xi_{s,a}^{(t)} := \min \left\{ 2, 4\sqrt{\frac{\log\left(\frac{6(|\mathcal{S}||\mathcal{A}|H)^H (8NH^2)^{|\mathcal{S}|} \log(N_t(s,a))}{\delta}\right)}{N_t(s, a)}} \right\}. \tag{7}$$

In this definition whenever $N_t(s, a) = 0$, that is, when a state-action pair hasn't been visited yet, we define $\xi_{s,a}^{(t)}$ to be equal to 2. Finally, we define the optimistic reward functions

$$\bar{\mu}_t(\mathbf{w}, \tau) := \min \left\{ \mu\left(\mathbf{w}^\top \phi(\tau)\right) + \sqrt{\kappa}\beta_t(\delta)\|\phi(\tau)\|_{\mathbf{\Sigma}_t^{-1}}, 1 \right\} \quad \text{and} \tag{8a}$$

$$\widetilde{\mu}_t(\mathbf{w}, \tau) := \bar{\mu}_t(\mathbf{w}, \tau) + \sum_{h=1}^{H-1} \xi_{s_h, a_h}^{(t)}. \tag{8b}$$

The first reward function $\bar{\mu}_t$ is defined as above to account for the uncertainty in the predicted value of $\mathbf{w}_\star$ in light of Lemma 3.1, and the second reward function $\widetilde{\mu}_t$ is designed to account for the error in the estimation of the transition dynamics $\mathbb{P}$. With these additional definitions in place we are ready to present our algorithms and main results.

## 3.1 UCBVI with Trajectory Labels

Our first algorithm is an adaptation of the UCBVI algorithm [4] to our setting with labeled trajectories.

---

**Algorithm 1:** UCBVI with trajectory labels.

---

1 **Input:** State and action spaces $\mathcal{S}, \mathcal{A}$.

2 **Initialize** $\widehat{\mathbb{P}}_1 = \mathbf{0}$, visitation set $\mathcal{K} = \emptyset$.

3 **for** $t = 1, \cdots$ **do**

4    1. Calculate the $\widehat{\mathbf{w}}_t$ by solving equation (3).

5    2. If $t > 1$, compute $\pi^{(t)}$

$$\pi^{(t)} \in \arg\max_{\pi \in \Pi} \mathbb{E}_{s_1 \sim \rho, \, \tau \sim \widehat{\mathbb{P}}_t^\pi(\cdot|s_1)} \left[\widetilde{\mu}_t(\widehat{\mathbf{w}}_t, \tau)\right]. \tag{9}$$

   Else for all $h, s, \tau_{h-1} \in [H] \times \mathcal{S} \times \Gamma_{h-1}$, set $\pi_h^{(1)}(\cdot|s, \tau_{h-1})$ to be the uniform distribution over the action set.

6    3. Observe the trajectory $\tau^{(t)} \sim \mathbb{P}^{\pi^{(t)}}$ and update the design matrix

$$\mathbf{\Sigma}_{t+1} = \kappa \mathbf{I} + \sum_{q=1}^{t} \phi(\tau^{(q)})\phi(\tau^{(q)})^\top. \tag{10}$$

7    4. Update the visitation set $\mathcal{K} = \{(s, a) \in \mathcal{S} \times \mathcal{A} : N_t(s, a) > 0\}$.

8    5. For all $(s, a) \in \mathcal{K}$, update $\widehat{\mathbb{P}}_{t+1}(\cdot|s, a)$ according to equation (6).

9    6. For all $(s, a) \notin \mathcal{K}$, set $\widehat{\mathbb{P}}_{t+1}(\cdot|s, a)$ to be the uniform distribution over states.

---

**Theorem 3.2.** *For any $\bar{\delta} \in (0,1]$, set $\delta = \bar{\delta}/(6N)$ then under Assumptions 2.1 and 2.2 the regret of Algorithm 1 is upper bounded as follows:*

$$\mathcal{R}(N) \leq \widetilde{O}\left(\left[H\sqrt{(H+|\mathcal{S}|)|\mathcal{S}||\mathcal{A}|} + H^2 + \sqrt{\kappa}d(d^3 + B^{3/2})\right]\sqrt{N} + (H+|\mathcal{S}|)H|\mathcal{S}||\mathcal{A}|\right),$$

*with probability at least $1 - \bar{\delta}$.*

The regret of our algorithm scales with $\sqrt{N}$ and polynomially with the horizon, number of states, number of actions, $\kappa$, dimension of the feature maps and length of the reward parameters ($B$). The minimax regret in the standard episodic reinforcement learning is $O(\sqrt{H|\mathcal{S}||\mathcal{A}|N})$ [27, 4]. Here we pay for additional factors in $H$, $|\mathcal{S}|$ and $\kappa$ since our rewards are non-Markovian and are revealed to the learner only at the end of the episode. We provide a proof of this theorem in Appendix B. For a more detailed bound on the regret with the logarithmic factors and constants specified we point the interested reader to inequality (41) in the appendix.

**Proof sketch.** First we show that with high probability at each episode the value function of the optimal policy $V_\star$ is upper bounded by $\widetilde{V}^{(t)} := \mathbb{E}_{s_1 \sim \rho, \, \tau \sim \widehat{\mathbb{P}}_t^{\pi(t)}(\cdot|s_1)}[\widetilde{\mu}_t(\widehat{\mathbf{w}}_t, \tau)]$ (the value function of the policy $\pi^{(t)}$ when the rewards are dictated by $\widetilde{\mu}_t$ and the transition dynamics are given by $\widehat{\mathbb{P}}_t$). Then we provide a high probability bound on the difference between the optimistic value function $\widetilde{V}^{(t)}$ and the true value function $V^{(t)}$ to obtain our upper bound on the regret. In both of these steps we need to relate expectations with respect to the true transition dynamics $\mathbb{P}$ to expectations with respect to the empirical estimate of the transition dynamics $\widehat{\mathbb{P}}_t$. We do this by using our concentration results: Lemmas B.1 and B.2 proved in the appendix. While analogs of these concentration lemmas do exist in previous theoretical studies of episodic reinforcement learning, here we had to prove these lemmas in our setting with non-Markovian trajectory-level feedback (which explains why we pay extra factors in $H$ and $|\mathcal{S}|$).

## 3.2 UCBVI with Added Exploration

Although the regret of Algorithm 1 is sublinear it is not guaranteed to be computationally efficient since finding the optimistic policy $\pi^{(t)}$ (in equation (9)) at every episode might prove to be difficult. In this section, we will show that when the features are sum-decomposable and the MDP satisfies an explorability assumption then it will be possible to find a computationally efficient algorithm with sublinear regret (albeit with a slightly worse scaling with the number of episodes $N$).

**Assumption 3.3** (Sum-decomposable features). *We assume that the feature maps $\phi \in \mathbb{R}^d$ are sum-decomposable over the different steps of the trajectory, that is, $\phi(\tau) = \sum_{h=1}^{H} \phi_h(s_h, a_h)$.*

Under this assumption, given any $\mathbf{w} \in \mathbb{R}^d$ and any trajectory $\tau \in \Gamma$, $\mathbf{w}^\top \phi(\tau) = \sum_{h=1}^{H} \mathbf{w}^\top \phi_h(s_h, a_h)$. We stress that even under this sum-decomposablity assumption, the optimal policy is potentially non-Markovian due to the presence of the logistic map that governs the reward.

We also make the following explorability assumption.

**Assumption 3.4** (Explorability). *For any $s, s' \in \mathcal{S}$, $a, a' \in \mathcal{A}$, and $h \neq h' \in [H]$, suppose that*

$$\phi_h(s,a)^\top \phi_{h'}(s',a') = 0.$$

*Further assume that there exists $\omega \in (0,1)$ such that for any unit vector $\mathbf{v} \in \mathbb{R}^d$ we have that*

$$\sup_{\pi \in \Pi} \mathbb{E}_{s_1 \sim \rho, \tau \sim \mathbb{P}^\pi}\left[\sum_{h \in [H]} \mathbf{v}^\top \phi_h(s_h, a_h)\right] \geq \omega.$$

In a setting with Markovian rewards a similar assumption has been made previously by Zanette et al. [40]. This assumption allows us to efficiently "explore" the feature space, and construct a sum-decomposable bonus $\sqrt{\kappa}\beta_t(\delta)\sum_{h=1}^{H}\|\phi_h(s_h,a_h)\|_{\Sigma_t^{-1}}$ that we will use instead of

$\sqrt{\kappa}\beta_t(\delta)\|\phi(\tau)\|_{\boldsymbol{\Sigma}_t^{-1}}$ in the definition of $\bar{\mu}_t$ (see equation (8a)). Define the reward functions

$$\bar{\mu}_t^{\mathsf{sd}}(\mathbf{w},\tau) := \min\left\{\mu\left(\mathbf{w}^\top\phi(\tau)\right) + \sqrt{\kappa}\beta_t(\delta)\sum_{h=1}^{H}\|\phi_h(s_h,a_h)\|_{\boldsymbol{\Sigma}_t^{-1}}, 1\right\} \quad \text{and} \quad (11\text{a})$$

$$\widetilde{\mu}_t^{\mathsf{sd}}(\mathbf{w},\tau) := \bar{\mu}_t^{\mathsf{sd}}(\mathbf{w},\tau) + \sum_{h=1}^{H-1}\xi_{s_h,a_h}^{(t)}. \tag{11b}$$

To prove a regret bound for an algorithm that uses these rewards our first step shall be to prove that the sum-decomposable bonus also leads to an optimistic reward function (that is, the value function defined by these rewards sufficiently over-estimates the true value function). To this end, we will first use Algorithm 2 to find an exploration mixture policy $\bar{U}$ and play according to it at episode $t$ with probability $1/t^{1/3}$. This policy $\bar{U}$ will be such that the minimum eigenvalue of

$$\mathbb{E}_{s_1\sim\rho,\ \tau\sim\mathbb{P}^{\bar{U}}(\cdot|s_1)}\left[\phi(\tau)\phi(\tau)^\top\right] \tag{12}$$

is lower bounded by a function of $d$, $\omega$ and $N$ (see Lemma 3.5). This property shall allow us to upper bound the condition number of the design matrix $\boldsymbol{\Sigma}_t$ and subsequently ensure that the rewards $\bar{\mu}_t^{\mathsf{sd}}$ and $\widetilde{\mu}_t^{\mathsf{sd}}$ are optimistic. Given a unit vector $\mathbf{v}$ define a reward function at step $h$ as follows:

$$r_h^{\mathbf{v}}(s,a) := \mathbf{v}^\top\phi_h(s,a). \tag{13}$$

Let $r^{\mathbf{v}} := (r_1^{\mathbf{v}},\ldots,r_H^{\mathbf{v}})$ be a reward function over the entire episode. As a subroutine Algorithm 2 uses the EULER algorithm [39]. (We briefly note that other reinforcement learning algorithms with PAC or regret guarantees [e.g., 4, 19] could also be used here in place of EULER.)

---

**Algorithm 2:** Find exploration mixture.

1 **Input:** Initial unit vector $\mathbf{v}_1$, Exploration lower bound $\omega$, number of EULER episodes $N_{\mathsf{EUL}}$, number of evaluation episodes $N_{\mathsf{EVAL}}$.

2 **Initialize:** $\mathbf{A}_0 = \frac{\omega^2}{16}\mathbf{I}$, $n = 0$ and $\lambda_{\min} = \inf_{\mathbf{z}\in\mathbb{R}^d}\mathbf{z}^\top\mathbf{A}_0\mathbf{z}$.

3 **while** $\lambda_{\min} < \frac{\omega^2}{8}$ **do**

4      Update the counter $n \leftarrow n+1$.

5      Set $U_n \leftarrow \mathsf{EULER}(\{r^{\mathbf{v}_n},N_{\mathsf{EUL}})$ //run EULER for $N_{\mathsf{EUL}}$ episodes.

6      **for** $t=1,\ldots,N_{\mathsf{EVAL}}$ *episodes* **do**

7          Sample a trajectory $\tau_n^{(t)} \sim \rho \times \mathbb{P}^{U_n}$.

8      Calculate the average feature $\widehat{\mathbf{a}}_n = \sum_{t=1}^{N_{\mathsf{EVAL}}}\phi(\tau_n^{(t)})/N_{\mathsf{EVAL}}$.

9      Update the matrix $\mathbf{A}_n \leftarrow \mathbf{A}_{n-1} + \widehat{\mathbf{a}}_n\widehat{\mathbf{a}}_n^\top$.

10      Update the minimum eigenvalue: $\lambda_{\min} \leftarrow \inf_{\mathbf{z}\in\mathbb{R}^d}\mathbf{z}^\top\mathbf{A}_n\mathbf{z}$.

11      Set $\mathbf{v}_n$ to be the minimum eigenvector of $\mathbf{A}_n$.

12 Set $n_{\mathsf{loop}} = n$.

13 **Return:** (i) $\bar{U} = \mathsf{Unif}(U_1,\cdots,U_{n_{\mathsf{loop}}})$ //the uniform mixture over the policies;

14 (ii) $N_{\exp} = n_{\mathsf{loop}} \times (N_{\mathsf{EUL}} + N_{\mathsf{EVAL}})$ //total number of episodes.

---

**Lemma 3.5.** *There exist positive absolute constants $C_1$ and $C_2$ such that, under Assumptions 2.2, 3.3 and 3.4, if Algorithm 2 is run with $N_{\mathsf{EUL}} = \dfrac{C_1|\mathcal{S}|^2|\mathcal{A}|H^2\log\left(\frac{|\mathcal{S}||\mathcal{A}|N^2 d}{\delta\omega^2}\right)}{\omega^2}$ and $N_{\mathsf{EVAL}} = \dfrac{C_2 d^3\log^3\left(\frac{Nd^2}{\delta\omega^2}\right)}{\omega^4}$, and $N > \dfrac{d\log\left(1+\frac{16N}{d\omega^2}\right)}{\log(3/2)}\left(N_{\mathsf{EUL}} + N_{\mathsf{EVAL}}\right) =: \bar{N}_{\exp}$ then, with probability at least $1 - 2\delta$, we have $N_{\exp} \leq \bar{N}_{\exp}$ and furthermore:*

$$\mathbb{E}_{s_1\sim\rho,\ \tau\sim\mathbb{P}^{\bar{U}}(\cdot|s_1)}\left[\phi(\tau)\phi(\tau)^\top\right] \succeq \frac{\omega^2\log(3/2)}{32d\log\left(d\log\left(1+\frac{16N}{d\omega^2}\right)\right)}\mathbf{I}.$$

This lemma is proved in Appendix C. With this lemma in place we now present our modified algorithm under the explorability assumption. In the first few episodes this algorithm finds the exploration mixture policy $\bar{U}$. In a subsequent episode $t$ this algorithm acts according to the policy $\pi^{(t)}$ which maximizes the value function associated with the rewards $\widetilde{\mu}_t^{\mathsf{sd}}(\widehat{\mathbf{w}}_t,\tau)$ with probability $1 - \frac{1}{t^{1/3}}$. Otherwise it uses the exploration mixture policy $\bar{U}$.

---

**Algorithm 3:** UCBVI with trajectory labels and added exploration.

---

1 **Input:** State and action spaces $\mathcal{S}, \mathcal{A}$, Initial unit vector $\mathbf{v}_1$, Exploration lower bound $\omega$, number of EULER episodes $N_{\mathsf{EUL}}$, number of evaluation episodes $N_{\mathsf{EVAL}}$.

2 **Initialize** $\widehat{\mathbb{P}}_1 = \mathbf{0}$, visitation set $\mathcal{K} = \emptyset$.

3 Find exploration mixture policy $\bar{U}$ in $N_{\exp}$ episodes by running Algorithm 2.

4 **for** $t = N_{\exp} + 1, \cdots, N$ **do**

5     1. Calculate $\widehat{\mathbf{w}}_t$ by solving equation (3).

6     2. If $t > N_{\exp} + 1$, compute $\pi^{(t)}$

$$\pi^{(t)} \in \arg\max_{\pi} \mathbb{E}_{s_1 \sim \rho, \ \tau \sim \widehat{\mathbb{P}}_t^\pi(\cdot | s_1)} \left[ \widetilde{\mu}_t^{\mathsf{sd}}(\widehat{\mathbf{w}}_t, \tau) \right]. \tag{14}$$

    Else for all $h, s, \tau_{h-1} \in [H] \times \mathcal{S} \times \Gamma_{h-1}$, set $\pi_h^{(1)}(\cdot | s, \tau_{h-1})$ to be the uniform distribution over the action set.

7     3. Sample $b_t = \begin{cases} 0 & \text{w.p. } 1 - \frac{1}{t^{1/3}}, \\ 1 & \text{w.p. } \frac{1}{t^{1/3}}. \end{cases}$

8     4. If $b_t = 1$ then set $\pi^{(t)} \leftarrow \bar{U}$.

9     5. Observe the trajectory $\tau^{(t)} \sim \mathbb{P}^{\pi^{(t)}}$ and update the design matrix

$$\boldsymbol{\Sigma}_{t+1} = \kappa \mathbf{I} + \sum_{q=N_{\exp}+1}^{t} \phi(\tau^{(q)}) \phi(\tau^{(q)})^\top. \tag{15}$$

10     6. Update the visitation set $\mathcal{K} = \{(s, a) \in \mathcal{S} \times \mathcal{A} : N_t(s, a) > 0\}$.

11     7. For all $(s, a) \in \mathcal{K}$, update $\widehat{\mathbb{P}}_{t+1}(\cdot | s, a)$ according to equation (6).

12     8. For all $(s, a) \notin \mathcal{K}$, set $\widehat{\mathbb{P}}_{t+1}(\cdot | s, a)$ to be the uniform distribution over states.

---

The following is our regret bound for Algorithm 3.

**Theorem 3.6.** *For any $\bar{\delta} \in (0, 1]$, set $\delta = \bar{\delta}/(12N)$. Under Assumptions 2.1, 2.2, 3.3 and 3.4, and for all $N > \bar{N}_{\exp}$ (see its definition in Lemma 3.5) if Algorithm 3 is run with the parameters $N_{\mathsf{EUL}}$ and $N_{\mathsf{EVAL}}$ set as specified in Lemma 3.5 then its regret is upper bounded as follows:*

$$\mathcal{R}(N) \leq \widetilde{O}\left( \frac{\sqrt{\kappa H} d}{\omega}(d^3 + B^{3/2})N^{2/3} + \left[ H\sqrt{(H + |\mathcal{S}|)|\mathcal{S}||\mathcal{A}|} + H^2 \right] \sqrt{N} \right.$$
$$\left. + (H + |\mathcal{S}|)H|\mathcal{S}||\mathcal{A}| + \frac{d^2}{\omega^2}\left( \frac{d^2}{\omega^2} + |\mathcal{S}|^2|\mathcal{A}|H^2 \right) \right),$$

*with probability at least $1 - \bar{\delta}$.*

The proof of Theorem 3.6 is in Appendix D. For a more detailed bound on the regret with the logarithmic factors and constants specified we point the interested reader to inequality (58) in the appendix. The bound on the regret of this algorithm scales with $N^{2/3}$ up to poly-logarithmic factors. This is larger than the $\sqrt{N}$ regret bound (again up to poly-logarithmic factors) that we proved above for Algorithm 1 since here the learner plays according to the exploration policy $\bar{U}$ with probability $1/t^{1/3}$ throughout the run of the algorithm. However, the next proposition shows that by using the sum-decomposable reward function $\widetilde{\mu}_t^{\mathsf{sd}}$ the policy $\pi^{(t)}$ defined in equation (14) can be efficiently approximated.

**Proposition 3.7.** *For any $t \in [N]$ define $\widetilde{V}_t^{\mathsf{sd}}(\pi) := \mathbb{E}_{s_1 \sim \rho, \ \tau \sim \widehat{\mathbb{P}}_t^\pi(\cdot | s_1)} \left[ \widetilde{\mu}_t^{\mathsf{sd}}(\widehat{\mathbf{w}}_t, \tau) \right]$. Given any $\varepsilon > 0$, under Assumptions 2.2, 3.3 and 3.4 it is possible to find a policy $\widehat{\pi}^{(t)}$ that satisfies*

$$\widetilde{V}_t^{\mathsf{sd}}(\pi^{(t)}) - \widetilde{V}_t^{\mathsf{sd}}(\widehat{\pi}^{(t)}) \leq \varepsilon,$$

*using at most* $\mathsf{poly}\left( |\mathcal{S}|, |\mathcal{A}|, H, d, B, \|\widehat{\mathbf{w}}_t\|_2, \frac{1}{\varepsilon}, \log\left(\frac{N}{\delta}\right) \right)$ *time and memory.*

We describe the approximate dynamic programming algorithm that can be used to find this policy $\widehat{\pi}^{(t)}$ and present a proof of this proposition in Appendix E. We also note that if we use an

$\varepsilon$-approximate policy $\widehat{\pi}^{(t)}$ instead of $\pi^{(t)}$ in Algorithm 3 then its regret increases by an additive factor of at most $\varepsilon N$. (It is possible to easily check this by inspecting the proof of Theorem 3.6.) Thus, for example a choice of $\varepsilon = 1/N^{1/3}$ ensures that the regret of Algorithm 3 is bounded by $O(N^{2/3})$ with high probability if the approximate policy $\widehat{\pi}^{(t)}$ (which can be found efficiently) is used instead.

## 4 Additional Related Work

There have been many theoretical results that analyze regret minimization in standard episodic reinforcement [18, 29, 16, 28, 4, 19, 8, 39, 34, 10, 30]. Recently Efroni et al. [12] introduced a framework of "sequential budgeted learning" which includes as a special case the setting of episodic reinforcement learning with the constraint that the learner is allowed to query the reward function only a limited number of times per episode. They show learning is possible in this setting by using a modified UCBVI algorithm.

As stated above to estimate the reward parameter we rely on the recent results by Russac et al. [31] who in term built on earlier work [13, 14] that analyzed the GLM-UCB algorithm. Dong et al. [9] provided and analyzed a Thompson sampling approach for the logistic bandits problem.

## 5 Discussion

We have shown that efficient learning is possible when the rewards are non-Markovian and delivered to the learner only once per episode. It would be interesting to see if one can establish guarantees under more general reward models than the logistic model that we study here. Another interesting question is if faster rates of learning are possible when the learner obtains ranked trajectories (that is, moving beyond binary labels).

### Acknowledgments

The authors would like to thank Louis Faury, Tor Lattimore, Yoan Russac and Csaba Szepesvári for helpful conversations regarding the literature on logistic bandits. We thank Yonathan Efroni, Nadav Merlis and Shie Mannor for pointing us to prior related work.

### Funding

We gratefully acknowledge the support of the NSF through the grant DMS-2023505 in support of the FODSI Institute.

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
