# Contents

## A   Technical Lemmas

In this section we collect some useful technical results used in the proofs that follow. First we present a time-uniform martingale concentration inequality.

**Lemma A.1.** *Let $\{x_t\}_{t=1}^{\infty}$ be a martingale difference sequence with $|x_t| \leq \zeta$ and let $\delta \in (0,1]$. Then with probability $1 - \delta$ for all $T \in \mathbb{N}$*

$$\sum_{t=1}^{T} x_t \leq 2\zeta\sqrt{T \log\left(\frac{6\log T}{\delta}\right)}.$$

**Proof** Observe that $\frac{|x_t|}{\zeta} \leq 1$. By invoking a time-uniform Hoeffding-style concentration inequality [17, Equation (11)] we find that

$$\mathbb{P}\left[\forall\, t \in \mathbb{N}\; :\; \sum_{t=1}^{T} \frac{x_t}{\zeta} \leq 1.7\sqrt{T\left(\log\log(T) + 0.72\log\left(\frac{5.2}{\delta}\right)\right)}\right] \geq 1 - \delta.$$

Rounding up the constants for the sake of simplicity we get

$$\mathbb{P}\left[\forall\, t \in \mathbb{N}\; :\; \sum_{t=1}^{T} x_t \leq 2\zeta\sqrt{T\left(\log\left(\frac{6\log(T)}{\delta}\right)\right)}\right] \geq 1 - \delta,$$

which establishes our claim. ∎

Next we state a matrix concentration theorem [35, Theorem 1.1].

**Theorem A.2** (Matrix Freedman inequality). *Consider a matrix martingale $\{\mathbf{Y}_k\}_{k=1}^{\infty}$ whose values are self adjoint matrices with dimension $d$ and let $\{\mathbf{X}_k\}_{k=1}^{\infty}$ be its difference sequence. Assume the difference sequence is uniformly bounded in the sense that:*

$$\lambda_{\max}(\mathbf{X}_k) \leq R \quad \text{almost surely for all } k = 1, 2 \ldots.$$

*Define the predictable quadratic variation process of the martingale*

$$\mathbf{W}_k := \sum_{j=1}^{k} \mathbb{E}\left[\mathbf{X}_j^2 \mid \mathbf{X}_1, \ldots, \mathbf{X}_{j-1}\right] \quad \text{for} \quad k = 1, 2, \ldots.$$

*Then for all $x \geq 0$ and $V \geq 0$,*

$$\mathbb{P}\left(\exists\, k : \lambda_{\max}(\mathbf{Y}_k) \geq x \text{ and } \|\mathbf{W}_k\|_{op} \leq V\right) \leq d \cdot \exp\left(\frac{-x^2/2}{V + Rx/3}\right).$$

The following result that bounds the norm of sequence of vectors in terms of the norm induced by its inverse Gram matrix.

**Lemma A.3** (Determinant Lemma). *For any sequence of vectors $\boldsymbol{x}^{(1)}, \ldots, \boldsymbol{x}^{(T)} \in \mathbb{R}^d$ such that $\|\boldsymbol{x}^{(q)}\|_2 \leq L$ for all $q \in [T]$. Given a $\lambda \geq 0$ define $\bar{\boldsymbol{\Sigma}}_1 := \lambda\mathbf{I}$ and for $t \in \{2, \ldots, T\}$ define $\bar{\boldsymbol{\Sigma}}_t := \lambda\mathbf{I} + \sum_{q=1}^{t-1} \boldsymbol{x}^{(q)}\boldsymbol{x}^{(q)\top}$. Then for all $T \in \mathbb{N}$*

$$\sum_{t=1}^{N} \|\phi(\tau^{(t)})\|_{\bar{\boldsymbol{\Sigma}}_t^{-1}}^2 \leq 2d\max\left\{1, \frac{1}{\lambda}\right\}\log\left(1 + \frac{TL^2}{\lambda d}\right)$$

*and*

$$\log\left(\frac{\det(\bar{\boldsymbol{\Sigma}}_t)}{\det(\lambda\mathbf{I})}\right) \leq d\log\left(1 + \frac{tL^2}{\lambda d}\right).$$

**Proof** The first part follows by combining the results of Lemmas 15 and 16 from [13]. The second part is a restatement of [22, Lemma 19.4]. ∎

## B    Regret Analysis of Algorithm 1

In this appendix we analyze the regret of Algorithm 1 and prove Theorem 3.2. We begin by establishing some useful concentration lemmas.

## B.1 Concentration Lemmas Required to Bound the Regret

We will now prove a lemma that relates the expectation of rewards between the true model $\mathbb{P}$ and an empirical model $\widehat{\mathbb{P}}_t$ when using any fixed policy $\pi$. Given any $\eta > 0$ define

$$\bar{\xi}_{s,a}^{(t)}(\eta) = \min\left\{2\eta, 4\eta\sqrt{\frac{H\log\left(|\mathcal{S}||\mathcal{A}|\right) + \log\left(\frac{6\log(N_t(s,a))}{\delta}\right)}{N_t(s,a)}}\right\}. \tag{16}$$

**Lemma B.1.** *Given any fixed policy $\pi \in \Pi$, and any scalar function $\check{\mu}_\tau$ that depends on the trajectory and satisfies $|\check{\mu}_\tau| \leq \eta$, with probability at least $1 - \delta$ for all $t \in \mathbb{N}$*

$$\mathbb{E}_{s_1\sim\rho,\ \tau\sim\mathbb{P}^\pi(\cdot|s_1)}[\check{\mu}_\tau] - \mathbb{E}_{s_1\sim\rho,\ \tau\sim\widehat{\mathbb{P}}_t^\pi(\cdot|s_1)}[\check{\mu}_\tau] \leq \mathbb{E}_{s_1\sim\rho,\ \tau\sim\widehat{\mathbb{P}}_t^\pi(\cdot|s_1)}\left[\sum_{h=1}^{H-1}\bar{\xi}_{s_h,a_h}^{(t)}(\eta)\right]. \tag{17}$$

**Proof** Define $\mathbb{P}_{(h)}^\pi$ to be a trajectory distribution where the initial state is $s_1 \sim \rho$, the state-action pairs up to the end of step $h$ are drawn from $\widehat{\mathbb{P}}_t^\pi$, and the state-action pairs from step $h+1$ up until the last step $H$ are drawn from $\mathbb{P}^\pi$. Notice that $\mathbb{P}_{(0)}^\pi(s_1, \cdot) = \rho(s_1)\mathbb{P}^\pi(\cdot|s_1)$ and $\mathbb{P}_{(H)}^\pi(s_1, \cdot) = \rho(s_1)\widehat{\mathbb{P}}_t^\pi(\cdot|s_1)$. Thus,

$$\mathbb{E}_{s_1\sim\rho,\ \tau\sim\mathbb{P}^\pi(\cdot|s_1)}[\check{\mu}_\tau] - \mathbb{E}_{s_1\sim\rho,\ \tau\sim\widehat{\mathbb{P}}_t^\pi(\cdot|s_1)}[\check{\mu}_\tau] = \sum_{h=1}^{H}\mathbb{E}_{\tau\sim\mathbb{P}_{(h-1)}^\pi}[\check{\mu}_\tau] - \mathbb{E}_{\tau\sim\mathbb{P}_{(h)}^\pi}[\check{\mu}_\tau]. \tag{18}$$

Consider the term where $h = 1$. The trajectory distributions $\mathbb{P}_{(0)}^\pi$ and $\mathbb{P}_{(1)}^\pi$ differ only their distributions of state-action pairs in step 1, thus,

$$\mathbb{E}_{\tau\sim\mathbb{P}_{(0)}^\pi}[\check{\mu}_\tau] - \mathbb{E}_{\tau\sim\mathbb{P}_{(1)}^\pi}[\check{\mu}_\tau]$$
$$= \mathbb{E}_{s_1\sim\rho}\left[\mathbb{E}_{a_1\sim\pi(\cdot|s_1)}\mathbb{E}_{\tau\sim\mathbb{P}_{(0)}^\pi}[\check{\mu}_\tau|(s_1,a_1)]\right] - \mathbb{E}_{s_1\sim\rho}\left[\mathbb{E}_{a_1\sim\pi(\cdot|s_1)}\mathbb{E}_{\tau\sim\mathbb{P}_{(0)}^\pi}[\check{\mu}_\tau|(s_1,a_1)]\right] = 0. \tag{19}$$

Consider any other term in this sum. Again the trajectory distributions $\mathbb{P}_{(h-1)}^\pi$ and $\mathbb{P}_{(h)}^\pi$ differ only their distributions of state-action pairs in step $h$ and hence

$$\mathbb{E}_{\tau\sim\mathbb{P}_{(h-1)}^\pi}[\check{\mu}_\tau] - \mathbb{E}_{\tau\sim\mathbb{P}_{(h)}^\pi}[\check{\mu}_\tau]$$
$$= \mathbb{E}_{s_1\sim\rho,\ \tau_{h-1}\sim\widehat{\mathbb{P}}_t^\pi(\cdot|s_1)}\left(\mathbb{E}_{\tau\sim\mathbb{P}_{(h-1)}^\pi}[\check{\mu}_\tau|\tau_{h-1}] - \mathbb{E}_{\tau\sim\mathbb{P}_{(h)}^\pi}[\check{\mu}_\tau|\tau_{h-1}]\right)$$
$$= \mathbb{E}_{s_1\sim\rho,\ \tau_{h-1}\sim\widehat{\mathbb{P}}_t^\pi(\cdot|s_1)}\left(\mathbb{E}_{s_h\sim\mathbb{P}(\cdot|s_{h-1},a_{h-1})}\left[\mathbb{E}_{a_h\sim\pi_h(\cdot|s_h,\tau_{h-1})}\mathbb{E}_{\tau\sim\mathbb{P}_{(h-1)}^\pi}[\check{\mu}_\tau|(s_h,a_h,\tau_{h-1})]\right]\right.$$
$$\left. - \mathbb{E}_{s_h\sim\widehat{\mathbb{P}}_t(\cdot|s_{h-1},a_{h-1})}\left[\mathbb{E}_{a_h\sim\pi_h(\cdot|s_h,\tau_{h-1})}\mathbb{E}_{\tau\sim\mathbb{P}_{(h-1)}^\pi}[\check{\mu}_\tau|(s_h,a_h,\tau_{h-1})]\right]\right). \tag{20}$$

Define the random variable

$$z_h(s, \tau_{h-1}') := \mathbb{E}_{a\sim\pi_h(\cdot|s,\tau_{h-1}')}\mathbb{E}_{\tau\sim\mathbb{P}_{(h-1)}^\pi}[\check{\mu}_\tau|\tau_h = \{s,a,\tau_{h-1}'\}].$$

Observe that $|z_h(s,\tau_{h-1}')| \leq \eta$, since $|\check{\mu}_\tau| \leq \eta$ by assumption. Furthermore, the distribution of $z_h(s,\tau_{h-1}')$ only depends on the true transition dynamics $\mathbb{P}$ and on the policy $\pi$ but it does not depend on the empirical estimate of the transition dynamics $\widehat{\mathbb{P}}_t$. With this definition in hand continuing from equation (20) we have

$$\mathbb{E}_{\tau\sim\mathbb{P}_{(h-1)}^\pi}[\check{\mu}_\tau] - \mathbb{E}_{\tau\sim\mathbb{P}_{(h)}^\pi}[\check{\mu}_\tau]$$
$$= \mathbb{E}_{s_1\sim\rho,\ \tau_{h-1}\sim\widehat{\mathbb{P}}_t^\pi(\cdot|s_1)}\left[\underbrace{\mathbb{E}_{s'\sim\mathbb{P}(\cdot|s_{h-1},a_{h-1})}[z_h(s',\tau_{h-1})] - \mathbb{E}_{s'\sim\widehat{\mathbb{P}}_t(\cdot|s_{h-1},a_{h-1})}[z_h(s',\tau_{h-1})]}\right]. \tag{21}$$

We will upper bound the term in the under-brace above with high probability uniformly over all sub-trajectories $\tau_{h-1}$.

Recall that $N_t(s,a)$ is the number of times the state action pair $(s,a)$ has been visited before episode $t$, and $N_t(s';s,a)$ is the number of times the state $s'$ is visited starting from state-action pair $(s,a)$ before episode $t$. When $N_t(s,a) > 0$, by definition $\widehat{\mathbb{P}}_t(s'|s,a) = \frac{N_t(s';s,a)}{N_t(s,a)}$. Thus for any fixed sub-trajectory $\tau_{h-1} \in \Gamma_{h-1}$ such that $N_t(s_{h-1,a_{h-1}}) > 0$ we have

$$\mathbb{E}_{s' \sim \mathbb{P}(\cdot|s_{h-1},a_{h-1})}[z_h(s',\tau_{h-1})] - \mathbb{E}_{s' \sim \widehat{\mathbb{P}}_t(\cdot|s_{h-1},a_{h-1})}[z_h(s',\tau_{h-1})]$$

$$= \mathbb{E}_{s' \sim \mathbb{P}(\cdot|s_{h-1},a_{h-1})}[z_h(s',\tau_{h-1})] - \sum_{s' \in \mathcal{S}} \frac{N_t(s';s_{h-1},a_{h-1})}{N_t(s_{h-1},a_{h-1})} z_h(s',\tau_{h-1})$$

$$= \frac{1}{N_t(s_{h-1},a_{h-1})}\left[N_t(s_{h-1},a_{h-1})\mathbb{E}_{s' \sim \mathbb{P}(\cdot|s_{h-1},a_{h-1})}[z_h(s',\tau_{h-1})]\right.$$

$$\left. - \sum_{s' \in \mathcal{S}} N_t(s';s_{h-1},a_{h-1})z_h(s',\tau_{h-1})\right]$$

$$\overset{(i)}{=} \frac{1}{N_t(s_{h-1},a_{h-1})}\left[N_t(s_{h-1},a_{h-1})\mathbb{E}_{s' \sim \mathbb{P}(\cdot|s_{h-1},a_{h-1})}[z_h(s',\tau_{h-1})] - \sum_{\ell=1}^{N_t(s_{h-1},a_{h-1})} z_h(s_\ell,\tau_{h-1})\right]$$

$$= \frac{1}{N_t(s_{h-1},a_{h-1})}\sum_{\ell=1}^{N_t(s_{h-1},a_{h-1})} \mathbb{E}_{s' \sim \mathbb{P}(\cdot|s_{h-1},a_{h-1})}[z_h(s',\tau_{h-1})] - z_h(s_\ell,\tau_{h-1}),$$

where in $(i)$ $s_\ell$ is the state that was visited immediately after $\ell$th visit to the state-action pair $(s_{h-1},a_{h-1})$. Note that $|\mathbb{E}_{s' \sim \mathbb{P}(\cdot|s_{h-1},a_{h-1})}[z_h(s',\tau_{h-1})] - z_h(s_\ell,\tau_{h-1})| \leq 2\eta$, thus by invoking Lemma A.1 we have: given any fixed sub-trajectory $\tau_{h-1} \in \Gamma_{h-1}$, for all $t \in \mathbb{N}$ such that $N_t(s_{h-1},a_{h-1}) > 0$

$$\mathbb{E}_{s' \sim \mathbb{P}(\cdot|s_{h-1},a_{h-1})}[z_h(s',\tau_{h-1})] - \mathbb{E}_{s' \sim \widehat{\mathbb{P}}_t(\cdot|s_{h-1},a_{h-1})}[z_h(s',\tau_{h-1})]$$

$$\leq 4\eta\sqrt{\frac{\log\left(\frac{6\log(N_t(s_{h-1},a_{h-1}))}{\delta'}\right)}{N_t(s_{h-1},a_{h-1})}} \qquad (22)$$

with probability at least $1 - \delta'$. In the case where $N_t(s_{h-1},a_{h-1}) = 0$ we have the uniform upper bound

$$\mathbb{E}_{s' \sim \mathbb{P}(\cdot|s_{h-1},a_{h-1})}[z_h(s',\tau_{h-1})] - \mathbb{E}_{s' \sim \widehat{\mathbb{P}}_t(\cdot|s_{h-1},a_{h-1})}[z_h(s',\tau_{h-1})] \leq 2\eta. \qquad (23)$$

Therefore combining inequalities (22) and (23), we can conclude that for any fixed sub-trajectory $\tau_{h-1} \in \Gamma_{h-1}$

$$\forall\, t \in \mathbb{N} : \mathbb{E}_{s' \sim \mathbb{P}(\cdot|s_{h-1},a_{h-1})}[z_h(s',\tau_{h-1})] - \mathbb{E}_{s' \sim \widehat{\mathbb{P}}_t(\cdot|s_{h-1},a_{h-1})}[z_h(s',\tau_{h-1})]$$

$$\leq \min\left\{2\eta, 4\eta\sqrt{\frac{\log\left(\frac{6\log(N_t(s_{h-1},a_{h-1}))}{\delta'}\right)}{N_t(s_{h-1},a_{h-1})}}\right\}$$

with probability at least $1 - \delta'$. By a union bound over all sub-trajectories we find that for all $t \in \mathbb{N}$ and all $\tau_{h-1} \in \Gamma_{h-1}$

$$\mathbb{E}_{s' \sim \mathbb{P}(\cdot|s_{h-1},a_{h-1})}[z_h(s',\tau_{h-1})] - \mathbb{E}_{s' \sim \widehat{\mathbb{P}}_t(\cdot|s_{h-1},a_{h-1})}[z_h(s',\tau_{h-1})]$$

$$\leq \min\left\{2\eta, 4\eta\sqrt{\frac{\log\left(\frac{6\log(N_t(s_{h-1},a_{h-1}))}{\delta'}\right)}{N_t(s_{h-1},a_{h-1})}}\right\}$$

with probability at least $1 - \delta'|\mathcal{S}|^{H-1}|\mathcal{A}|^{H-1}$. Finally a union bound over all $h \in [H]$ lets us conclude that for all $t \in \mathbb{N}$, all $h \in [H]$, and all $\tau_{h-1} \in \Gamma_{h-1}$

$$\mathbb{E}_{s' \sim \mathbb{P}(\cdot|s_{h-1},a_{h-1})}[z_h(s',\tau_{h-1})] - \mathbb{E}_{s' \sim \widehat{\mathbb{P}}_t(\cdot|s_{h-1},a_{h-1})}[z_h(s',\tau_{h-1})]$$

$$\leq \min\left\{2\eta, 4\eta\sqrt{\frac{\log\left(\frac{6\log(N_t(s_{h-1},a_{h-1}))}{\delta'}\right)}{N_t(s_{h-1},a_{h-1})}}\right\}$$

with probability at least $1 - \delta'|\mathcal{S}|^{H-1}|\mathcal{A}|^{H-1}H$. Setting $\delta' = \frac{\delta}{|\mathcal{S}|^{H-1}|\mathcal{A}|^{H-1}H}$ and using equation (21) from above we get that for all $t \in \mathbb{N}$, all $h \in \{2, \ldots, H\}$,

$$\mathbb{E}_{\tau \sim \mathbb{P}^\pi_{(h-1)}}[\check{\mu}_\tau] - \mathbb{E}_{\tau \sim \mathbb{P}^\pi_{(h)}}[\check{\mu}_\tau]$$

$$\leq \mathbb{E}_{s_1 \sim \rho, \ \tau_{h-1} \sim \widehat{\mathbb{P}}^\pi_t(\cdot|s_1)} \left[ \min\left\{ 2\eta, 4\eta \sqrt{\frac{(H-1)\log(|\mathcal{S}||\mathcal{A}|H) + \log\left(\frac{6\log(N_t(s_{h-1}, a_{h-1}))}{\delta}\right)}{N_t(s_{h-1}, a_{h-1})}} \right\} \right]$$

$$\leq \mathbb{E}_{s_1 \sim \rho, \ \tau_{h-1} \sim \widehat{\mathbb{P}}^\pi_t(\cdot|s_1)} \left[ \bar{\xi}^{(t)}_{s_{h-1}, a_{h-1}} \right]$$

with probability at least $1 - \delta$. Summing over all $h \in [H]$ and using equations (18) and (19) we conclude that

$$\mathbb{E}_{s_1 \sim \rho, \ \tau \sim \mathbb{P}^\pi(\cdot|s_1)}[\check{\mu}_\tau] - \mathbb{E}_{s_1 \sim \rho, \ \tau \sim \widehat{\mathbb{P}}^\pi_t(\cdot|s_1)}[\check{\mu}_\tau] \leq \mathbb{E}_{s_1 \sim \rho, \ \tau \sim \widehat{\mathbb{P}}^\pi_t(\cdot|s_1)} \left[ \sum_{h=2}^{H} \bar{\xi}^{(t)}_{s_{h-1}, a_{h-1}} \right]$$

$$= \mathbb{E}_{s_1 \sim \rho, \ \tau \sim \widehat{\mathbb{P}}^\pi_t(\cdot|s_1)} \left[ \sum_{h=1}^{H-1} \bar{\xi}^{(t)}_{s_h, a_h} \right]$$

with the same probability. This establishes our claim. ∎

Next, we shall prove a stronger version of Lemma B.1 that holds uniformly over all policies. Given any bounded scalar function $\check{\mu}$ that maps trajectories to $\mathbb{R}$ and satisfies $|\check{\mu}_\tau| \leq \eta$, any transition dynamics $\bar{\mathbb{P}}$ and any policy $\pi$ define

$$z^{\check{\mu}, \bar{\mathbb{P}}^\pi}_h(s, \tau'_{h-1}) := \mathbb{E}_{a \sim \pi_h(\cdot|s, \tau'_{h-1})} \left[ \mathbb{E}_{\tau \sim \mathbb{P}^\pi}[\check{\mu}_\tau \mid \tau_h = \{s, a, \tau'_{h-1}\}] \right]. \tag{24}$$

This function is different from the $z_h$ that was defined and used locally in the proof of the preceding lemma. The absolute value of the functions $z^{\check{\mu}, \bar{\mathbb{P}}^\pi}_h$ are also bounded by $\eta$.

Suppose that $\Psi(\varepsilon) := \{f_j\}_{j=1}^{\mathcal{N}_{\text{cover}}(\varepsilon)}$ is a set of bounded functions from $\mathcal{S} \mapsto [-\eta, \eta]$, such that for any $h \in [H]$ and for any sub-trajectory $\tau_{h-1} \in \Gamma_{h-1}$, there exists a $f \in \Psi(\varepsilon)$ such that

$$\max_{s \in \mathcal{S}} \left| z^{\check{\mu}, \bar{\mathbb{P}}^\pi}_h(s, \tau_{h-1}) - f(s) \right| \leq \frac{\varepsilon}{2H}. \tag{25}$$

We will construct such a net of functions of size $\mathcal{N}_{\text{cover}}(\varepsilon) \leq \left( \left\lceil \frac{\eta - (-\eta)}{\varepsilon/(2H)} \right\rceil \right)^{|\mathcal{S}|} = \left( \left\lceil \frac{4\eta H}{\varepsilon} \right\rceil \right)^{|\mathcal{S}|}$. Such a set of functions can be built as follows. For each $s \in \mathcal{S}$ we pick an element of the set $\{-\eta, -\eta + \frac{\varepsilon}{2H}, \ldots, \eta\}$. There are at most $\left\lceil \frac{4\eta H}{\varepsilon} \right\rceil$ choices for each state, and therefore there are at most $\left( \left\lceil \frac{4\eta H}{\varepsilon} \right\rceil \right)^{|\mathcal{S}|}$ unique functions that can be defined that map from the state space $\mathcal{S}$ to the set $\{-\eta, -\eta + \frac{\varepsilon}{2H}, \ldots, \eta\}$. Let $\Psi(\varepsilon)$ be these functions. It is easy to check that this set of functions $\Psi(\varepsilon)$ satisfies the condition specified in inequality (25). Also define the function

$$\check{\xi}^{(t)}_{s,a}(\varepsilon; \eta) := \min\left\{ 2\eta, 4\eta \sqrt{\frac{H\log(|\mathcal{S}||\mathcal{A}|H) + |\mathcal{S}|\log\left(\left\lceil \frac{4\eta H}{\varepsilon} \right\rceil\right) + \log\left(\frac{6\log(N_t(s_{h-1}, a_{h-1}))}{\delta}\right)}{N_t(s_{h-1}, a_{h-1})}} \right\}.$$

**Lemma B.2.** *Suppose that $\varepsilon > 0$. Then with probability at least $1 - \delta$, for all $t \in \mathbb{N}$, all policies $\pi \in \Pi$ and all $\check{\mu}_\tau$ such that $|\check{\mu}_\tau| \leq \eta$,*

$$\mathbb{E}_{s_1 \sim \rho, \ \tau \sim \widehat{\mathbb{P}}^\pi_t(\cdot|s_1)}[\check{\mu}_\tau] - \mathbb{E}_{s_1 \sim \rho, \ \tau \sim \mathbb{P}^\pi(\cdot|s_1)}[\check{\mu}_\tau] \leq \mathbb{E}_{s_1 \sim \rho, \ \tau \sim \mathbb{P}^\pi(\cdot|s_1)} \left[ \sum_{h=1}^{H-1} \check{\xi}^{(t)}_{s_h, a_h}(\varepsilon; \eta) \right] + \varepsilon.$$

**Proof** Define $\mathbb{P}^\pi_{(h)}$ to be a trajectory distribution where the initial state is $s_1 \sim \rho$, the state-action pairs up to the end of step $h$ are drawn from $\mathbb{P}^\pi$, and the state-action pairs from step $h + 1$ until the last

step $H$ is drawn from $\widehat{\mathbb{P}}_t^\pi$. Notice that $\mathbb{P}_{(0)}^\pi(s_1,\cdot) = \rho(s_1)\widehat{\mathbb{P}}_t^\pi(\cdot|s_1)$ and $\mathbb{P}_{(H)}^\pi(s_1) = \rho(s_1)\mathbb{P}^\pi(\cdot|s_1)$.

$$\mathbb{E}_{s_1\sim\rho,\ \tau\sim\widehat{\mathbb{P}}_t^\pi(\cdot|s_1)}[\check\mu_\tau] - \mathbb{E}_{s_1\sim\rho,\ \tau\sim\mathbb{P}^\pi(\cdot|s_1)}[\check\mu_\tau] = \sum_{h=1}^{H}\mathbb{E}_{\tau\sim\mathbb{P}_{(h-1)}^\pi}[\check\mu_\tau] - \mathbb{E}_{\tau\sim\mathbb{P}_{(h)}^\pi}[\check\mu_\tau]. \tag{26}$$

Consider the term where $h = 1$. The trajectory distributions $\mathbb{P}_{(0)}^\pi$ and $\mathbb{P}_{(1)}^\pi$ differ only their distributions of state-action pairs in step 1, thus,

$$\mathbb{E}_{\tau\sim\mathbb{P}_{(0)}^\pi}[\check\mu_\tau] - \mathbb{E}_{\tau\sim\mathbb{P}_{(1)}^\pi}[\check\mu_\tau]$$
$$= \mathbb{E}_{s_1\sim\rho}\left[\mathbb{E}_{a_1\sim\pi(\cdot|s_1)}\mathbb{E}_{\tau\sim\mathbb{P}_{(0)}^\pi}[\check\mu_\tau|(s_1,a_1)]\right] - \mathbb{E}_{s_1\sim\rho}\left[\mathbb{E}_{a_1\sim\pi(\cdot|s_1)}\mathbb{E}_{\tau\sim\mathbb{P}_{(0)}^\pi}[\check\mu_\tau|(s_1,a_1)]\right] = 0. \tag{27}$$

Consider any other term in this sum. Again the trajectory distributions $\mathbb{P}_{(h-1)}^\pi$ and $\mathbb{P}_{(h)}^\pi$ differ only their distributions of state-action pairs in step $h$ and hence

$$\mathbb{E}_{\tau\sim\mathbb{P}_{(h-1)}^\pi}[\check\mu_\tau] - \mathbb{E}_{\tau\sim\mathbb{P}_{(h)}^\pi}[\check\mu_\tau]$$
$$= \mathbb{E}_{s_1\sim\rho,\ \tau_{h-1}\sim\mathbb{P}^\pi(\cdot|s_1)}\left(\mathbb{E}_{\tau\sim\mathbb{P}_{(h-1)}^\pi}[\check\mu_\tau|\tau_{h-1}] - \mathbb{E}_{\tau\sim\mathbb{P}_{(h)}^\pi}[\check\mu_\tau|\tau_{h-1}]\right)$$
$$= \mathbb{E}_{s_1\sim\rho,\ \tau_{h-1}\sim\mathbb{P}^\pi(\cdot|s_1)}\Bigg[\mathbb{E}_{s_h\sim\widehat{\mathbb{P}}_t(\cdot|s_{h-1},a_{h-1})}\left[\mathbb{E}_{a_h\sim\pi_h(\cdot|s_h,\tau_{h-1})}\mathbb{E}_{\tau\sim\widehat{\mathbb{P}}_t^\pi}[\check\mu_\tau|(s_h,a_h,\tau_{h-1})]\right]$$
$$- \mathbb{E}_{s_h\sim\mathbb{P}(\cdot|s_{h-1},a_{h-1})}\left[\mathbb{E}_{a_h\sim\pi_h(\cdot|s_h,\tau_{h-1})}\mathbb{E}_{\tau\sim\widehat{\mathbb{P}}_t^\pi}[\check\mu_\tau|(s_h,a_h,\tau_{h-1})]\right]\Bigg]$$
$$= \mathbb{E}_{s_1\sim\rho,\ \tau_{h-1}\sim\mathbb{P}^\pi(\cdot|s_1)}\Bigg[\underbrace{\mathbb{E}_{s'\sim\widehat{\mathbb{P}}_t(\cdot|s_{h-1},a_{h-1})}[z_h^{\check\mu,\widehat{\mathbb{P}}_t^\pi}(s',\tau_{h-1})] - \mathbb{E}_{s'\sim\mathbb{P}(\cdot|s_{h-1},a_{h-1})}[z_h^{\check\mu,\widehat{\mathbb{P}}_t^\pi}(s',\tau_{h-1})]}\Bigg] \tag{28}$$

where $z_h^{\check\mu,\widehat{\mathbb{P}}_t^\pi}$ is defined in equation (24) above. We shall now upper bound the term in the under-brace above with high probability uniformly over all sub-trajectories $\tau_{h-1}$.

Recall that $N_t(s,a)$ is the number of times the state action pair $(s,a)$ has been visited before episode $t$, and $N_t(s';s,a)$ is the number of times the state $s'$ is visited starting from state-action pair $(s,a)$ before episode $t$. When $N_t(s,a) > 0$ by its definition $\widehat{\mathbb{P}}_t(s'|s,a) = \frac{N_t(s';s,a)}{N_t(s,a)}$. Thus for any fixed sub-trajectory $\tau_{h-1} \in \Gamma_{h-1}$ and episode $t \in \mathbb{N}$ where $N_t(s_{h-1},a_{h-1}) > 0$ we have

$$\mathbb{E}_{s'\sim\widehat{\mathbb{P}}_t(\cdot|s_{h-1},a_{h-1})}[z_h^{\check\mu,\widehat{\mathbb{P}}_t}(s',\tau_{h-1})] - \mathbb{E}_{s'\sim\mathbb{P}(\cdot|s_{h-1},a_{h-1})}[z_h^{\check\mu,\widehat{\mathbb{P}}_t}(s',\tau_{h-1})]$$
$$= \sum_{s'\in\mathcal{S}}\frac{N_t(s';s_{h-1},a_{h-1})}{N_t(s_{h-1},a_{h-1})}z_h^{\check\mu,\widehat{\mathbb{P}}_t}(s',\tau_{h-1}) - \mathbb{E}_{s'\sim\mathbb{P}(\cdot|s_{h-1},a_{h-1})}[z_h^{\check\mu,\widehat{\mathbb{P}}_t}(s',\tau_{h-1})]$$
$$= \frac{1}{N_t(s_{h-1},a_{h-1})}\Bigg[\sum_{s'\in\mathcal{S}}N_t(s';s_{h-1},a_{h-1})z_h^{\check\mu,\widehat{\mathbb{P}}_t}(s',\tau_{h-1})$$
$$- N_t(s_{h-1},a_{h-1})\mathbb{E}_{s'\sim\mathbb{P}(\cdot|s_{h-1},a_{h-1})}[z_h^{\check\mu,\widehat{\mathbb{P}}_t}(s',\tau_{h-1})]\Bigg]$$
$$\overset{(i)}{=} \frac{1}{N_t(s_{h-1},a_{h-1})}\Bigg[\sum_{\ell=1}^{N_t(s_{h-1},a_{h-1})}z_h^{\check\mu,\widehat{\mathbb{P}}_t}(s_\ell,\tau_{h-1}) - N_t(s_{h-1},a_{h-1})\mathbb{E}_{s'\sim\mathbb{P}(\cdot|s_{h-1},a_{h-1})}[z_h^{\check\mu,\widehat{\mathbb{P}}_t}(s',\tau_{h-1})]\Bigg]$$
$$= \frac{1}{N_t(s_{h-1},a_{h-1})}\sum_{\ell=1}^{N_t(s_{h-1},a_{h-1})}z_h^{\check\mu,\widehat{\mathbb{P}}_t}(s_\ell,\tau_{h-1}) - \mathbb{E}_{s'\sim\mathbb{P}(\cdot|s_{h-1},a_{h-1})}[z_h^{\check\mu,\widehat{\mathbb{P}}_t}(s',\tau_{h-1})], \tag{29}$$

where in $(i)$ $s_\ell$ is the state that was visited immediately after the $\ell$th visit to the state-action pair $(s_{h-1},a_{h-1})$. Let $\widehat{f} \in \Psi(\varepsilon)$ be a function such that

$$\max_{s\in\mathcal{S}}\left|z_h^{\check\mu,\widehat{\mathbb{P}}_t}(s,\tau_{h-1}) - \widehat{f}(s)\right| \le \frac{\varepsilon}{2H}.$$

Such a function exists by the definition of the set $\Psi(\varepsilon)$. Therefore,

$$\max_{s \in \mathcal{S}} \left| z_h^{\check{\mu}, \widehat{\mathbb{P}}_t^\pi}(s, \tau_{h-1}) - \mathbb{E}_{s' \sim \mathbb{P}(\cdot|s_{h-1}, a_{h-1})} \left[ z_h^{\check{\mu}, \widehat{\mathbb{P}}_t^\pi}(s', \tau_{h-1}) \right] - \widehat{f}(s) + \mathbb{E}_{s' \sim \mathbb{P}(\cdot|s_{h-1}, a_{h-1})} \left[ \widehat{f}(s') \right] \right| \leq \frac{\varepsilon}{H}.$$

Continuing from equation (29) we have

$$\mathbb{E}_{s' \sim \widehat{\mathbb{P}}_t(\cdot|s_{h-1}, a_{h-1})} [z_h^{\check{\mu}, \widehat{\mathbb{P}}_t^\pi}(s', \tau_{h-1})] - \mathbb{E}_{s' \sim \mathbb{P}(\cdot|s_{h-1}, a_{h-1})} [z_h^{\check{\mu}, \widehat{\mathbb{P}}_t}(s', \tau_{h-1})]$$

$$\leq \frac{1}{N_t(s_{h-1}, a_{h-1})} \sum_{\ell=1}^{N_t(s_{h-1}, a_{h-1})} \left( \widehat{f}(s_\ell) - \mathbb{E}_{s' \sim \mathbb{P}(\cdot|s_{h-1}, a_{h-1})} \left[ \widehat{f}(s_\ell) \right] \right) + \frac{\varepsilon}{H}. \tag{30}$$

Observe that for all $\ell$, $\left| \widehat{f}(s_\ell) - \mathbb{E}_{s' \sim \mathbb{P}(\cdot|s_{h-1}, a_{h-1})} [\widehat{f}(s')] \right| \leq 2\eta$. Thus by invoking Lemma A.1 and by a union bound over the elements of $\Psi(\varepsilon)$, we have that, given any fixed sub-trajectory $\tau_{h-1} \in \Gamma_{h-1}$, for all $f \in \Psi(\varepsilon)$ and all $t \in \mathbb{N}$ such that $N_t(s_{h-1}, a_{h-1}) > 0$:

$$\frac{1}{N_t(s_{h-1}, a_{h-1})} \sum_{\ell=1}^{N_t(s_{h-1}, a_{h-1})} f(s_\ell) - \mathbb{E}_{s' \sim \mathbb{P}(\cdot|s_{h-1}, a_{h-1})} [f(s_\ell)] \leq 4\eta \sqrt{\frac{\log\left( \frac{6 \log(N_t(s_{h-1}, a_{h-1}))}{\delta'} \right)}{N_t(s_{h-1}, a_{h-1})}}$$

with probability at least $1 - |\mathcal{N}_{\mathsf{cover}}(\varepsilon)|\delta'$. Combined with inequality (30) we have that given any fixed sub-trajectory $\tau_{h-1} \in \Gamma_{h-1}$, for all policies $\pi \in \Pi$, for all $\check{\mu}$ bounded by $\eta$ and all $t \in \mathbb{N}$ such that $N_t(s_{h-1}, a_{h-1}) > 0$:

$$\mathbb{E}_{s' \sim \widehat{\mathbb{P}}_t(\cdot|s_{h-1}, a_{h-1})} [z_h^{\check{\mu}, \widehat{\mathbb{P}}_t^\pi}(s', \tau_{h-1})] - \mathbb{E}_{s' \sim \mathbb{P}(\cdot|s_{h-1}, a_{h-1})} [z_h^{\check{\mu}, \widehat{\mathbb{P}}_t}(s', \tau_{h-1})]$$

$$\leq 4\eta \sqrt{\frac{\log\left( \frac{6 \log(N_t(s_{h-1}, a_{h-1}))}{\delta'} \right)}{N_t(s_{h-1}, a_{h-1})}} + \frac{\varepsilon}{H} \tag{31}$$

with probability at least $1 - |\mathcal{N}_{\mathsf{cover}}(\varepsilon)|\delta'$. We also have a simple upper bound,

$$\mathbb{E}_{s' \sim \widehat{\mathbb{P}}_t(\cdot|s_{h-1}, a_{h-1})} [z_h^{\check{\mu}, \widehat{\mathbb{P}}_t^\pi}(s', \tau_{h-1})] - \mathbb{E}_{s' \sim \mathbb{P}(\cdot|s_{h-1}, a_{h-1})} [z_h^{\check{\mu}, \widehat{\mathbb{P}}_t}(s', \tau_{h-1})] \leq 2\eta. \tag{32}$$

Combining inequalities (31) and (32) we get that for any fixed sub-trajectory $\tau_{h-1} \in \Gamma_{h-1}$, for all $t \in \mathbb{N}$, for all $\pi \in \Pi$, for all $\check{\mu}$ bounded by $\eta$,

$$\mathbb{E}_{s' \sim \widehat{\mathbb{P}}_t(\cdot|s_{h-1}, a_{h-1})} [z_h^{\check{\mu}, \widehat{\mathbb{P}}_t^\pi}(s', \tau_{h-1})] - \mathbb{E}_{s' \sim \mathbb{P}(\cdot|s_{h-1}, a_{h-1})} [z_h^{\check{\mu}, \widehat{\mathbb{P}}_t}(s', \tau_{h-1})]$$

$$\leq \min \left\{ 2\eta, 4\eta \sqrt{\frac{\log\left( \frac{6 \log(N_t(s_{h-1}, a_{h-1}))}{\delta'} \right)}{N_t(s_{h-1}, a_{h-1})}} + \frac{\varepsilon}{H} \right\}.$$

By a union bound over all sub-trajectories we find that for all $t \in \mathbb{N}$, all policies $\pi \in \Pi$, all $\check{\mu}$ bounded by $\eta$ and all $\tau_{h-1} \in \Gamma_{h-1}$

$$\mathbb{E}_{s' \sim \widehat{\mathbb{P}}_t(\cdot|s_{h-1}, a_{h-1})} [z_h^{\check{\mu}, \widehat{\mathbb{P}}_t^\pi}(s', \tau_{h-1})] - \mathbb{E}_{s' \sim \mathbb{P}(\cdot|s_{h-1}, a_{h-1})} [z_h^{\check{\mu}, \widehat{\mathbb{P}}_t}(s', \tau_{h-1})]$$

$$\leq \min \left\{ 2\eta, 4\eta \sqrt{\frac{\log\left( \frac{6 \log(N_t(s_{h-1}, a_{h-1}))}{\delta'} \right)}{N_t(s_{h-1}, a_{h-1})}} + \frac{\varepsilon}{H} \right\}$$

with probability at least $1 - (|\mathcal{S}||\mathcal{A}|)^{H-1} |\mathcal{N}_{\mathsf{cover}}(\varepsilon)|\delta'$. Finally a union bound over the steps of the episode $h \in [H]$ lets us conclude that for all $t \in \mathbb{N}$, all policies $\pi \in \Pi$, all $\check{\mu}$ bounded by $\eta$, all $h \in [H]$ and all $\tau_{h-1} \in \Gamma_{h-1}$

$$\mathbb{E}_{s' \sim \widehat{\mathbb{P}}_t(\cdot|s_{h-1}, a_{h-1})} [z_h^{\check{\mu}, \widehat{\mathbb{P}}_t^\pi}(s', \tau_{h-1})] - \mathbb{E}_{s' \sim \mathbb{P}(\cdot|s_{h-1}, a_{h-1})} [z_h^{\check{\mu}, \widehat{\mathbb{P}}_t}(s', \tau_{h-1})]$$

$$\leq \min \left\{ 2\eta, 4\eta \sqrt{\frac{\log\left( \frac{6 \log(N_t(s_{h-1}, a_{h-1}))}{\delta'} \right)}{N_t(s_{h-1}, a_{h-1})}} + \frac{\varepsilon}{H} \right\}$$

$$\leq \min \left\{ 2\eta, 4\eta \sqrt{\frac{\log\left( \frac{6 \log(N_t(s_{h-1}, a_{h-1}))}{\delta'} \right)}{N_t(s_{h-1}, a_{h-1})}} \right\} + \frac{\varepsilon}{H}$$

with probability at least $1 - H(|\mathcal{S}||\mathcal{A}|)^{H-1}|\mathcal{N}_{\text{cover}}(\varepsilon)|\delta'$. Setting

$$\delta' = \frac{\delta}{|\mathcal{S}|^{H-1}|\mathcal{A}|^{H-1}H\left(\left\lceil\frac{4\eta H}{\varepsilon}\right\rceil\right)^{|\mathcal{S}|}} \leq \frac{\delta}{|\mathcal{S}|^{H-1}|\mathcal{A}|^{H-1}H|\mathcal{N}_{\text{cover}}(\varepsilon)|}$$

and using equation (28) from above we get that for all $t \in \mathbb{N}$, all $h \in \{2, \ldots, H\}$, all $\pi \in \Pi$, and all $\check{\mu}$ bounded by $\eta$ we have

$$\mathbb{E}_{\tau \sim \mathbb{P}^\pi_{(h-1)}}[\check{\mu}_\tau] - \mathbb{E}_{\tau \sim \mathbb{P}^\pi_{(h)}}[\check{\mu}_\tau]$$

$$\leq \mathbb{E}_{s_1 \sim \rho,\, \tau_{h-1} \sim \mathbb{P}^\pi(\cdot|s_1)}\left[\min\left\{2\eta, 4\eta\sqrt{\frac{\log\left(\frac{6(|\mathcal{S}||\mathcal{A}|H)^{H-1}\left\lceil\frac{4\eta H}{\varepsilon}\right\rceil^{|\mathcal{S}|}\log(N_t(s_{h-1},a_{h-1}))}{\delta}\right)}{N_t(s_{h-1}, a_{h-1})}}\right\}\right] + \frac{\varepsilon}{H}$$

$$\leq \mathbb{E}_{s_1 \sim \rho,\, \tau_{h-1} \sim \mathbb{P}^\pi(\cdot|s_1)}\left[\check{\xi}^{(t)}_{s_{h-1},a_{h-1}}(\varepsilon; \eta)\right] + \frac{\varepsilon}{H}$$

with probability at least $1 - \delta$. Summing over all $h \in [H]$ and using equations (26) and (27) we conclude that for $t \in \mathbb{N}$, all $\pi \in \Pi$ and all $\check{\mu}$ bounded by $\eta$,

$$\mathbb{E}_{s_1 \sim \rho,\, \tau \sim \widehat{\mathbb{P}}^\pi_t(\cdot|s_1)}[\check{\mu}_\tau] - \mathbb{E}_{\tau \sim \mathbb{P}^\pi(\cdot|s_1)}[\check{\mu}_\tau] \leq \mathbb{E}_{s_1 \sim \rho,\, \tau \sim \mathbb{P}^\pi(\cdot|s_1)}\left[\sum_{h=2}^{H}\check{\xi}^{(t)}_{s_{h-1},a_{h-1}}(\varepsilon; \eta)\right] + H \times \frac{\varepsilon}{H}$$

$$= \mathbb{E}_{s_1 \sim \rho,\, \tau \sim \mathbb{P}^\pi(\cdot|s_1)}\left[\sum_{h=1}^{H-1}\check{\xi}^{(t)}_{s_h,a_h}(\varepsilon; \eta)\right] + \varepsilon$$

again with probability at least $1 - \delta$. This completes the proof of this lemma. ∎

## B.2 Proof of Lemma 3.1

Recall the statement of the lemma from above.

**Lemma 3.1.** *For any $\delta \in (0, 1]$, define the event*

$$\mathcal{E}_\delta := \left\{\text{for all } t \in [N], \tau \in \Gamma : \left|\mu(\mathbf{w}_\star^\top \phi(\tau)) - \mu(\widehat{\mathbf{w}}_t^\top \phi(\tau))\right| \leq \sqrt{\kappa}\beta_t(\delta)\|\phi(\tau)\|_{\mathbf{\Sigma}_t^{-1}}\right\}. \quad (5)$$

*Then $\mathbb{P}(\mathcal{E}_\delta) \geq 1 - \delta$.*

**Proof** We invoke [31, Proposition 7] by noting that in our paper: $c_\mu = 1/\kappa$, $\kappa_\mu = 1$ (Lipschitz constant of $\mu$), $m = 1$ (scale of the rewards), $\lambda = 1$ (the $\ell_2$ regularization parameter), $\tau = N$ (length of the sliding window) and $\mathcal{T}(\tau) = [N]$ (in their paper $\mathcal{T}(\tau)$ corresponds to the set of episodes where the underlying parameter $\mathbf{w}_\star$ remains unchanged. In our setting $\mathbf{w}_\star$ is constant for all episodes). ∎

## B.3 Definition and Properties of a "Good Event" $\mathcal{E}_{\text{good}}$

The proof of Theorem 3.2 proceeds by showing that a favorable event $\mathcal{E}_{\text{good}}$ that occurs with high probability. We shall then upper bound the regret of Algorithm 1 when this event occurs. Before defining this event we need some additional notation.

**Definition B.3.** *For all $t \in [N]$, given any policy $\pi$ define*

$$\bar{V}_t^\pi := \mathbb{E}_{s_1 \sim \rho,\, \tau \sim \mathbb{P}^\pi(\cdot|s_1)}\left[\bar{\mu}_t(\widehat{\mathbf{w}}_t, \tau)\right],$$

*where recall from equation (8a) that $\bar{\mu}_t(\widehat{\mathbf{w}}_t, \tau) = \min\left\{\mu\left(\widehat{\mathbf{w}}_t^\top \phi(\tau)\right) + \sqrt{\kappa}\beta_t(\delta)\|\phi(\tau)\|_{\mathbf{\Sigma}_t^{-1}}, 1\right\}$. Further, for all episodes $t \in [N]$ also define $\bar{V}^{(t)} := \bar{V}_t^{\pi^{(t)}}$ and $\bar{V}_\star^{(t)} := \bar{V}_t^{\pi_\star}$.*

Also define the value function when the average rewards are $\widetilde{\mu}_t(\widehat{\mathbf{w}}_t, \tau)$ and the transition dynamics are governed by $\widehat{\mathbb{P}}_t$.

**Definition B.4.** *For any episode $t \in [N]$, given any policy $\pi \in \Pi$ define*

$$\widetilde{V}_t^\pi := \mathbb{E}_{s_1 \sim \rho, \, \tau \sim \widehat{\mathbb{P}}_t^\pi(\cdot|s_1)} \left[ \widetilde{\mu}_t(\widehat{\mathbf{w}}_t, \tau) \right] \tag{33}$$

*where $\widetilde{\mu}_t$ is defined above in equation (8b). To simplify notation we additionally define $\widetilde{V}^{(t)} := \widetilde{V}_t^{\pi^{(t)}}$ and $\widetilde{V}_\star^{(t)} := \widetilde{V}_t^{\pi_\star}$.*

Consider the following events:

$$\mathcal{E}_1 := \left\{ \sum_{t=1}^N V_\star \leq \sum_{t=1}^N \widetilde{V}_\star^{(t)} \right\}; \tag{34a}$$

$$\mathcal{E}_2 := \left\{ \sum_{t=1}^N \bar{V}^{(t)} - V^{(t)} \leq \beta_N(\delta) \sqrt{8Nd \max\{\kappa, 1\} \log\left(1 + \frac{N}{\kappa d}\right)} \right.$$

$$\left. + 4\sqrt{N \log\left(\frac{6 \log(N)}{\delta}\right)} \right\}; \tag{34b}$$

$$\mathcal{E}_3 := \left\{ \sum_{t=1}^N \widetilde{V}^{(t)} - \bar{V}^{(t)} \leq (2H+1) \sum_{t=1}^N \sum_{h=1}^{H-1} \xi^{(t)}_{s_h^{(t)}, a_h^{(t)}} + 4H^2 \sqrt{N \log\left(\frac{6 \log(N)}{\delta}\right)} + 1 \right\}, \tag{34c}$$

where $(s_h^{(t)}, a_h^{(t)})$ is the state-action pair visited at step $h$ during episode $t$.

**Lemma B.5.** *Define the event $\mathcal{E}_{\text{good}} := \mathcal{E}_1 \cap \mathcal{E}_2 \cap \mathcal{E}_3$. Then $\mathbb{P}\left[\mathcal{E}_{\text{good}}\right] \geq 1 - 6N\delta$.*

The good event occurs when the value function $\widetilde{V}_\star^{(t)}$ is optimistic, that is, it over estimates the true value function of the optimal policy $V_\star$ and when the sums of $\bar{V}^{(t)} - V^{(t)}$ and $\widetilde{V}^{(t)} - \bar{V}^{(t)}$ over the episodes can be bounded.

**Proof** We will show that each of the three events $\mathcal{E}_1$, $\mathcal{E}_2$ and $\mathcal{E}_3$ occurs with a high probability and take union bound to prove our claim.

**Event $\mathcal{E}_1$:** By invoking Lemma B.1 $N$ times, once per episode, with the choice $\eta = 1$ we get

$$\sum_{t=1}^N V_\star = \sum_{t=1}^N \mathbb{E}_{s_1 \sim \rho, \, \tau \sim \mathbb{P}^{\pi_\star}(\cdot|s_1)} \left[ \mu(\mathbf{w}_\star^\top \phi(\tau)) \right]$$

$$\leq \sum_{t=1}^N \mathbb{E}_{s_1 \sim \rho, \, \tau \sim \widehat{\mathbb{P}}_t^{\pi_\star}(\cdot|s_1)} \left[ \mu(\mathbf{w}_\star^\top \phi(\tau)) + \sum_{h=1}^{H-1} \bar{\xi}^{(t)}_{s_h, a_h}(1) \right]$$

$$\leq \sum_{t=1}^N \mathbb{E}_{s_1 \sim \rho, \, \tau \sim \widehat{\mathbb{P}}_t^{\pi_\star}(\cdot|s_1)} \left[ \mu(\mathbf{w}_\star^\top \phi(\tau)) + \sum_{h=1}^{H-1} \xi^{(t)}_{s_h, a_h} \right] \tag{35}$$

(by the definition of $\xi^{(t)}_{s_h, a_h}$ in equation (7))

with probability at least $1 - N\delta$. Recall the definition of the event $\mathcal{E}_\delta$ from equation (5) and observe that it occurs with probability at least $1 - \delta$ by Lemma 3.1. Under event $\mathcal{E}_\delta$ for any $t \in [N]$ and any $\tau \in \Gamma$

$$\mu(\mathbf{w}_\star^\top \phi(\tau)) = \min\left\{ \mu(\mathbf{w}_\star^\top \phi(\tau)), 1 \right\} \leq \min\left\{ \mu(\widehat{\mathbf{w}}_t^\top \phi(\tau)) + \sqrt{\kappa} \beta_t(\delta) \|\phi(\tau)\|_{\mathbf{\Sigma}_t^{-1}}, 1 \right\}$$

$$= \bar{\mu}_t(\widehat{\mathbf{w}}_t, \tau).$$

Therefore by a union bound over $\mathcal{E}_\delta$ and the event where inequality (35) holds we infer that

$$\sum_{t=1}^N V_\star \leq \sum_{t=1}^N \mathbb{E}_{s_1 \sim \rho, \, \tau \sim \widehat{\mathbb{P}}_t^{\pi_\star}(\cdot|s_1)} \left[ \bar{\mu}_t(\widehat{\mathbf{w}}_t, \tau) + \sum_{h=1}^{H-1} \xi^{(t)}_{s_h, a_h} \right] = \sum_{t=1}^N \widetilde{V}_\star^{(t)},$$

with probability at least $1 - (N+1)\delta$.

**Event $\mathcal{E}_2$:** Assume that the event $\mathcal{E}_\delta$ occurs. Lemma 3.1 guarantees that this happens with probability at least $1 - \delta$. Consider the following martingale difference sequence

$$D_t := \bar{V}^{(t)} - V^{(t)} - \left[\bar{\mu}_t\left(\widehat{\mathbf{w}}_t, \tau^{(t)}\right) - \mu\left(\mathbf{w}_\star^\top \phi(\tau^{(t)})\right)\right].$$

Note that $|D_t| \leq 2$ since both $\bar{\mu}_t$ and $\mu$ take values between 0 and 1. Therefore, by applying Lemma A.1 we have that

$$\sum_{t=1}^N \bar{V}^{(t)} - V^{(t)} \leq \sum_{t=1}^N \bar{\mu}_t\left(\widehat{\mathbf{w}}_t, \tau^{(t)}\right) - \mu\left(\mathbf{w}_\star^\top \phi(\tau^{(t)})\right) + 4\sqrt{N \log\left(\frac{6\log(N)}{\delta}\right)} \qquad (36)$$

with probability at least $1 - \delta$. Let us now upper bound the sum in the RHS above

$$\sum_{t=1}^N \bar{\mu}_t\left(\widehat{\mathbf{w}}_t, \tau^{(t)}\right) - \mu\left(\mathbf{w}_\star^\top \phi(\tau^{(t)})\right)$$

$$\overset{(i)}{=} \sum_{t=1}^N \min\left\{\mu\left(\widehat{\mathbf{w}}_t^\top \phi(\tau^{(t)})\right) + \sqrt{\kappa}\beta_t(\delta)\|\phi(\tau^{(t)})\|_{\boldsymbol{\Sigma}_t^{-1}}, 1\right\} - \min\left\{\mu\left(\mathbf{w}_\star^\top \phi(\tau^{(t)})\right), 1\right\}$$

$$\overset{(ii)}{\leq} \sum_{t=1}^N \left|\mu\left(\widehat{\mathbf{w}}_t^\top \phi(\tau^{(t)})\right) + \sqrt{\kappa}\beta_t(\delta)\|\phi(\tau^{(t)})\|_{\boldsymbol{\Sigma}_t^{-1}} - \mu\left(\mathbf{w}_\star^\top \phi(\tau^{(t)})\right)\right|$$

$$\overset{(iii)}{\leq} 2\sqrt{\kappa}\sum_{t=1}^N \beta_t(\delta)\|\phi(\tau^{(t)})\|_{\boldsymbol{\Sigma}_t^{-1}}$$

$$\overset{(iv)}{\leq} 2\sqrt{\kappa}\beta_N(\delta)\sum_{t=1}^N \|\phi(\tau^{(t)})\|_{\boldsymbol{\Sigma}_t^{-1}},$$

where $(i)$ follows by the definition of $\bar{\mu}_t$ and since $\mu$ is bounded between 0 and 1, $(ii)$ follows since for the function $z \mapsto \min\{z, 1\}$ is 1-Lipschitz, $(iii)$ follows since we have assumed that the event $\mathcal{E}_\delta$ occurs which provides the bound $|\mu\left(\widehat{\mathbf{w}}_t^\top \phi(\tau^{(t)})\right) - \mu\left(\mathbf{w}_\star^\top \phi(\tau^{(t)})\right)| \leq \sqrt{\kappa}\beta_t(\delta)\|\phi(\tau^{(t)})\|_{\boldsymbol{\Sigma}_t^{-1}}$, and $(iv)$ follows since $\beta_t(\delta)$ is an increasing function of $t$.

Continuing, since for any vector $\mathbf{z} \in \mathbb{R}^N$ $\|\mathbf{z}\|_1 \leq \sqrt{N}\|\mathbf{z}\|_2$, thus

$$\sum_{t=1}^N \bar{\mu}_t\left(\widehat{\mathbf{w}}_t, \tau^{(t)}\right) - \mu\left(\mathbf{w}_\star^\top \phi(\tau^{(t)})\right) \leq 2\sqrt{\kappa}\beta_N(\delta)\sqrt{N}\sqrt{\sum_{t=1}^N \|\phi(\tau^{(t)})\|_{\boldsymbol{\Sigma}_t^{-1}}^2}$$

$$\leq \beta_N(\delta)\sqrt{8Nd\max\{\kappa, 1\}\log\left(1 + \frac{N}{\kappa d}\right)}$$

where the final inequality follows by invoking the determinant lemma (Lemma A.3) from above. A union bound over the event $\mathcal{E}_\delta$ and the event where inequality (36) holds proves that this bound holds with probability at least $1 - 2\delta$.

**Event $\mathcal{E}_3$:** We wish to establish a bound on $\sum_{t=1}^N \widetilde{V}^{(t)} - \bar{V}^{(t)}$. By definition

$$\sum_{t=1}^N \widetilde{V}^{(t)} = \sum_{t=1}^N \mathbb{E}_{s_1 \sim \rho, \; \tau \sim \widehat{\mathbb{P}}_\tau^{\pi^{(t)}}(\cdot|s_1)}\left[\bar{\mu}_t(\widehat{\mathbf{w}}_t, \tau) + \sum_{h=1}^{H-1} \xi_{s_h, a_h}^{(t)}\right].$$

For each $t \in [N]$ define the trajectory score function $\check{\mu}_\tau^{(t)} = \bar{\mu}_t(\widehat{\mathbf{w}}_t, \tau) + \sum_{h=1}^{H-1} \xi_{s_h, a_h}^{(t)}$. Notice that since $|\xi_{s,a}^{(t)}| \leq 2$ we have that $|\check{\mu}_\tau^{(t)}| < 2H$. Thus, by invoking Lemma B.2 $N$ times, once per episode, with the choices $\eta = 2H$ and $\varepsilon = \frac{1}{N}$ we infer that

$$\sum_{t=1}^N \widetilde{V}^{(t)} \leq \sum_{t=1}^N \left(\mathbb{E}_{s_1 \sim \rho, \; \tau \sim \mathbb{P}^{\pi^{(t)}}(\cdot|s_1)}\left[\bar{\mu}_t(\widehat{\mathbf{w}}_t, \tau) + \sum_{h=1}^{H-1} \xi_{s_h, a_h}^{(t)} + 2H\sum_{h=1}^{H-1} \check{\xi}_{s_h, a_h}^{(t)}\left(\frac{1}{N}, 2H\right)\right] + \frac{1}{N}\right)$$

$$= \sum_{t=1}^N \mathbb{E}_{s_1 \sim \rho, \; \tau \sim \mathbb{P}^{\pi^{(t)}}(\cdot|s_1)}\left[\bar{\mu}_t(\widehat{\mathbf{w}}_t, \tau) + (2H+1)\sum_{h=1}^{H-1} \xi_{s_h, a_h}^{(t)}\right] + 1 \qquad (37)$$

with probability $1 - N\delta$. Assume that the event where inequality (37) holds occurs going forward. Under this event the difference

$$\sum_{t=1}^{N} \widetilde{V}^{(t)} - \bar{V}^{(t)} = \sum_{t=1}^{N} \widetilde{V}^{(t)} - \sum_{t=1}^{N} \mathbb{E}_{s_1 \sim \rho, \, \tau \sim \mathbb{P}^{\pi^{(t)}}(\cdot|s_1)} \left[ \bar{\mu}_t(\widehat{\mathbf{w}}_t, \tau) \right]$$

$$\leq (2H+1) \sum_{t=1}^{N} \mathbb{E}_{s_1 \sim \rho, \, \tau \sim \mathbb{P}^{\pi^{(t)}}(\cdot|s_1)} \left[ \sum_{h=1}^{H-1} \xi_{s_h, a_h}^{(t)} \right] + 1. \qquad (38)$$

Finally, define the martingale-difference sequence

$$D_t := (2H+1) \sum_{t=1}^{N} \mathbb{E}_{s_1 \sim \rho, \, \tau \sim \mathbb{P}^{\pi^{(t)}}(\cdot|s_1)} \left[ \sum_{h=1}^{H-1} \xi_{s_h, a_h}^{(t)} \right] - (2H+1) \sum_{h=1}^{H-1} \xi_{s_h^{(t)}, a_h^{(t)}}^{(t)}.$$

Notice that $|D_t| \leq (2H+1)(H-1) \leq 2H^2$. Applying Lemma A.1 with $\zeta = 2H^2$ we find that

$$(2H+1) \sum_{t=1}^{N} \mathbb{E}_{s_1 \sim \rho, \, \tau \sim \mathbb{P}^{\pi^{(t)}}(\cdot|s_1)} \left[ \sum_{h=1}^{H-1} \xi_{s_h, a_h}^{(t)} \right]$$

$$\leq (2H+1) \sum_{t=1}^{N} \sum_{h=1}^{H-1} \xi_{s_h^{(t)}, a_h^{(t)}}^{(t)} + 4H^2 \sqrt{N \log \left( \frac{6 \log(N)}{\delta} \right)}$$

with probability at least $1 - \delta$. Combining this with inequality (38) we conclude that

$$\sum_{t=1}^{N} \widetilde{V}^{(t)} - \bar{V}^{(t)} \leq (2H+1) \sum_{t=1}^{N} \sum_{h=1}^{H-1} \xi_{s_h^{(t)}, a_h^{(t)}}^{(t)} + 4H^2 \sqrt{N \log \left( \frac{6 \log(N)}{\delta} \right)} + 1$$

with probability at least $1 - (N+1)\delta$. This proves that $\mathbb{P}[\mathcal{E}_3] \geq 1 - (N+1)\delta$.

**Union bound over the three events:** A union bound over the three events shows that $\mathbb{P}[\mathcal{E}_{\text{good}}] \geq 1 - \mathbb{P}[\mathcal{E}_1^c] - \mathbb{P}[\mathcal{E}_2^c] - \mathbb{P}[\mathcal{E}_3^c] \geq 1 - (2N+4)\delta \geq 1 - 6N\delta$, which completes the proof. ∎

## B.4 Proof of Theorem 3.2

Recall the statement of the theorem.

**Theorem 3.2.** *For any $\bar{\delta} \in (0,1]$, set $\delta = \bar{\delta}/(6N)$ then under Assumptions 2.1 and 2.2 the regret of Algorithm 1 is upper bounded as follows:*

$$\mathcal{R}(N) \leq \widetilde{O} \left( \left[ H \sqrt{(H+|\mathcal{S}|)|\mathcal{S}||\mathcal{A}|} + H^2 + \sqrt{\kappa} d(d^3 + B^{3/2}) \right] \sqrt{N} + (H+|\mathcal{S}|)H|\mathcal{S}||\mathcal{A}| \right),$$

*with probability at least $1 - \bar{\delta}$.*

**Proof** Let us assume that the event $\mathcal{E}_{\text{good}}$ defined in Lemma B.5 occurs. By Lemma B.5 we know that $\mathbb{P}[\mathcal{E}_{\text{good}}] \geq 1 - 6N\delta$. By the definition of the event $\mathcal{E}_1$ we know that the regret (which is defined in equation (2) above) is upper bounded as follows:

$$\mathcal{R}(N) = \sum_{t=1}^{N} V_\star - V^{(t)} \leq \sum_{t=1}^{N} \widetilde{V}_\star^{(t)} - V^{(t)}.$$

By the definition of the policy $\pi^{(t)}$ (see equation (9)) we have that

$$\widetilde{V}_\star^{(t)} = \mathbb{E}_{s_1 \sim \rho, \, \tau \sim \widehat{\mathbb{P}}_t^{\pi^\star}(\cdot|s_1)} \left[ \widetilde{\mu}_t(\widehat{\mathbf{w}}_t, \tau) \right] \leq \mathbb{E}_{s_1 \sim \rho, \, \tau \sim \widehat{\mathbb{P}}_t^{\pi^{(t)}}(\cdot|s_1)} \left[ \widetilde{\mu}_t(\widehat{\mathbf{w}}_t, \tau) \right] = \widetilde{V}^{(t)}.$$

Thus,

$$\mathcal{R}(N) \leq \sum_{t=1}^{N} \widetilde{V}^{(t)} - V^{(t)}.$$

Under event $\mathcal{E}_2$ we know that

$$\sum_{t=1}^{N} \bar{V}^{(t)} - V^{(t)} \leq \beta_N(\delta)\sqrt{8Nd\max\{\kappa,1\}\log\left(1+\frac{N}{\kappa d}\right)} + 4\sqrt{N\log\left(\frac{6\log(N)}{\delta}\right)}.$$

By combining the previous two inequalities we find that

$$\mathcal{R}(N) \leq \sum_{t=1}^{N} \widetilde{V}^{(t)} - \bar{V}^{(t)}$$

$$+ \beta_N(\delta)\sqrt{8Nd\max\{\kappa,1\}\log\left(1+\frac{N}{\kappa d}\right)} + 4\sqrt{N\log\left(\frac{6\log(N)}{\delta}\right)}.$$

Finally under event $\mathcal{E}_3$ we have a bound on the first term on the right hand side above, this leads to the bound:

$$\mathcal{R}(N) \leq (2H+1)\sum_{t=1}^{N}\sum_{h=1}^{H-1}\xi^{(t)}_{s_h^{(t)},a_h^{(t)}} + 4H^2\sqrt{N\log\left(\frac{6\log(N)}{\delta}\right)}$$

$$+ \beta_N(\delta)\sqrt{8Nd\max\{\kappa,1\}\log\left(1+\frac{N}{\kappa d}\right)} + 4\sqrt{N\log\left(\frac{6\log(N)}{\delta}\right)} + 1. \quad (39)$$

It remains to bound the term $\sum_{t=1}^{N}\sum_{h=1}^{H-1}\xi^{(t)}_{s_h^{(t)},a_h^{(t)}}$. First, note that

$$\sum_{t=1}^{N}\sum_{h=1}^{H-1}\xi^{(t)}_{s_h^{(t)},a_h^{(t)}} = \sum_{t=1}^{N}\sum_{h=1}^{H-1}\min\left\{2, 4\sqrt{\frac{\log\left(\frac{6(|\mathcal{S}||\mathcal{A}|H)^H(8NH^2)^{|\mathcal{S}|}\log(N_t(s_{h-1}^{(t)},a_{h-1}^{(t)}))}{\delta}\right)}{N_t(s_{h-1}^{(t)},a_{h-1}^{(t)})}}\right\}$$

$$\leq \sum_{t=1}^{N}\sum_{h=1}^{H-1}\min\left\{2, 4\sqrt{\frac{\log\left(\frac{6(|\mathcal{S}||\mathcal{A}|H)^H(8NH^2)^{|\mathcal{S}|}\log(N)}{\delta}\right)}{N_t(s_{h-1}^{(t)},a_{h-1}^{(t)})}}\right\}.$$

For every state-action pair $(s,a)$, the minimum in the terms above will be 2 until it is visited at least

$$N_t(s,a) \geq 4\log\left(\frac{6(|\mathcal{S}||\mathcal{A}|H)^H(8H^2N)^{|\mathcal{S}|}\log(N)}{\delta}\right) =: \spadesuit$$

number of times. Therefore,

$$\sum_{t=1}^{N}\sum_{h=1}^{H-1}\xi^{(t)}_{s_h^{(t)},a_h^{(t)}}$$

$$\leq 2|\mathcal{S}||\mathcal{A}|\spadesuit + 4\sqrt{\log\left(\frac{6(|\mathcal{S}||\mathcal{A}|H)^H(8NH^2)^{|\mathcal{S}|}\log(N))}{\delta}\right)}\sum_{t=1}^{N}\sum_{h=1}^{H-1}\frac{1}{\sqrt{N_t(s_{h-1}^{(t)},a_{h-1}^{(t)})}}$$

$$= 2|\mathcal{S}||\mathcal{A}|\spadesuit + 4\sqrt{\log\left(\frac{6(|\mathcal{S}||\mathcal{A}|H)^H(8NH^2)^{|\mathcal{S}|}\log(N))}{\delta}\right)}\sum_{s,a\in\mathcal{S}\times\mathcal{A}}\sum_{\ell=1}^{N_N(s,a)}\frac{1}{\sqrt{\ell}}$$

$$\stackrel{(i)}{<} 2|\mathcal{S}||\mathcal{A}|\spadesuit + 8\sqrt{\log\left(\frac{6(|\mathcal{S}||\mathcal{A}|H)^H(8NH^2)^{|\mathcal{S}|}\log(N))}{\delta}\right)}\sum_{s,a\in\mathcal{S}\times\mathcal{A}}\sqrt{N_N(s,a)}$$

$$\stackrel{(ii)}{\leq} 2|\mathcal{S}||\mathcal{A}|\spadesuit + 8\sqrt{\log\left(\frac{6(|\mathcal{S}||\mathcal{A}|H)^H(8NH^2)^{|\mathcal{S}|}\log(N))}{\delta}\right)|\mathcal{S}||\mathcal{A}|N}$$

$$= 8|\mathcal{S}||\mathcal{A}|\log\left(\frac{6(|\mathcal{S}||\mathcal{A}|H)^H(8H^2N)^{|\mathcal{S}|}\log(N)}{\delta}\right)$$

$$+ 8\sqrt{\log\left(\frac{6(|\mathcal{S}||\mathcal{A}|H)^H(8NH^2)^{|\mathcal{S}|}\log(N))}{\delta}\right)|\mathcal{S}||\mathcal{A}|N} \quad (40)$$

where $(i)$ follows since for all $n \in \mathbb{N}$, $\sum_{\ell=1}^n \frac{1}{\sqrt{\ell}} < 2\sqrt{n}$, and $(ii)$ follows since $\sum_{s,a \in \mathcal{S} \times \mathcal{A}} N_N(s,a) = N$ and by Jensen's inequality. Plugging this upper bound into inequality (39) we get that

$$
\begin{aligned}
\mathcal{R}(N) \leq{} & 8(2H+1)|\mathcal{S}||\mathcal{A}| \cdot \log \left( \frac{6(|\mathcal{S}||\mathcal{A}|H)^H (8H^2 N)^{|\mathcal{S}|} \log(N)}{\delta} \right) \\
& + 8(2H+1)\sqrt{\log \left( \frac{6(|\mathcal{S}||\mathcal{A}|H)^H (8NH^2)^{|\mathcal{S}|} \log(N))}{\delta} \right) |\mathcal{S}||\mathcal{A}|N} \\
& + 4H^2 \sqrt{N \log \left( \frac{6 \log(N)}{\delta} \right)} + \beta_N(\delta) \sqrt{8Nd \max\{\kappa, 1\} \log \left( 1 + \frac{N}{\kappa d} \right)} \\
& + 4\sqrt{N \log \left( \frac{6 \log(N)}{\delta} \right) + 1} \qquad (41) \\
={} & \widetilde{O}\left( \left[ H\sqrt{(H+|\mathcal{S}|)|\mathcal{S}||\mathcal{A}|} + H^2 + \sqrt{\kappa d}(d^3 + B^{3/2}) \right] \sqrt{N} + (H+|\mathcal{S}|)H|\mathcal{S}||\mathcal{A}| \right).
\end{aligned}
$$

where the last equality follows since by its definition $\beta_N(\delta) = \widetilde{O}(d^3 + B^{3/2})$ and by simplifying the expression in equation (41). This bound holds with probability $1 - 6N\delta$. Recalling that $\bar\delta = 6N\delta$ completes our proof. ∎

## C  Proof of Lemma 3.5

We begin by presenting some additional technical lemmas.

### C.1  Additional Technical Results

The first lemma pertains to a pair of positive semi-definite matrices.

**Lemma C.1.** *If $\mathbf{B} \succeq \mathbf{C} \succ \mathbf{0}$ be $d \times d$ dimensional matrices then,*

$$
\sup_{\mathbf{x} \neq 0} \frac{\mathbf{x}^\top \mathbf{B} \mathbf{x}}{\mathbf{x}^\top \mathbf{C} \mathbf{x}} \leq \frac{\det(\mathbf{B})}{\det(\mathbf{C})}.
$$

**Proof** Given any $\mathbf{y} \in \mathbb{R}^d$ let $\mathbf{x} = \mathbf{C}^{-1/2}\mathbf{y}$. Then

$$
\sup_{\mathbf{x} \neq 0} \frac{\mathbf{x}^\top \mathbf{B} \mathbf{x}}{\mathbf{x}^\top \mathbf{C} \mathbf{x}} = \sup_{\mathbf{y} \neq 0} \frac{\mathbf{y}^\top \mathbf{C}^{-1/2} \mathbf{B} \mathbf{C}^{-1/2} \mathbf{y}}{\|\mathbf{y}\|_2^2} = \left\| \mathbf{C}^{-1/2} \mathbf{B} \mathbf{C}^{-1/2} \right\|_{op}
$$

by the definition of the operator norm. Recall that by assumption $\mathbf{B} - \mathbf{C} \succeq 0$ therefore $\mathbf{C}^{-1/2} \mathbf{B} \mathbf{C}^{-1/2} - \mathbf{I} \succeq 0$, and hence all the eigenvalues of $\mathbf{C}^{-1/2} \mathbf{B} \mathbf{C}^{-1/2}$ are at least 1. Thus,

$$
\sup_{\mathbf{x} \neq 0} \frac{\mathbf{x}^\top \mathbf{B} \mathbf{x}}{\mathbf{x}^\top \mathbf{C} \mathbf{x}} \leq \left\| \mathbf{C}^{-1/2} \mathbf{B} \mathbf{C}^{-1/2} \right\|_{op} \leq \det(\mathbf{C}^{-1/2} \mathbf{B} \mathbf{C}^{-1/2}) = \frac{\det(\mathbf{B})}{\det(\mathbf{C})},
$$

where the last equality follows since $\frac{\det(\mathbf{B})}{\det(\mathbf{C})} = \det(\mathbf{C}^{-1/2}) \det(\mathbf{B}) \det(\mathbf{C}^{-1/2}) = \det(\mathbf{C}^{-1/2} \mathbf{B} \mathbf{C}^{-1/2})$. This completes the proof. ∎

Next we present a lemma that establishes guarantees for the EULER algorithm. With some abuse of notation let $r^{\mathbf{v}}(\tau) = \sum_{h \in [H]} r_h^{\mathbf{v}}$ denote the total reward over a trajectory (see definition of $r_h^{\mathbf{v}}$ in equation (13)). Let $V_{\mathbf{v}} := \max_{\pi \in \Pi} \mathbb{E}_{s_1 \sim \rho, \tau \sim \mathbb{P}^\pi(\cdot | s_1)} [r^{\mathbf{v}}(\tau)]$ denote the optimal value achieved by any policy when the reward function is $r^{\mathbf{v}}$. The following is a restatement of Lemma 3.4 from [20].

**Lemma C.2.** *There exists an absolute constant $c > 0$ such that for any $N_{\mathsf{EUL}} > 0$ and any $\delta \in (0, 1)$, with probability at least $1 - \delta$ the EULER algorithm run for $N_{\mathsf{EUL}}$ episodes outputs a set of policies*

set $\{\pi^{(\ell)}\}_{\ell=1}^{N_{\mathsf{EUL}}}$ such that $U = \mathsf{Unif}(\pi^{(1)}, \cdots, \pi^{(N_{\mathsf{EUL}})})$ satisfies:

$$V_{\mathbf{v}} - \mathbb{E}_{s_1 \sim \rho, \tau \sim \mathbb{P}^U(\cdot|s_1)} \left[ r^{\mathbf{v}}(\tau) \right] \leq c \left[ \sqrt{\frac{|\mathcal{S}||\mathcal{A}|H \log\left(\frac{|\mathcal{S}||\mathcal{A}|N_{\mathsf{EUL}}}{\delta}\right) V_{\mathbf{v}}}{N_{\mathsf{EUL}}}} + \frac{|\mathcal{S}|^2|\mathcal{A}|H^2 \log\left(\frac{|\mathcal{S}||\mathcal{A}|N_{\mathsf{EUL}}}{\delta}\right)}{N_{\mathsf{EUL}}} \right].$$

An immediate corollary is the following result.

**Corollary C.3.** *There exists an absolute constant $C_1$ such that under Assumptions 2.2, 3.3 and 3.4 if $N_{\mathsf{EUL}} \geq \dfrac{C_1|\mathcal{S}|^2|\mathcal{A}|H^2 \log\left(\frac{|\mathcal{S}||\mathcal{A}|N^2 d}{\delta\omega^2}\right)}{\omega^2}$ then with probability at least $1 - \delta$, for all $i \in \left\{1, \ldots, \dfrac{2d\log\left(1+\frac{16N}{d\omega^2}\right)}{\log(3/2)}\right\}$*

$$\mathbb{E}_{s_1 \sim \rho, \tau \sim \mathbb{P}^{U_i}(\cdot|s_1)} \left[ r^{\mathbf{v}_i}(\tau) \right] \geq \frac{\omega}{2}.$$

**Proof** Fix an $i \in \left\{1, \ldots, \dfrac{2d\log\left(1+\frac{16N}{d\omega^2}\right)}{\log(3/2)}\right\}$. By the explorability assumption (Assumption 3.4) we have that $V_{\mathbf{v}_i} \geq \omega$. By Assumption 2.2 since the feature vectors are bounded by 1 we find that

$$V_{\mathbf{v}_i} = \max_{\pi \in \Pi} \mathbb{E}_{s_1 \sim \rho, \tau \sim \mathbb{P}^\pi(\cdot|s_1)} \left[ r^{\mathbf{v}_i}(\tau) \right]$$

$$= \max_{\pi \in \Pi} \mathbb{E}_{s_1 \sim \rho, \tau \sim \mathbb{P}^\pi(\cdot|s_1)} \left[ \sum_{h=1}^H r_h^{\mathbf{v}_i}(s_h, a_h) \right]$$

$$= \max_{\pi \in \Pi} \mathbb{E}_{s_1 \sim \rho, \tau \sim \mathbb{P}^\pi(\cdot|s_1)} \left[ \sum_{h=1}^H \mathbf{v}_i^\top \phi_h(s_h, a_h) \right]$$

$$= \max_{\pi \in \Pi} \mathbb{E}_{s_1 \sim \rho, \tau \sim \mathbb{P}^\pi(\cdot|s_1)} \left[ \mathbf{v}_i^\top \phi(\tau) \right] \leq \|\phi(\tau)\|_2 \|\mathbf{v}_i\|_2 \leq 1$$

where the last inequality follows since $\mathbf{v}$ is a unit vector. Thus for any $i \in \left\{1, \ldots, \dfrac{2d\log\left(1+\frac{16N}{d\omega^2}\right)}{\log(3/2)}\right\}$, because

$$N_{\mathsf{EUL}} \geq \frac{C_1|\mathcal{S}|^2|\mathcal{A}|H^2 \log\left(\frac{|\mathcal{S}||\mathcal{A}|N^2 d}{\delta\omega^2}\right)}{\omega^2}$$

$$\geq \frac{4c|\mathcal{S}||\mathcal{A}|H \log\left(\frac{2|\mathcal{S}||\mathcal{A}|Nd\log\left(1+\frac{16N}{d\omega^2}\right)}{\delta \log(3/2)}\right)}{\omega} \max\left\{|\mathcal{S}|H, \frac{4c}{\omega}\right\},$$

where $C_1$ is a sufficiently large constant, we have the guarantee that

$$\mathbb{E}_{s_1 \sim \rho, \tau \sim \mathbb{P}^{U_i}(\cdot|s_1)} \left[ r^{\mathbf{v}_i}(\tau) \right] \geq \omega/2$$

with probability at least $1 - \dfrac{\delta \log(3/2)}{2d\log\left(1+\frac{16N}{d\omega^2}\right)}$. A union bound completes the proof. $\blacksquare$

The following lemma controls the operator norm of

$$\widehat{\mathbf{a}}_i \widehat{\mathbf{a}}_i^\top - \mathbb{E}_{s_1 \sim \rho, \tau \sim \mathbb{P}^{U_i}(\cdot|s_1)}[\phi(\tau)] \mathbb{E}_{s_1 \sim \rho, \tau \sim \mathbb{P}^{U_i}(\cdot|s_1)}[\phi(\tau)]^\top$$

when the number of evaluation episodes $N_{\mathsf{EVAL}}$ is sufficiently large.

**Lemma C.4.** *There exists a positive absolute constant $C_2$ such that for any $\omega \in (0,1)$ under Assumption 2.2 if $N_{\mathsf{EVAL}} \geq \dfrac{C_2 d^3 \log^3\left(\frac{Nd^2}{\delta\omega^2}\right)}{\omega^4}$ then with probability at least $1 - \delta$, for all $i \in \left\{1, \ldots, \dfrac{2d\log\left(1+\frac{16N}{d\omega^2}\right)}{\log(3/2)}\right\}$*

$$\left\| \widehat{\mathbf{a}}_i \widehat{\mathbf{a}}_i^\top - \mathbb{E}_{s_1 \sim \rho, \tau \sim \mathbb{P}^{U_i}(\cdot|s_1)} \left[\phi(\tau)\right] \mathbb{E}_{s_1 \sim \rho, \tau \sim \mathbb{P}^{U_i}(\cdot|s_1)} \left[\phi(\tau)\right]^\top \right\|_{op} \leq \frac{\omega^2}{32d\log\left(d\log\left(1+\frac{16N}{d\omega^2}\right)\right)}.$$

**Proof** Fix an index $i \in \left\{ 1, \ldots, \frac{2d \log\left(1 + \frac{16N}{d\omega^2}\right)}{\log(3/2)} \right\}$. Recall that the trajectories $\tau_i^{(t)}$ are drawn i.i.d. from the distribution $\rho \times \mathbb{P}^{U_i}$. By Assumption 2.2 the absolute value of each entry of $\phi(\tau_i^{(t)})$ is bounded by 1. Thus by applying Hoeffding's inequality to each coordinate and then taking a union bound over all the coordinates we get that

$$\left\| \widehat{\mathbf{a}}_i - \mathbb{E}_{s_1 \sim \rho, \tau \sim \mathbb{P}^{U_i}(\cdot|s_1)} \left[ \phi(\tau) \right] \right\|_2^2 = \left\| \frac{1}{N_{\mathsf{EVAL}}} \sum_{t=1}^{N_{\mathsf{EVAL}}} \phi(\tau_i^{(t)}) - \mathbb{E}_{s_1 \sim \rho, \tau \sim \mathbb{P}^{U_i}(\cdot|s_1)} \left[ \phi(\tau) \right] \right\|_2^2$$
$$\leq \frac{c' d \log \left( \frac{2d^2 \log\left(1 + \frac{16N}{d\omega^2}\right)}{\delta \log(3/2)} \right)}{N_{\mathsf{EVAL}}}$$

with probability at least $1 - \delta \log(3/2)/\left(2d \log\left(1 + \frac{16N}{d\omega^2}\right)\right)$, where $c'$ is a positive absolute constant. Assume that this event above holds, then by the triangle inequality

$$\left\| \widehat{\mathbf{a}}_i \widehat{\mathbf{a}}_i^\top - \mathbb{E}_{s_1 \sim \rho, \tau \sim \mathbb{P}^{U_i}(\cdot|s_1)} \left[ \phi(\tau) \right] \mathbb{E}_{s_1 \sim \rho, \tau \sim \mathbb{P}^{U_i}(\cdot|s_1)} \left[ \phi(\tau) \right]^\top \right\|_{op}$$

$$\leq \left\| \widehat{\mathbf{a}}_i - \mathbb{E}_{s_1 \sim \rho, \tau \sim \mathbb{P}^{U_i}(\cdot|s_1)} \left[ \phi(\tau) \right] \right\|_2^2$$
$$+ 2 \left\| \mathbb{E}_{s_1 \sim \rho, \tau \sim \mathbb{P}^{U_i}(\cdot|s_1)} \left[ \phi(\tau) \right] \right\|_2 \left\| \widehat{\mathbf{a}}_i - \mathbb{E}_{s_1 \sim \rho, \tau \sim \mathbb{P}^{U_i}(\cdot|s_1)} \left[ \phi(\tau) \right] \right\|_2$$

$$\overset{(i)}{\leq} 2 \sqrt{ \frac{c' d \log \left( \frac{2d^2 \log\left(1 + \frac{16N}{d\omega^2}\right)}{\delta \log(3/2)} \right)}{N_{\mathsf{EVAL}}} } + \frac{c' d \log \left( \frac{2d^2 \log\left(1 + \frac{16N}{d\omega^2}\right)}{\delta \log(3/2)} \right)}{N_{\mathsf{EVAL}}}$$

$$\overset{(ii)}{\leq} \frac{\omega^2}{32 d \log \left( d \log \left(1 + \frac{16N}{d\omega^2}\right) \right)}$$

where $(i)$ follows since $\|\phi(\tau)\| \leq 1$, and $(ii)$ holds because since $\omega < 1$ and since

$$N_{\mathsf{EVAL}} \geq \frac{C_2 d^3 \log^3 \left( \frac{N d^2}{\delta \omega^2} \right)}{\omega^4} \geq \frac{c'(32)^2 d^3 \log \left( \frac{2d^2 \log\left(1 + \frac{16N}{d\omega^2}\right)}{\delta \log(3/2)} \right) \log^2 \left( d \log(1 + \frac{16N}{d\omega^2}) \right)}{\omega^4}$$

where $C_2$ is a large enough positive constant. This shows that the operator norm bound holds for a fixed index $i$ with probability at least $1 - \delta \log(3/2)/\left(2d \log\left(1 + \frac{16N}{d\omega^2}\right)\right)$. Taking a union bound over all $i \in \left\{ 1, \ldots, \frac{2d \log\left(1 + \frac{16N}{d\omega^2}\right)}{\log(3/2)} \right\}$ completes the proof. ∎

With these lemmas in place we are now ready to prove Lemma 3.5.

## C.2 The Proof

First we restate the lemma.

**Lemma 3.5.** *There exist positive absolute constants $C_1$ and $C_2$ such that, under Assumptions 2.2, 3.3 and 3.4, if Algorithm 2 is run with $N_{\mathsf{EUL}} = \frac{C_1 |\mathcal{S}|^2 |\mathcal{A}| H^2 \log \left( \frac{|\mathcal{S}||\mathcal{A}| N^2 d}{\delta \omega^2} \right)}{\omega^2}$ and $N_{\mathsf{EVAL}} = \frac{C_2 d^3 \log^3 \left( \frac{N d^2}{\delta \omega^2} \right)}{\omega^4}$, and $N > \frac{d \log\left(1 + \frac{16N}{d\omega^2}\right)}{\log(3/2)} \left(N_{\mathsf{EUL}} + N_{\mathsf{EVAL}}\right) =: \bar{N}_{\exp}$ then, with probability at least $1 - 2\delta$, we have $N_{\exp} \leq \bar{N}_{\exp}$ and furthermore:*

$$\mathbb{E}_{s_1 \sim \rho, \, \tau \sim \mathbb{P}^{\bar{U}}(\cdot|s_1)} \left[ \phi(\tau) \phi(\tau)^\top \right] \succeq \frac{\omega^2 \log(3/2)}{32 d \log \left( d \log \left(1 + \frac{16N}{d\omega^2}\right) \right)} \mathbf{I}.$$

**Proof** Assume the events described in both Corollary C.3 and Lemma C.4 occur. Since $N_{\mathsf{EUL}}$ and $N_{\mathsf{EVAL}}$ are both appropriately large this happens with probability at least $1 - 2\delta$.

We shall begin by showing that the number of while loop iterations $n_{\mathsf{loop}}$ is bounded by $\frac{d \log\left(1 + \frac{16N}{d\omega^2}\right)}{\log(3/2)}$. At any iteration $n \leq \frac{2d \log\left(1 + \frac{16N}{d\omega^2}\right)}{\log(3/2)}$ by the event in Corollary C.3 we know that

the mixture $U_n$ satisfies

$$\left(\mathbb{E}_{s_1 \sim \rho, \tau \sim \mathbb{P}^{U_n}(\cdot | s_1)}\left[\mathbf{v}_n^\top \phi(\tau)\right]\right)^2 \geq \frac{\omega^2}{4}.$$

Thus,

$$
\begin{aligned}
\mathbf{v}_n^\top \mathbf{A}_n \mathbf{v}_n &= \mathbf{v}_n^\top \left(\mathbf{A}_{n-1} + \widehat{\mathbf{a}}_n \widehat{\mathbf{a}}_n^\top\right) \mathbf{v}_n \\
&\geq \mathbf{v}_n^\top \widehat{\mathbf{a}}_n \widehat{\mathbf{a}}_n^\top \mathbf{v}_n \\
&= \left(\mathbb{E}_{s_1 \sim \rho, \tau \sim \mathbb{P}^{U_n}(\cdot | s_1)}\left[\mathbf{v}_n^\top \phi(\tau)\right]\right)^2 + \left(\mathbf{v}_n^\top \widehat{\mathbf{a}}_n\right)^2 - \left(\mathbb{E}_{s_1 \sim \rho, \tau \sim \mathbb{P}^{U_n}(\cdot | s_1)}\left[\phi(\tau)^\top \mathbf{v}_n\right]\right)^2 \\
&\geq \frac{\omega^2}{4} + \left(\mathbf{v}_n^\top \widehat{\mathbf{a}}_n\right)^2 - \left(\mathbb{E}_{s_1 \sim \rho, \tau \sim \mathbb{P}^{U_n}(\cdot | s_1)}\left[\mathbf{v}_n^\top \phi(\tau)\right]\right)^2 \\
&\geq \frac{\omega^2}{4} - \left\|\widehat{\mathbf{a}}_n \widehat{\mathbf{a}}_n^\top - \mathbb{E}_{s_1 \sim \rho, \tau \sim \mathbb{P}^{U_n}(\cdot | s_1)}\left[\phi(\tau)\right] \mathbb{E}_{s_1 \sim \rho, \tau \sim \mathbb{P}^{U_n}(\cdot | s_1)}\left[\phi(\tau)\right]^\top\right\|_{op} \\
&\stackrel{(i)}{\geq} \frac{\omega^2}{4} - \frac{\omega^2}{32 d \log\left(d \log\left(1 + \frac{16N}{d\omega^2}\right)\right)} > \frac{3\omega^2}{16},
\end{aligned}
$$

where inequality $(i)$ follows by the event in Lemma C.4. Further if $n \leq n_{\mathsf{loop}}$ and the algorithm didn't terminate after iteration $n-1$, we must have

$$\mathbf{v}_n^\top \mathbf{A}_{n-1} \mathbf{v}_n < \frac{\omega^2}{8}.$$

Therefore by Lemma C.1 for any $n \leq \min\left\{n_{\mathsf{loop}}, \frac{2d\log\left(1 + \frac{16N}{d\omega^2}\right)}{\log(3/2)}\right\}$ we have

$$\frac{3}{2} = \frac{\frac{3\omega^2}{16}}{\frac{\omega^2}{8}} < \frac{\mathbf{v}_n^\top \mathbf{A}_n \mathbf{v}_n}{\mathbf{v}_n^\top \mathbf{A}_{n-1} \mathbf{v}_n} \leq \frac{\det(\mathbf{A}_n)}{\det(\mathbf{A}_{n-1})}.$$

Thus for any $n \leq \min\left\{n_{\mathsf{loop}}, \frac{2d\log\left(1 + \frac{16N}{d\omega^2}\right)}{\log(3/2)}\right\}$,

$$\det(\mathbf{A}_n) > \frac{3}{2}\det(\mathbf{A}_{n-1}) \geq \left(\frac{3}{2}\right)^n \det(\mathbf{A}_0) = \left(\frac{3}{2}\right)^n \left(\frac{\omega^2}{16}\right)^d. \tag{42}$$

The matrix $\mathbf{A}_n$ is obtained as a result of a sequence of rank 1 updates, where each update has its norm bounded ($\|\widehat{\mathbf{a}}_n\|_2 \leq 1$ for all $n$), so by Lemma A.3:

$$\log\left(\det(\mathbf{A}_n)\right) \leq d\log\left(\frac{\omega^2}{16} + \frac{n}{d}\right). \tag{43}$$

Combining inequalities (42) and (43) we conclude that, for any $n \leq \min\left\{n_{\mathsf{loop}}, \frac{2d\log\left(1 + \frac{16N}{d\omega^2}\right)}{\log(3/2)}\right\}$

$$n\log(3/2) + d\log\left(\frac{\omega^2}{16}\right) \leq \log\left(\det(\mathbf{A}_n)\right) \leq d\log\left(\frac{\omega^2}{16} + \frac{n}{d}\right).$$

Therefore, if $n_{\mathsf{loop}} < N$, then then while loop must terminate after at most

$$
\begin{aligned}
n_{\mathsf{loop}} &\leq \frac{d\log\left(1 + \frac{16n_{\mathsf{loop}}}{d\omega^2}\right)}{\log(3/2)} \\
&\leq \frac{d\log\left(1 + \frac{16N}{d\omega^2}\right)}{\log(3/2)}
\end{aligned} \tag{44}
$$

loops. To verify that $n_{\mathsf{loop}} < N$, notice that by assumption $N$ is such that

$$\frac{N}{N_{\mathsf{EUL}} + N_{\mathsf{EVAL}}} > \frac{d\log\left(1 + \frac{16N}{d\omega^2}\right)}{\log(3/2)}. \tag{45}$$

Therefore inequality (44) is a valid upper bound on $n_{\text{loop}}$.

Thus, we know that the total number of episodes taken by the algorithm to terminate

$$N_{\text{exp}} = n_{\text{loop}} \times (N_{\text{EUL}} + N_{\text{EVAL}}) \leq \frac{d \log \left(1 + \frac{16N}{d\omega^2}\right)}{\log(3/2)} (N_{\text{EUL}} + N_{\text{EVAL}}) = \bar{N}_{\text{exp}}.$$

This proves the first part of the lemma. For the second part notice that for an arbitrary unit vector $\mathbf{v} \in \mathbb{R}^d$

$$\mathbf{v}^\top \mathbb{E}_{s_1 \sim \rho, \tau \sim \mathbb{P}^{\bar{U}}(\cdot|s_1)} \left[\phi(\tau)\phi(\tau)^\top\right] \mathbf{v}$$

$$= \mathbb{E}_{s_1 \sim \rho, \tau \sim \mathbb{P}^{\bar{U}}(\cdot|s_1)} \left[\left(\mathbf{v}^\top \phi(\tau)\right)^2\right]$$

$$\stackrel{(i)}{=} \frac{1}{n_{\text{loop}}} \sum_{i=1}^{n_{\text{loop}}} \mathbb{E}_{s_1 \sim \rho, \tau \sim \mathbb{P}^{U_i}(\cdot|s_1)} \left[\left(\mathbf{v}^\top \phi(\tau)\right)^2\right]$$

$$\stackrel{(ii)}{\geq} \frac{1}{n_{\text{loop}}} \sum_{i=1}^{n_{\text{loop}}} \left(\mathbf{v}^\top \mathbb{E}_{s_1 \sim \rho, \tau \sim \mathbb{P}^{U_i}(\cdot|s_1)} \left[\phi(\tau)\right]\right)^2$$

$$= \frac{1}{n_{\text{loop}}} \left[\mathbf{v}^\top \mathbf{A}_{n_{\text{loop}}} \mathbf{v} - \mathbf{v}^\top \mathbf{A}_{n_{\text{loop}}} \mathbf{v} + \sum_{i=1}^{n_{\text{loop}}} \left(\mathbf{v}^\top \mathbb{E}_{s_1 \sim \rho, \tau \sim \mathbb{P}^{U_i}(\cdot|s_1)} \left[\phi(\tau)\right]\right)^2\right]$$

$$\stackrel{(iii)}{\geq} \frac{\omega^2}{8n_{\text{loop}}} - \frac{\omega^2}{16n_{\text{loop}}} + \frac{1}{n_{\text{loop}}} \left[\sum_{i=1}^{n_{\text{loop}}} \left(\left(\mathbf{v}^\top \mathbb{E}_{s_1 \sim \rho, \tau \sim \mathbb{P}^{U_i}(\cdot|s_1)} \left[\phi(\tau)\right]\right)^2 - \left(\mathbf{v}^\top \widehat{\mathbf{a}}_i\right)^2\right)\right]$$

$$\geq \frac{\omega^2}{16n_{\text{loop}}} - \max_{i \in n_{\text{loop}}} \left\|\widehat{\mathbf{a}}_i \widehat{\mathbf{a}}_i^\top - \mathbb{E}_{s_1 \sim \rho, \tau \sim \mathbb{P}^{U_i}(\cdot|s_1)} \left[\phi(\tau)\right] \mathbb{E}_{s_1 \sim \rho, \tau \sim \mathbb{P}^{U_i}(\cdot|s_1)} \left[\phi(\tau)\right]^\top\right\|_{op}$$

$$\stackrel{(iv)}{\geq} \frac{\omega^2 \log(3/2)}{32d \log \left(d \log \left(1 + \frac{16N}{d\omega^2}\right)\right)}$$

where $(i)$ is by the definition of $\bar{U}$ as the uniform mixture over $U_1, \ldots, U_{n_{\text{loop}}}$, $(ii)$ follows by Jensen's inequality, $(iii)$ is because the minimum eigenvalue of $\mathbf{A}_{n_{\text{loop}}}$ is at least $\omega^2/8$ and since $\mathbf{A}_{n_{\text{loop}}} = \frac{\omega^2}{16}\mathbf{I} + \sum_{i=1}^{n_{\text{loop}}} \widehat{\mathbf{a}}_i \widehat{\mathbf{a}}_i^\top$, and finally $(iv)$ is by the upper bound on $n_{\text{loop}} \leq d \log \left(d \log \left(1 + \frac{16N}{d\omega^2}\right)\right)$ established above and by the bound on the operator norm of the difference of the matrices established in Lemma C.4. This wraps up our proof. ■

# D    Regret Analysis of Algorithm 3 under the explorability assumption

In this algorithm we use the sum-decomposable bonus functions. Throughout this section we assume that Assumptions 2.1, 2.2, 3.3 and 3.4 are in force, and that $N_{\text{EXP}}$ and $N_{\text{EVAL}}$ are chosen as specified by the statement of Theorem 3.6. We also assume that the number of episodes $N > \bar{N}_{\text{exp}}$ (see its definition in Lemma 3.5). Define the following two quantities that shall be useful in this section

$$t_0 := C_3 \left[\frac{d^2 \log^2(d \log(1 + \frac{16N}{d\omega^2}))}{\omega^4} \sqrt{\log(N/\delta)} + N_{\text{exp}}^{2/3}\right]^{3/2} \tag{46a}$$

$$\Psi_t := \frac{128d \log \left(d \log \left(1 + \frac{16N}{d\omega^2}\right)\right)}{3 \log(3/2)\omega^2} \cdot \frac{t - N_{\text{exp}}}{t^{2/3} - (N_{\text{exp}} + 1)^{2/3}}, \tag{46b}$$

where $C_3$ is a large enough positive absolute constant.

## D.1    A Sandwich Inequality

As a first step to showing that these bonuses also lead to optimistic value functions we have a sandwich inequality that relates $\|\phi(\tau)\|_{\boldsymbol{\Sigma}_t^{-1}}$ to the sum decomposable bonus $\sum_{h=1}^{H} \|\phi_h(s_h, a_h)\|_{\boldsymbol{\Sigma}_t^{-1}}$.

**Lemma D.1.** *For any $\tau \in \Gamma$*

$$\|\phi(\tau)\|_{\mathbf{\Sigma}_t^{-1}} \overset{(a)}{\leq} \sum_{h=1}^{H} \|\phi_h(s_h, a_h)\|_{\mathbf{\Sigma}_t^{-1}} \overset{(b)}{\leq} \sqrt{H \frac{\lambda_{\max}(\mathbf{\Sigma}_t)}{\lambda_{\min}(\mathbf{\Sigma}_t)}} \|\phi(\tau)\|_{\mathbf{\Sigma}_t^{-1}}.$$

**Proof** Since $\phi(\tau) = \sum_{h=1}^{H} \phi_h(s_h, a_h)$ the inequality $(a)$ holds by invoking the triangle inequality. Now to prove inequality $(b)$ note that

$$
\begin{aligned}
\sum_{h=1}^{H} \|\phi_h(s_h, a_h)\|_{\mathbf{\Sigma}_t^{-1}} &\leq \sqrt{\lambda_{\max}(\mathbf{\Sigma}_t^{-1})} \sum_{h=1}^{H-1} \|\phi_h(s_h, a_h)\|_2 \\
&\overset{(i)}{\leq} \sqrt{H \lambda_{\max}(\mathbf{\Sigma}_t^{-1})} \sqrt{\sum_{h=1}^{H-1} \|\phi_h(s_h, a_h)\|_2^2} \\
&\overset{(ii)}{=} \sqrt{H \lambda_{\max}(\mathbf{\Sigma}_t^{-1})} \|\phi(\tau)\|_2 \\
&\leq \sqrt{H \frac{\lambda_{\max}(\mathbf{\Sigma}_t^{-1})}{\lambda_{\min}(\mathbf{\Sigma}_t^{-1})}} \|\phi(\tau)\|_{\mathbf{\Sigma}_t^{-1}} \\
&= \sqrt{H \frac{\lambda_{\max}(\mathbf{\Sigma}_t)}{\lambda_{\min}(\mathbf{\Sigma}_t)}} \|\phi(\tau)\|_{\mathbf{\Sigma}_t^{-1}}
\end{aligned}
$$

where $(i)$ holds because for any vector $\mathbf{z} \in \mathbb{R}^H$, $\|\mathbf{z}\|_1 \leq \sqrt{H}\|\mathbf{z}\|_2$ and $(ii)$ is a consequence of Assumption 3.4 since $\phi_h$ and $\phi_{h'}$ are orthogonal for $h \neq h'$ and because $\phi$ is sum-decomposable by Assumption 3.3. ∎

In light of the previous lemma we now establish bounds on the condition number of the matrices $\mathbf{\Sigma}_t^{-1}$ in the next subsection.

## D.2 Bound on the Condition Number of $\mathbf{\Sigma}_t$

To bound the condition number we separately upper bound the maximum eigenvalue and lower bound the minimum eigenvalue. Since we have assumed that $\|\phi(\tau)\|_2 \leq 1$, a simple upper bound on the maximum eigenvalue of $\mathbf{\Sigma}_t = \kappa \mathbf{I} + \sum_{q=N_{\exp}+1}^{t} \phi(\tau^{(q)})\phi(\tau^{(q)})^\top$ is

$$\lambda_{\max}(\mathbf{\Sigma}_t) \leq \kappa + (t - N_{\exp}). \tag{47}$$

Let us now derive a lower bound for the smallest eigenvalue. To do this we shall relate the smallest eigenvalue of $\mathbf{\Sigma}_t$ to the smallest eigenvalue of the covariance matrix associated with the exploration policy

$$\bar{\mathbf{\Sigma}} := \mathbb{E}_{s_1 \sim \rho, \tau \sim \mathbb{P}^{\bar{U}}(\cdot|s_1)} \left[ \phi(\tau)\phi(\tau)^\top \right].$$

In Lemma 3.5 we derived a high probability lower bound on the minimum eigenvalue of this matrix.

**Lemma D.2.** *With probability at least $1 - 3\delta$ for all $t \in \{N_{\exp} + 1, \ldots, N\}$:*

$$\lambda_{\min}(\mathbf{\Sigma}_t) \geq \begin{cases} \kappa + \frac{3\left(t^{2/3} - (N_{\exp}+1)^{2/3}\right)\omega^2 \log(3/2)}{128d \log\left(d \log\left(1 + \frac{16N}{d\omega^2}\right)\right)} & \text{when } t \geq t_0 \\ \kappa & \text{o.w.,} \end{cases}$$

*where $t_0$ is defined in equation (46a).*

**Proof** First let us dispense of the case where $N < t_0$. Since $\mathbf{\Sigma}_t \succeq \kappa \mathbf{I}$ we are done.

Therefore going forward let us assume that $N \geq t_0$. Recall that $b_q$ are the Bernoulli random variables used in Algorithm 3 and that $\mathbb{P}(b_q = 1) = 1/q^{1/3}$. Notice that the following holds

$$
\Sigma_t = \kappa\mathbf{I} + \sum_{q=N_{\exp}+1}^{t} \phi(\tau^{(q)})\phi(\tau^{(q)})^\top
$$

$$
\succeq \kappa\mathbf{I} + \sum_{q=N_{\exp}+1}^{t} b_q\phi(\tau^{(q)})\phi(\tau^{(q)})^\top
$$

$$
= \kappa\mathbf{I} + \sum_{q=N_{\exp}+1}^{t} \frac{1}{q^{1/3}}\bar{\Sigma} + \underbrace{\sum_{q=N_{\exp}+1}^{t} \left( b_q\phi(\tau^{(q)})\phi(\tau^{(q)})^\top - \frac{1}{q^{1/3}}\bar{\Sigma} \right)}_{=:\mathbf{E}_t}.
$$

Thus we have

$$
\lambda_{\min}(\Sigma_t) \geq \kappa + \lambda_{\min}(\bar{\Sigma}) \sum_{q=N_{\exp}+1}^{t} \frac{1}{q^{1/3}} - \lambda_{\max}(\mathbf{E}_t)
$$

$$
\geq \kappa + \lambda_{\min}(\bar{\Sigma}) \int_{q=N_{\exp}+1}^{t} \frac{1}{q^{1/3}} \, dq - \lambda_{\max}(\mathbf{E}_t)
$$

$$
= \kappa + \frac{3\left(t^{2/3} - (N_{\exp}+1)^{2/3}\right)}{2}\lambda_{\min}(\bar{\Sigma}) - \lambda_{\max}(\mathbf{E}_t). \tag{48}
$$

First by Lemma 3.5 we know that

$$
\lambda_{\min}(\bar{\Sigma}) \geq \frac{\omega^2 \log(3/2)}{32 d \log\left(d \log\left(1 + \frac{16N}{d\omega^2}\right)\right)} \tag{49}
$$

with probability at least $1 - 2\delta$.

Next to upper bound the maximum eigenvalue of $\mathbf{E}_t$ define the matrix martingale difference sequence

$$
\mathbf{D}_q := b_q\phi(\tau^{(q)})\phi(\tau^{(q)})^\top - \frac{1}{q^{1/3}}\bar{\Sigma}.
$$

Observe that $\mathbf{E}_t = \sum_{q=N_{\exp}+1}^{t} \mathbf{D}_q$. We will use the matrix Freedman inequality (Theorem A.2) to upper bound the maximum eigenvalue of $\mathbf{E}_t$. To this end first note that

$$
\lambda_{\max}(\mathbf{D}_q) \leq \|\phi(\tau^{(q)})\phi(\tau^{(q)})^\top\|_{op} \leq 1.
$$

Further note that

$$
\left\| \sum_{q=N_{\exp}+1}^{t} \mathbb{E}\left[ \mathbf{D}_q^2 \mid \mathbf{D}_{N_{\exp}+1}, \ldots, \mathbf{D}_{q-1} \right] \right\|_{op}
$$

$$
\leq \sum_{q=N_{\exp}+1}^{t} \left\| \mathbb{E}\left[ \mathbf{D}_q^2 \mid \mathbf{D}_{N_{\exp}+1}, \ldots, \mathbf{D}_{q-1} \right] \right\|_{op}
$$

$$
= \sum_{q=N_{\exp}+1}^{t} \left\| \mathbb{E}\left[ b_q^2\|\phi(\tau^{(q)})\|_2^2\phi(\tau^{(q)})\phi(\tau^{(q)})^\top + \frac{\bar{\Sigma}^2}{q^{2/3}} \right.\right.
$$

$$
\left.\left. - b_q\left( \phi(\tau^{(q)})\phi(\tau^{(q)})^\top\bar{\Sigma} + \bar{\Sigma}\phi(\tau^{(q)})\phi(\tau^{(q)})^\top \right) \mid \mathbf{D}_{N_{\exp}+1}, \ldots, \mathbf{D}_{q-1} \right]\right\|_{op}
$$

$$
\overset{(i)}{\leq} \sum_{q=N_{\exp}+1}^{t} \left( \frac{1}{q^{1/3}} + \frac{1}{q^{2/3}} + \frac{2}{q^{1/3}} \right)
$$

$$
\leq 4 \sum_{q=N_{\exp}+1}^{t} \frac{1}{q^{1/3}} \leq 4 \int_{q=N_{\exp}}^{t} \frac{1}{q^{1/3}} \, dq = 6\left(t^{2/3} - N_{\exp}^{2/3}\right)
$$

where $(i)$ follows since $\mathbb{E}[b_q] = \mathbb{E}[b_q^2] = 1/q^{1/3}$ and because $\|\phi(\tau)\|_2 \leq 1$.

Now we apply Theorem A.2 with the choices

$$x = \frac{3\left(t^{2/3} - (N_{\exp}+1)^{2/3}\right)\omega^2 \log(3/2)}{128d \log\left(d \log\left(1 + \frac{16N}{d\omega^2}\right)\right)};$$

$$V = 6\left(t^{2/3} - N_{\exp}^{2/3}\right);$$

$$R = 1,$$

to get

$$\mathbb{P}\left[\lambda_{\max}(\mathbf{E}_t) \geq x\right] \leq d \exp\left(-\frac{x^2/2}{V + x/3}\right) \leq d \exp\left(-\frac{x^2}{4V}\right)$$

where the second inequality follows since $V > x/3$. Now by the choice of $x$ and $V$ we know that

$$\mathbb{P}\left[\lambda_{\max}(\mathbf{E}_t) \geq \frac{3\left(t^{2/3} - (N_{\exp}+1)^{2/3}\right)\omega^2 \log(3/2)}{128d \log\left(d \log\left(1 + \frac{16N}{d\omega^2}\right)\right)}\right] \leq \delta/N$$

whenever

$$t \geq C_3 \left[\frac{d^2 \log^2(d \log(1 + \frac{16N}{d\omega^2}))}{\omega^4} - N_{\exp}^{2/3}\right]^{3/2} = t_0$$

because the constant $C_3$ is chosen to be large enough. Thus, by a union bound we know that

$$\mathbb{P}\left[\exists t \in \{t_0, \ldots, N\} \ : \ \lambda_{\max}(\mathbf{E}_t) \geq \frac{3\left(t^{2/3} - (N_{\exp}+1)^{2/3}\right)\omega^2 \log(3/2)}{128d \log\left(d \log\left(1 + \frac{16N}{d\omega^2}\right)\right)}\right] \leq \delta. \tag{50}$$

Combining the inequalities (48), (49) and (50) completes our proof. ∎

Next we have a lemma that bounds the condition number

**Lemma D.3.** *With probability at least $1 - 3\delta$ for all $t \in \{N_{\exp} + 1, \ldots, N\}$*

$$\frac{\lambda_{\max}(\mathbf{\Sigma}_t)}{\lambda_{\min}(\mathbf{\Sigma}_t)} \leq \begin{cases} \Psi_t & \text{when } t \geq t_0 \\ 1 + \frac{(t - N_{\exp})}{\kappa} & \text{o.w.,} \end{cases}$$

*where $t_0$ and $\Psi_N$ are defined in equations (46a) and (46b) respectively.*

**Proof** The following bound holds with probability at least $1 - 3\delta$ by combing the upper bound on the maximum eigenvalue in inequality (47) with the results of Lemma D.2

$$\frac{\lambda_{\max}(\mathbf{\Sigma}_t)}{\lambda_{\min}(\mathbf{\Sigma}_t)} \leq \begin{cases} \frac{\kappa + (t - N_{\exp})}{\kappa + \frac{3\left(t^{2/3} - (N_{\exp}+1)^{2/3}\right)\omega^2 \log(3/2)}{128d \log\left(d \log\left(1 + \frac{16N}{d\omega^2}\right)\right)}} & \text{when } t \geq t_0 \\ 1 + \frac{(t - N_{\exp})}{\kappa} & \text{o.w.} \end{cases}$$

Now for any $a, b, c > 0$: $\frac{a+c}{b+c} \leq \frac{a}{b}$ if $a > b$. Therefore we can simplify the expression above in case where $t \geq t_0$ to get

$$\frac{\lambda_{\max}(\mathbf{\Sigma}_t)}{\lambda_{\min}(\mathbf{\Sigma}_t)} \leq \begin{cases} \frac{128d \log\left(d \log\left(1 + \frac{16N}{d\omega^2}\right)\right)}{3 \log(3/2)\omega^2} \cdot \frac{t - N_{\exp}}{t^{2/3} - (N_{\exp}+1)^{2/3}} & \text{when } t \geq t_0 \\ 1 + \frac{(t - N_{\exp})}{\kappa} & \text{o.w.} \end{cases}$$

By recalling the definition of $\Psi_t$ from above the claim follows. ∎

### D.3 Definition and Properties of Another "Good Event" $\mathcal{E}_{\text{good}}^{\text{sd}}$

Similar to the proof of Theorem 3.2 the proof of Theorem 3.6 also proceeds by showing that a different favorable event $\mathcal{E}_{\text{good}}^{\text{sd}}$ occurs with high probability. We shall upper bound the regret of Algorithm 3 when this favorable event occurs. Before defining this event we need some additional notation.

**Definition D.4.** *For all $t \in [N]$, given any policy $\pi$ define*

$$\bar{V}_t^{\pi,\text{sd}} := \mathbb{E}_{s_1 \sim \rho, \ \tau \sim \mathbb{P}^\pi(\cdot|s_1)} \left[ \bar{\mu}_t^{\text{sd}}(\widehat{\mathbf{w}}_t, \tau) \right],$$

*where recall from equation (11a) that $\bar{\mu}_t^{\text{sd}}(\widehat{\mathbf{w}}_t, \tau) = \min \left\{ \mu\left(\widehat{\mathbf{w}}_t^\top \phi(\tau)\right) + \sqrt{\kappa}\beta_t(\delta) \sum_{h=1}^{H} \|\phi_h(s_h, a_h)\|_{\boldsymbol{\Sigma}_t^{-1}}, 1 \right\}$. Further, for all episodes $t \in [N]$ also define $\bar{V}^{(t),\text{sd}} := \bar{V}_t^{\pi^{(t)},\text{sd}}$ and $\bar{V}_\star^{(t),\text{sd}} := \bar{V}_t^{\pi_\star,\text{sd}}$.*

Also define the value function when the average rewards are $\widetilde{\mu}_t^{\text{sd}}(\widehat{\mathbf{w}}_t, \tau)$ and the transition dynamics are governed by $\widehat{\mathbb{P}}_t$.

**Definition D.5.** *For any episode $t \in [N]$, given any policy $\pi \in \Pi$ define*

$$\widetilde{V}_t^{\pi,\text{sd}} := \mathbb{E}_{s_1 \sim \rho, \ \tau \sim \widehat{\mathbb{P}}_t^\pi(\cdot|s_1)} \left[ \widetilde{\mu}_t^{\text{sd}}(\widehat{\mathbf{w}}_t, \tau) \right] \tag{51}$$

*where $\widetilde{\mu}_t^{\text{sd}}$ is defined above in equation (11b). To simplify notation we additionally define $\widetilde{V}^{(t),\text{sd}} := \widetilde{V}_t^{\pi^{(t)},\text{sd}}$ and $\widetilde{V}_\star^{(t),\text{sd}} := \widetilde{V}_t^{\pi_\star,\text{sd}}$.*

Recall the definition of $t_0$ from equation (46a) above and consider the following events:

$$\mathcal{E}_1^{\text{sd}} := \left\{ \sum_{t=t_0+1}^{N} (1 - b_t)V_\star \leq \sum_{t=t_0+1}^{N} (1 - b_t)\widetilde{V}_\star^{(t),\text{sd}} \right\}; \tag{52a}$$

$$\mathcal{E}_2^{\text{sd}} := \left\{ \sum_{t=t_0+1}^{N} (1 - b_t)\left(\bar{V}^{(t),\text{sd}} - V^{(t)}\right) \leq \beta_N(\delta)(1 + \sqrt{H\Psi_N})\sqrt{8Nd \max\{\kappa, 1\} \log\left(1 + \frac{N}{\kappa d}\right)} \right.$$

$$\left. + 4\sqrt{N \log\left(\frac{6\log(N)}{\delta}\right)} \right\}; \tag{52b}$$

$$\mathcal{E}_3^{\text{sd}} := \left\{ \sum_{t=t_0+1}^{N} (1 - b_t)\left(\widetilde{V}^{(t),\text{sd}} - \bar{V}^{(t),\text{sd}}\right) \leq (2H + 1) \sum_{t=t_0+1}^{N} \sum_{h=1}^{H-1} \xi_{s_h^{(t)}, a_h^{(t)}}^{(t)} \right.$$

$$\left. + 4H^2 \sqrt{N \log\left(\frac{6\log(N)}{\delta}\right)} + 1 \right\}; \tag{52c}$$

$$\mathcal{E}_4^{\text{sd}} := \left\{ \sum_{t=t_0+1}^{N} b_t \leq \left(\frac{20}{3}\log\left(\frac{1}{\delta}\right)\right)^{3/2} + 4N^{2/3} \right\}; \tag{52d}$$

$$\mathcal{E}_5^{\text{sd}} := \left\{ N_{\text{EXP}} \leq \frac{d \log\left(1 + \frac{16N}{d\omega^2}\right)}{\log(3/2)}(N_{\text{EUL}} + N_{\text{EVAL}}) \right\}, \tag{52e}$$

where $(s_h^{(t)}, a_h^{(t)})$ is the state-action pair visited at step $h$ during episode $t$. In the definitions of the events above if $N < t_0 + 1$ and the sums are "empty" then we take their value to be zero.

**Lemma D.6.** *Define the event $\mathcal{E}_{\text{good}}^{\text{sd}} := \mathcal{E}_1^{\text{sd}} \cap \mathcal{E}_2^{\text{sd}} \cap \mathcal{E}_3^{\text{sd}} \cap \mathcal{E}_4^{\text{sd}} \cap \mathcal{E}_5^{\text{sd}}$. If $N > \bar{N}_{\text{exp}}$ then $\mathbb{P}\left[\mathcal{E}_{\text{good}}^{\text{sd}}\right] \geq 1 - 12N\delta$.*

**Proof** We will show that each of the five events $\mathcal{E}_1^{\text{sd}}$, $\mathcal{E}_2^{\text{sd}}$, $\mathcal{E}_3^{\text{sd}}$, $\mathcal{E}_4^{\text{sd}}$ and $\mathcal{E}_5^{\text{sd}}$ occurs with a high probability and take union bound to prove our claim.

**Event $\mathcal{E}_1^{\sf sd}$:** By invoking Lemma B.1 $N - t_0$ times, once per episode, with the choice $\eta = 1$ we get

$$
\sum_{t=t_0}^{N} (1 - b_t) V_\star = \sum_{t=t_0}^{N} (1 - b_t) \mathbb{E}_{s_1 \sim \rho, \; \tau \sim \mathbb{P}^{\pi_\star}(\cdot | s_1)} \left[ \mu(\mathbf{w}_\star^\top \phi(\tau)) \right]
$$

$$
\leq \sum_{t=t_0}^{N} (1 - b_t) \mathbb{E}_{s_1 \sim \rho, \; \tau \sim \widehat{\mathbb{P}}_t^{\pi_\star}(\cdot | s_1)} \left[ \mu(\mathbf{w}_\star^\top \phi(\tau)) + \sum_{h=1}^{H-1} \bar{\xi}_{s_h, a_h}^{(t)}(1) \right]
$$

$$
\leq \sum_{t=t_0}^{N} (1 - b_t) \mathbb{E}_{s_1 \sim \rho, \; \tau \sim \widehat{\mathbb{P}}_t^{\pi_\star}(\cdot | s_1)} \left[ \mu(\mathbf{w}_\star^\top \phi(\tau)) + \sum_{h=1}^{H-1} \xi_{s_h, a_h}^{(t)} \right] \tag{53}
$$

(by the definition of $\xi_{s_h, a_h}^{(t)}$ in equation (7))

with probability at least $1 - N\delta$. Recall the definition of the event $\mathcal{E}_\delta$ from equation (5) and observe that it occurs with probability at least $1 - \delta$ by Lemma 3.1. Under event $\mathcal{E}_\delta$ for any $t \in \{t_0, \ldots, N\}$ and any $\tau \in \Gamma$

$$
\mu(\mathbf{w}_\star^\top \phi(\tau)) = \min \left\{ \mu(\mathbf{w}_\star^\top \phi(\tau)), 1 \right\} \leq \min \left\{ \mu(\widehat{\mathbf{w}}_t^\top \phi(\tau)) + \sqrt{\kappa} \beta_t(\delta) \| \phi(\tau) \|_{\mathbf{\Sigma}_t^{-1}}, 1 \right\}
$$

$$
\leq \min \left\{ \mu(\widehat{\mathbf{w}}_t^\top \phi(\tau)) + \sqrt{\kappa} \beta_t(\delta) \sum_{h=1}^{H} \| \phi_h(s_h, a_h) \|_{\mathbf{\Sigma}_t^{-1}}, 1 \right\}
$$

(by Lemma D.1)

$$
= \bar{\mu}_t^{\sf sd}(\widehat{\mathbf{w}}_t, \tau).
$$

Therefore by a union bound over $\mathcal{E}_\delta$ and the event where inequality (53) holds we infer that

$$
\sum_{t=t_0}^{N} (1 - b_t) V_\star \leq \sum_{t=t_0}^{N} (1 - b_t) \mathbb{E}_{s_1 \sim \rho, \; \tau \sim \widehat{\mathbb{P}}_t^{\pi_\star}(\cdot | s_1)} \left[ \bar{\mu}_t^{\sf sd}(\widehat{\mathbf{w}}_t, \tau) + \sum_{h=1}^{H-1} \xi_{s_h, a_h}^{(t)} \right] = \sum_{t=1}^{N} (1 - b_t) \widetilde{V}_\star^{(t), \sf sd},
$$

with probability at least $1 - (N + 1)\delta$.

**Event $\mathcal{E}_2^{\sf sd}$:** Assume that the event $\mathcal{E}_\delta$ occurs and also that for all $t \in \{t_0, \ldots, N\}$

$$
\frac{\lambda_{\max}(\mathbf{\Sigma}_t)}{\lambda_{\min}(\mathbf{\Sigma}_t)} \leq \Psi_t. \tag{54}
$$

The results of Lemma 3.1 and Lemma D.3 along with a union bound guarantee that this happens with probability at least $1 - 4\delta$.

Consider the following martingale difference sequence

$$
D_t := (1 - b_t) \left( \bar{V}^{(t), \sf sd} - V^{(t)} - \left[ \bar{\mu}_t^{\sf sd} \left( \widehat{\mathbf{w}}_t, \tau^{(t)} \right) - \mu \left( \mathbf{w}_\star^\top \phi(\tau^{(t)}) \right) \right] \right).
$$

Note that $|D_t| \leq 2$ since both $\bar{\mu}_t^{\sf sd}$ and $\mu$ take values between $0$ and $1$. Therefore, by applying Lemma A.1 we have that

$$
\sum_{t=t_0}^{N} (1 - b_t) \left( \bar{V}^{(t), \sf sd} - V^{(t)} \right)
$$

$$
\leq \sum_{t=t_0}^{N} (1 - b_t) \left( \bar{\mu}_t^{\sf sd} \left( \widehat{\mathbf{w}}_t, \tau^{(t)} \right) - \mu \left( \mathbf{w}_\star^\top \phi(\tau^{(t)}) \right) \right) + 4 \sqrt{N \log \left( \frac{6 \log(N)}{\delta} \right)} \tag{55}
$$

with probability at least $1 - \delta$. Let us now upper bound the sum in the RHS above

$$\sum_{t=t_0}^{N} (1 - b_t) \bar{\mu}_t^{\mathsf{sd}} \left( \widehat{\mathbf{w}}_t, \tau^{(t)} \right) - \mu \left( \mathbf{w}_\star^\top \phi(\tau^{(t)}) \right)$$

$$\overset{(i)}{=} \sum_{t=t_0}^{N} (1 - b_t) \left( \min \left\{ \mu \left( \widehat{\mathbf{w}}_t^\top \phi(\tau^{(t)}) \right) + \sqrt{\kappa} \beta_t(\delta) \sum_{h=1}^{H} \|\phi_h(s_h, a_h)\|_{\boldsymbol{\Sigma}_t^{-1}}, 1 \right\} - \min \left\{ \mu \left( \mathbf{w}_\star^\top \phi(\tau^{(t)}) \right), 1 \right\} \right)$$

$$\overset{(ii)}{\leq} \sum_{t=t_0}^{N} \left| \mu \left( \widehat{\mathbf{w}}_t^\top \phi(\tau^{(t)}) \right) + \sqrt{\kappa} \beta_t(\delta) \sum_{h=1}^{H} \|\phi_h(s_h, a_h)\|_{\boldsymbol{\Sigma}_t^{-1}} - \mu \left( \mathbf{w}_\star^\top \phi(\tau^{(t)}) \right) \right|$$

$$\overset{(iii)}{\leq} 2\sqrt{\kappa} \sum_{t=t_0}^{N} \beta_t(\delta) \left( \|\phi(\tau^{(t)})\|_{\boldsymbol{\Sigma}_t^{-1}} + \sum_{h=1}^{H} \|\phi_h(s_h, a_h)\|_{\boldsymbol{\Sigma}_t^{-1}} \right)$$

$$\overset{(iv)}{\leq} 2\sqrt{\kappa} \beta_N(\delta) \sum_{t=t_0}^{N} \left( \|\phi(\tau^{(t)})\|_{\boldsymbol{\Sigma}_t^{-1}} + \sum_{h=1}^{H} \|\phi_h(s_h, a_h)\|_{\boldsymbol{\Sigma}_t^{-1}} \right)$$

$$\overset{(v)}{\leq} 2\sqrt{\kappa} \beta_N(\delta) \sum_{t=t_0}^{N} \left( 1 + \sqrt{H \frac{\lambda_{\max}(\boldsymbol{\Sigma}_t)}{\lambda_{\min}(\boldsymbol{\Sigma}_t)}} \right) \|\phi(\tau^{(t)})\|_{\boldsymbol{\Sigma}_t^{-1}}$$

$$\overset{(vi)}{\leq} 2\sqrt{\kappa} \beta_N(\delta) \left( 1 + \sqrt{H \Psi_N} \right) \sum_{t=t_0}^{N} \|\phi(\tau^{(t)})\|_{\boldsymbol{\Sigma}_t^{-1}}$$

where $(i)$ follows by the definition of $\bar{\mu}_t^{\mathsf{sd}}$ and since $\mu$ is bounded between $0$ and $1$, $(ii)$ follows since for the function $z \mapsto \min\{z, 1\}$ is 1-Lipschitz and since $1 - b_t \in \{0, 1\}$, $(iii)$ follows since we have assumed that the event $\mathcal{E}_\delta$ occurs which provides the bound $|\mu \left( \widehat{\mathbf{w}}_t^\top \phi(\tau^{(t)}) - \mu \left( \mathbf{w}_\star^\top \phi(\tau^{(t)}) \right) \right)| \leq \sqrt{\kappa} \beta_t(\delta) \|\phi(\tau^{(t)})\|_{\boldsymbol{\Sigma}_t^{-1}}$, $(iv)$ follows since $\beta_t(\delta)$ is an increasing function of $t$, $(v)$ follows by invoking Lemma D.1 and finally $(vi)$ follows since we have assumed a bound on the condition number of $\boldsymbol{\Sigma}_t$ in inequality (54) and because $\Psi_N > \Psi_t$.

Continuing, since for any vector $\mathbf{z} \in \mathbb{R}^N$ $\|\mathbf{z}\|_1 \leq \sqrt{N} \|\mathbf{z}\|_2$, thus

$$\sum_{t=t_0}^{N} (1 - b_t) \left( \bar{\mu}_t^{\mathsf{sd}} \left( \widehat{\mathbf{w}}_t, \tau^{(t)} \right) - \mu \left( \mathbf{w}_\star^\top \phi(\tau^{(t)}) \right) \right)$$

$$\leq 2\sqrt{\kappa} \beta_N(\delta) \left( 1 + \sqrt{H \Psi_N} \right) \sqrt{N} \sqrt{\sum_{t=1}^{N} \|\phi(\tau^{(t)})\|_{\boldsymbol{\Sigma}_t^{-1}}^2}$$

$$\leq \beta_N(\delta) \left( 1 + \sqrt{H \Psi_N} \right) \sqrt{8Nd \max\{\kappa, 1\} \log \left( 1 + \frac{N}{\kappa d} \right)}$$

where the final inequality follows by invoking the determinant lemma (Lemma A.3) from above.

A union bound over the event $\mathcal{E}_\delta$, the event where the condition number of $\boldsymbol{\Sigma}_t$ is bounded and the event where inequality (55) holds proves that this bound holds with probability at least $1 - 5\delta$.

**Event $\mathcal{E}_3^{\mathsf{sd}}$:** By mirroring the proof on the bound on the probability of the event $\mathcal{E}_3$ in Lemma B.5 we can show that $\mathbb{P}\left[ \mathcal{E}_3^{\mathsf{sd}} \right] \geq 1 - (N+1)\delta$.

**Event $\mathcal{E}_4^{\mathsf{sd}}$:** On applying Theorem A.2 with the martingale difference sequence $b_t - 1/t^{1/3}$ we know that with probability at least $1 - \delta$:

$$\sum_{t=1}^{N} b_t \leq 4N^{2/3}$$

if $N \geq \left( \frac{20}{3} \log \left( \frac{1}{\delta} \right) \right)^{3/2}$. Thus, with probability at least $1 - \delta$

$$\sum_{t=1}^{N} b_t \leq \left( \frac{20}{3} \log \left( \frac{1}{\delta} \right) \right)^{3/2} + 4N^{2/3}.$$

In other words $\mathbb{P}[\mathcal{E}_4^{\mathsf{sd}}] \geq 1 - \delta$.

**Event $\mathcal{E}_5^{\mathsf{sd}}$:** By invoking Lemma 3.5 it immediately follows that $\mathbb{P}[\mathcal{E}_5^{\mathsf{sd}}] \geq 1 - 2\delta$.

**Union bound over the five events:** A union bound over the five events shows that

$$\mathbb{P}\left[ \mathcal{E}_{\mathsf{good}}^{\mathsf{sd}} \right] \geq 1 - \mathbb{P}[(\mathcal{E}_1^{\mathsf{sd}})^c] - \mathbb{P}[(\mathcal{E}_2^{\mathsf{sd}})^c] - \mathbb{P}[(\mathcal{E}_3^{\mathsf{sd}})^c] - \mathbb{P}[(\mathcal{E}_4^{\mathsf{sd}})^c] - \mathbb{P}[(\mathcal{E}_5^{\mathsf{sd}})^c]$$
$$\geq 1 - (2N + 10)\delta \geq 1 - 12N\delta,$$

which completes the proof.

∎

### D.4 Proof of Theorem 3.6

Recall the statement of the theorem.

**Theorem 3.6.** *For any $\bar{\delta} \in (0, 1]$, set $\delta = \bar{\delta}/(12N)$. Under Assumptions 2.1, 2.2, 3.3 and 3.4, and for all $N > \bar{N}_{\mathsf{exp}}$ (see its definition in Lemma 3.5) if Algorithm 3 is run with the parameters $N_{\mathsf{EUL}}$ and $N_{\mathsf{EVAL}}$ set as specified in Lemma 3.5 then its regret is upper bounded as follows:*

$$\mathcal{R}(N) \leq \widetilde{O}\left( \frac{\sqrt{\kappa H} d}{\omega} (d^3 + B^{3/2}) N^{2/3} + \left[ H\sqrt{(H + |\mathcal{S}|)|\mathcal{S}||\mathcal{A}|} + H^2 \right] \sqrt{N} \right.$$
$$\left. + (H + |\mathcal{S}|) H |\mathcal{S}||\mathcal{A}| + \frac{d^2}{\omega^2} \left( \frac{d^2}{\omega^2} + |\mathcal{S}|^2 |\mathcal{A}| H^2 \right) \right),$$

*with probability at least $1 - \bar{\delta}$.*

**Proof** Let us assume that the event $\mathcal{E}_{\mathsf{good}}^{\mathsf{sd}}$ defined in Lemma D.6 occurs. By Lemma D.6 we know that $\mathbb{P}\left[ \mathcal{E}_{\mathsf{good}} \right] \geq 1 - 12N\delta$. First we decompose the regret as follows:

$$\mathcal{R}(N) = \sum_{t=1}^{N} V_\star - V^{(t)}$$
$$= \sum_{t=1}^{t_0} V_\star - V^{(t)} + \sum_{t=t_0+1}^{N} V_\star - V^{(t)}$$
$$= \sum_{t=1}^{t_0} V_\star - V^{(t)} + \sum_{t=t_0+1}^{N} b_t(V_\star - V^{(t)}) + \sum_{t=t_0+1}^{N} (1 - b_t)(V_\star - V^{(t)}).$$

Now since $V_\star - V^{(t)}$ is bounded between 0 and 1 we know that

$$\mathcal{R}(N) \leq t_0 + \sum_{t=t_0+1}^{N} b_t + \sum_{t=t_0+1}^{N} (1-b_t)(V_\star - V^{(t)})$$

$$\overset{(i)}{\leq} t_0 + \left(\frac{20}{3}\log\left(\frac{1}{\delta}\right)\right)^{3/2} + 4N^{2/3} + \sum_{t=t_0+1}^{N}(1-b_t)(V_\star - V^{(t)})$$

$$\overset{(ii)}{\leq} C_3 \left[\frac{d^2\log^2(d\log(1+\frac{16N}{d\omega^2}))}{\omega^4}\sqrt{\log(N/\delta)} + N_{\mathsf{exp}}^{2/3}\right]^{3/2} + \left(\frac{20}{3}\log\left(\frac{1}{\delta}\right)\right)^{3/2}$$

$$+ 4N^{2/3} + \sum_{t=t_0+1}^{N}(1-b_t)(V_\star - V^{(t)})$$

$$\overset{(iii)}{\leq} C_3 \left[\frac{d^2\log^2(d\log(1+\frac{16N}{d\omega^2}))}{\omega^4}\sqrt{\log(N/\delta)} + \left(\frac{d\log\left(1+\frac{16N}{d\omega^2}\right)}{\log(3/2)}(N_{\mathsf{EUL}} + N_{\mathsf{EVAL}})\right)^{2/3}\right]^{3/2}$$

$$+ \left(\frac{20}{3}\log\left(\frac{1}{\delta}\right)\right)^{3/2} + 4N^{2/3}$$

$$+ \sum_{t=t_0+1}^{N}(1-b_t)(V_\star - V^{(t)})$$

$$\tag{56}$$

where $(i)$ follows by the definition of the event $\mathcal{E}_4^{\mathsf{sd}}$, $(ii)$ is by the definition of $t_0$ in equation (46a) and $(iii)$ follows by the definition of $\mathcal{E}_5^{\mathsf{sd}}$ that bounds $N_{\mathsf{EXP}}$. It remains to bound the last term in the RHS above. Going forward let us assume that $N \geq t_0 + 1$, else we are done. To bound this term note that by the definition of the event $\mathcal{E}_1^{\mathsf{sd}}$ we know that

$$\sum_{t=t_0+1}^{N}(1-b_t)(V_\star - V^{(t)}) \leq \sum_{t=t_0+1}^{N}(1-b_t)\left(\widetilde{V}_\star^{(t),\mathsf{sd}} - V^{(t)}\right).$$

By the definition of the policy $\pi^{(t)}$ (see equation (14)) we have that

$$\widetilde{V}_\star^{(t),\mathsf{sd}} = \mathbb{E}_{s_1\sim\rho,\ \tau\sim\widehat{\mathbb{P}}_t^{\pi_\star}(\cdot|s_1)}\left[\widetilde{\mu}_t^{\mathsf{sd}}(\widehat{\mathbf{w}}_t,\tau)\right] \leq \mathbb{E}_{s_1\sim\rho,\ \tau\sim\widehat{\mathbb{P}}_t^{\pi^{(t)}}(\cdot|s_1)}\left[\widetilde{\mu}_t^{\mathsf{sd}}(\widehat{\mathbf{w}}_t,\tau)\right] = \widetilde{V}^{(t),\mathsf{sd}}.$$

Thus,

$$\sum_{t=t_0+1}^{N}(1-b_t)(V_\star - V^{(t)}) \leq \sum_{t=t_0+1}^{N}(1-b_t)\left(\widetilde{V}^{(t),\mathsf{sd}} - V^{(t)}\right).$$

Under event $\mathcal{E}_2^{\mathsf{sd}}$ we know that

$$\sum_{t=t_0+1}^{N}(1-b_t)\left(\bar{V}^{(t),\mathsf{sd}} - V^{(t)}\right)$$

$$\leq \beta_N(\delta)\left(1+\sqrt{H\Psi_N}\right)\sqrt{8Nd\max\{\kappa,1\}\log\left(1+\frac{N}{\kappa d}\right)} + 4\sqrt{N\log\left(\frac{6\log(N)}{\delta}\right)}.$$

By combining the previous two inequalities we find that

$$\sum_{t=t_0+1}^{N}(1-b_t)(V_\star - V^{(t)})$$

$$\leq \sum_{t=t_0+1}^{N}(1-b_t)\left(\widetilde{V}^{(t),\mathsf{sd}} - \bar{V}^{(t),\mathsf{sd}}\right)$$

$$+ \beta_N(\delta)\left(1+\sqrt{H\Psi_N}\right)\sqrt{8Nd\max\{\kappa,1\}\log\left(1+\frac{N}{\kappa d}\right)} + 4\sqrt{N\log\left(\frac{6\log(N)}{\delta}\right)}.$$

Finally under event $\mathcal{E}_3^{\mathsf{sd}}$ we have a bound on the first term on the right hand side above, this leads to the bound

$$\sum_{t=t_0+1}^{N} (1-b_t)(V_\star - V^{(t)})$$

$$\leq (2H+1) \sum_{t=t_0+1}^{N} \sum_{h=1}^{H-1} \xi_{s_h^{(t)},a_h^{(t)}}^{(t)} + 4H^2 \sqrt{N \log\left(\frac{6\log(N)}{\delta}\right)}$$

$$+ \beta_N(\delta)\left(1+\sqrt{H\Psi_N}\right)\sqrt{8Nd \max\{\kappa,1\}\log\left(1+\frac{N}{\kappa d}\right)} + 4\sqrt{N \log\left(\frac{6\log(N)}{\delta}\right)} + 1. \quad (57)$$

It remains to bound the term $\sum_{t=t_0+1}^{N}\sum_{h=1}^{H-1}\xi_{s_h^{(t)},a_h^{(t)}}^{(t)}$. By mirroring the logic used to arrive at inequality (40) we can show that

$$\sum_{t=t_0+1}^{N} \sum_{h=1}^{H-1} \xi_{s_h^{(t)},a_h^{(t)}}^{(t)} \leq 8|\mathcal{S}||\mathcal{A}|\log\left(\frac{6(|\mathcal{S}||\mathcal{A}|H)^H(8H^2N)^{|\mathcal{S}|}\log(N)}{\delta}\right)$$

$$+ 8\sqrt{\log\left(\frac{6(|\mathcal{S}||\mathcal{A}|H)^H(8NH^2)^{|\mathcal{S}|}\log(N))}{\delta}\right)|\mathcal{S}||\mathcal{A}|N}.$$

Plugging this upper bound into inequality (57) we get

$$\sum_{t=t_0+1}^{N} (1-b_t)(V_\star - V^{(t)})$$

$$\leq 8(2H+1)|\mathcal{S}||\mathcal{A}| \cdot \log\left(\frac{6(|\mathcal{S}||\mathcal{A}|H)^H(8H^2N)^{|\mathcal{S}|}\log(N)}{\delta}\right)$$

$$+ 8(2H+1)\sqrt{\log\left(\frac{6(|\mathcal{S}||\mathcal{A}|H)^H(8NH^2)^{|\mathcal{S}|}\log(N))}{\delta}\right)|\mathcal{S}||\mathcal{A}|N}$$

$$+ 4H^2\sqrt{N\log\left(\frac{6\log(N)}{\delta}\right)} + \beta_N(\delta)\left(1+\sqrt{H\Psi_N}\right)\sqrt{8Nd\max\{\kappa,1\}\log\left(1+\frac{N}{\kappa d}\right)}$$

$$+ 4\sqrt{N\log\left(\frac{6\log(N)}{\delta}\right)} + 1.$$

Now finally, by using this upper bound in inequality (56) we find that

$$\mathcal{R}(N)$$

$$\leq C_3 \left[ \frac{d^2 \log^2(d\log(1+\frac{16N}{d\omega^2}))}{\omega^4} \sqrt{\log(N/\delta)} + \left( \frac{d\log\left(1+\frac{16N}{d\omega^2}\right)}{\log(3/2)} (N_{\mathsf{EUL}} + N_{\mathsf{EVAL}}) \right)^{2/3} \right]^{3/2}$$

$$+ \left( \frac{20}{3} \log\left(\frac{1}{\delta}\right) \right)^{3/2} + 4N^{2/3}$$

$$+ 8(2H+1)|\mathcal{S}||\mathcal{A}| \cdot \log\left( \frac{6(|\mathcal{S}||\mathcal{A}|H)^H(8H^2N)^{|\mathcal{S}|}\log(N)}{\delta} \right)$$

$$+ 8(2H+1)\sqrt{\log\left( \frac{6(|\mathcal{S}||\mathcal{A}|H)^H(8NH^2)^{|\mathcal{S}|}\log(N))}{\delta} \right) |\mathcal{S}||\mathcal{A}|N}$$

$$+ 4H^2 \sqrt{N\log\left(\frac{6\log(N)}{\delta}\right)} + \beta_N(\delta)\left(1+\sqrt{H\Psi_N}\right)\sqrt{8Nd\max\{\kappa,1\}\log\left(1+\frac{N}{\kappa d}\right)}$$

$$+ 4\sqrt{N\log\left(\frac{6\log(N)}{\delta}\right)} + 1 \tag{58}$$

$$= \widetilde{O}\left( \frac{\sqrt{\kappa H}d}{\omega}(d^3+B^{3/2})N^{2/3} + \left[H\sqrt{(H+|\mathcal{S}|)|\mathcal{S}||\mathcal{A}|} + H^2\right]\sqrt{N} \right.$$

$$\left. + (H+|\mathcal{S}|)H|\mathcal{S}||\mathcal{A}| + \frac{d^2}{\omega^2}\left(\frac{d^2}{\omega^2}+|\mathcal{S}|^2|\mathcal{A}|H^2\right) \right)$$

where the last equality follows since by their definitions

$$N_{\mathsf{EUL}} = \widetilde{\Theta}\left(\frac{|\mathcal{S}|^2|\mathcal{A}|H^2}{\omega^2}\right); \qquad N_{\mathsf{EVAL}} = \widetilde{\Theta}\left(\frac{d^3}{\omega^4}\right);$$

$$\beta_N(\delta) = \widetilde{O}\left(d^3+B^{3/2}\right); \qquad \Psi_N = \widetilde{O}\left(\frac{dN^{1/3}}{\omega^2}\right),$$

and by simplifying the expression in equation (58). This bound holds with probability $1 - 12N\delta$. Recalling that $\bar{\delta} = 12N\delta$ completes our proof. ∎

# E   A Dynamic Programming Approach to Approximate $\pi^{(t)}$

In this section we present a computationally efficient dynamic programming algorithm that can be used to approximate the policy $\pi^{(t)}$ that is defined in equation (14) in Algorithm 3. We will also provide a proof for Proposition 3.7.

To avoid clashes of notation with the other sections of the paper we denote policies using $\theta$ here. We assume that we are given a transition dynamics model $\bar{\mathbb{P}}$, a vector $\mathbf{w} \in \mathbb{R}^d$, feature maps $\{\phi_h\}_{h\in[H]}$, a positive semi-definite matrix $\mathbf{\Sigma}$ and a bonus function $b_h : \mathcal{S}\times\mathcal{A} \to \mathbb{R}$ for every $h \in [H]$. Also assume that there exists $\zeta > 0$ such that $\mathbf{w}^\top\phi(\tau) \in [-\zeta,\zeta]$, $\sum_h\|\phi_h(s_h,a_h)\|_{\mathbf{\Sigma}^{-1}} \in [0,\zeta]$ and $\sum_h b_h(s_h,a_h) \in [0,\zeta]$ for all $\tau \in \Gamma$. Finally let $w_h(s,a) := \mathbf{w}^\top\phi_h(s,a)$ and $v_h(s,a) := \|\phi_h(s,a)\|_{\mathbf{\Sigma}^{-1}}$.

Given a policy $\theta$ and an initial state $s_1$ define an optimistic value-function with this vector $\mathbf{w}$, feature maps, positive semi-definite matrix $\Sigma$ and bonuses $\{b_h\}_{h\in[H]}$ as

$$\bar{V}_{\text{opt}}^\theta$$

$$:= \mathbb{E}_{s_1\sim\rho,\tau\sim\bar{\mathbb{P}}^\theta(\cdot|s_1)}\left[\min\left\{\mu\left(\mathbf{w}^\top\phi(\tau)\right) + \sum_{h=1}^H \|\phi_h(s_h,a_h)\|_{\Sigma^{-1}}, 1\right\} + \sum_{h=1}^H b_h(s_h,a_h)\right]$$

$$= \mathbb{E}_{s_1\sim\rho,\ \tau\sim\bar{\mathbb{P}}^\theta(\cdot|s_1)}\left[\min\left\{\mu\left(\sum_{h=1}^H w_h(s_h,a_h)\right) + \sum_{h=1}^H v_h(s_h,a_h), 1\right\} + \sum_{h=1}^H b_h(s_h,a_h)\right].$$

Define the optimal policy with respect to this optimistic value function:

$$\theta_\star \in \arg\max_{\theta\in\Pi} \bar{V}_{\text{opt}}^\theta.$$

Our goal is to find an $\varepsilon$-optimal policy $\widehat{\theta} = (\widehat{\theta}_1,\ldots,\widehat{\theta}_H)$ that satisfies

$$\bar{V}_{\text{opt}}^{\theta_\star} - \bar{V}_{\text{opt}}^{\widehat{\theta}} \le \varepsilon.$$

Also define the conditional optimistic value-function at any step $h \in [H]$:

$$\bar{V}_h^\theta(s,\tau'_{h-1}) := \mathbb{E}_{\tau\sim\bar{\mathbb{P}}^\theta}\left[\min\left\{\mu\left(\sum_{\ell=1}^H w_\ell(s_\ell,a_\ell)\right) + \sum_{\ell=1}^H v_\ell(s_\ell,a_\ell), 1\right\}\right.$$
$$\left. + \sum_{\ell=1}^H b_\ell(s_\ell,a_\ell) \,\middle|\, s_h = s, \tau_{h-1} = \tau'_{h-1}\right]. \quad (59)$$

Define $m := \left\lceil \frac{\zeta-(-\zeta)}{\varepsilon/(6H^2)} \right\rceil = \left\lceil \frac{12H^2\zeta}{\varepsilon} \right\rceil$ intervals

$$\psi_j := \left[-\zeta + \frac{(j-1)\varepsilon}{6H^2}, -\zeta + \frac{j\varepsilon}{6H^2}\right), \quad \text{if } j \in \{1,\ldots,m-1\}$$

$$\text{and,} \quad \psi_m := \left[-\zeta + \frac{(m-1)\varepsilon}{6H^2}, \zeta\right].$$

The centers of these intervals are $\nu_j := -\zeta + \frac{(j-\frac{1}{2})\varepsilon}{6H^2}$ for every $j \in [m]$. Define a map $\sigma : [-\zeta,\zeta] \to \{1,\ldots,m\}$ that maps each $x$ to the index of interval that $x$ lies in,

$$\sigma(x) = j, \quad \text{if } x \in \psi_j.$$

Our dynamic programming approach will require us to define tensors $\widehat{a}_h$ and $\widehat{V}_h$ for every $h \in [H]$. Given any quartet $(s,i,j,k) \in \mathcal{S} \times [m] \times [m] \times [m]$ define the following at the final step $H$

$$\widehat{a}_H(s,i,j,k) \in \arg\max_{a\in\mathcal{A}}\left\{\min\left\{\mu\left(\nu_i + w_H(s,a)\right) + \nu_j + v_H(s,a), 1\right\} + \nu_k + b_H(s,a)\right\};$$

$$\widehat{V}_H(s,i,j,k) := \max_{a\in\mathcal{A}}\left\{\min\left\{\mu\left(\nu_i + w_H(s,a)\right) + \nu_j + v_H(s,a), 1\right\} + \nu_k + b_H(s,a)\right\}.$$

The action $\widehat{a}_H(s,i,j,k)$ is the optimal action when the state is $s$ and the "histories" $\sum_{h=1}^{H-1} w_h(s_h,a_h), \sum_{h=1}^{H-1} v_h(s_h,a_h)$ and $\sum_{h=1}^{H-1} b_h(s_h,a_h)$ are equal to $\nu_i, \nu_j$ and $\nu_k$ respectively. Further, the tensor $\widehat{V}_H(s,i,j,k)$ stores the value of the conditional value function when this optimal action is taken given this quartet. Also recursively define the following in the preceding steps:

$$\widehat{a}_h(s,i,j,k) \in \arg\max_{a\in\mathcal{A}} \mathbb{E}_{s'\sim\mathbb{P}(\cdot|s,a)}\left[\widehat{V}_{h+1}(s', \sigma(w_h(s,a)+\nu_i), \sigma(v_h(s,a)+\nu_j), \sigma(b_h(s,a)+\nu_k))\right];$$

$$\widehat{V}_h(s,i,j,k) := \max_{a\in\mathcal{A}} \mathbb{E}_{s'\sim\mathbb{P}(\cdot|s,a)}\left[\widehat{V}_{h+1}(s', \sigma(w_h(s,a)+\nu_i), \sigma(v_h(s,a)+\nu_j), \sigma(b_h(s,a)+\nu_k))\right].$$

At the initial step $h = 1$ the expectation over the states $s' \sim \bar{\mathbb{P}}(\cdot|s,a)$ in the definition above is replaced by the expectation over the initial state $s_1 \sim \rho$.

To construct $\widehat{\theta}$, our strategy will be to use these near-optimal actions $(\widehat{a}_h)$ at every step over this "$\frac{\varepsilon}{6H^2}$-net" of representative histories that is defined. Then given any state and sub-trajectory we will map this state and sub-trajectory to its nearest neighbor in the net of histories and play the near-optimal action corresponding to this neighbor. To this end define the maps

$$i_h(\tau_{h-1}) := \sigma\left(\sum_{\ell=1}^{h-1} w_\ell(s_\ell, a_\ell)\right), \tag{60a}$$

$$j_h(\tau_{h-1}) := \sigma\left(\sum_{\ell=1}^{h-1} v_\ell(s_\ell, a_\ell)\right), \quad \text{and} \tag{60b}$$

$$k_h(\tau_{h-1}) := \sigma\left(\sum_{\ell=1}^{h-1} b_\ell(s_\ell, a_\ell)\right). \tag{60c}$$

At times we will use $i_h$, $j_h$ and $k_h$ as shorthand for $i_h(\tau_{h-1})$, $j_h(\tau_{h-1})$ and $k_h(\tau_{h-1})$ respectively. Given a state $s$ and sub-trajectory $\tau_{h-1}$ the policy at step $h \in [H]$, $\widehat{\theta}_h(\cdot|s, \tau_{h-1})$ puts all of its mass on the action

$$\widehat{a}_h\left(s_h, i_h(\tau_{h-1}), j_h(\tau_{h-1}), k_h(\tau_{h-1})\right)$$

(where we break ties among actions arbitrarily). Given a policy $\theta$, let $\theta_{h:H} = (\theta_h, \ldots, \theta_H)$ denote the set of policies from step $h$ onward. Let $\bar{P}^{\theta_{h:H}}(\cdot|s)$ denote the distribution of the trajectory in the steps $h, \ldots, H$ given that the state at step $h-1$ was $s$. Finally define the extended conditional value-functions for the policy $\widehat{\theta}$ to be

$$\check{V}_h^{\widehat{\theta}_{h+1:H}}(s, \alpha, \beta, \gamma) :=$$

$$\mathbb{E}_{\tau \sim \bar{\mathbb{P}}^{\widehat{\theta}_{h+1:H}}(\cdot|s)} \left[ \min\left\{ \mu\left( \alpha + w_h(s, \widehat{a}_h(s, \sigma(\alpha), \sigma(\beta), \sigma(\gamma))) + \sum_{\ell=h+1}^{H} w_\ell(s_\ell, a_\ell) \right) \right.\right.$$

$$\left. + \beta + v_h(s, \widehat{a}_h(s, \sigma(\alpha), \sigma(\beta), \sigma(\gamma))) + \sum_{\ell=h+1}^{H} v_\ell(s_\ell, a_\ell), 1 \right\}$$

$$\left. + \gamma + b_h(s, \widehat{a}_h(s, \sigma(\alpha), \sigma(\beta), \sigma(\gamma))) + \sum_{\ell=h+1}^{H} b_\ell(s_\ell, a_\ell) \right]$$

for any $h \in [H]$, $s \in \mathcal{S}$, $\alpha \in [-\zeta, \zeta]$, $\beta \in [0, \zeta]$ and $\gamma \in [0, \zeta]$. In the definition above the expectation is over the steps $h+1, \ldots, H$. The extended value function is the definition of the conditional value function by using the summary of the history: $\alpha, \beta$ and $\gamma$.

### E.1 The Policy $\widehat{\theta}$ is $\varepsilon$-Optimal

The following lemma shows that the policy $\widehat{\theta}$ is $\varepsilon$-optimal and can be found efficiently. We shall use this lemma to prove Proposition 3.7 below.

**Lemma E.1.** *The policy $\widehat{\theta}$ satisfies*

$$\bar{V}_{\text{opt}}^{\theta_\star} - \bar{V}_{\text{opt}}^{\widehat{\theta}} \leq \varepsilon.$$

*Furthermore the policy $\widehat{\theta}$ can be found in* $\text{poly}\left(|\mathcal{S}|, |\mathcal{A}|, H, \zeta, \frac{1}{\varepsilon}\right)$ *time and memory.*

**Proof** The proof shall proceed in two steps. First, we shall show via an inductive argument that certain properties are satisfied at all steps. In the second part we will use these properties to prove the lemma.

**Part I: The inductive hypothesis.** The induction will be over the steps $H, \ldots, 1$. We shall inductively show that:

(a) For any $s \in \mathcal{S}$, $\alpha \in [-\zeta, \zeta]$, $\beta \in [0, \zeta]$ and $\gamma \in [0, \zeta]$:

$$\left| \check{V}_h^{\widehat{\theta}_{h:H}}(s, \alpha, \beta, \gamma) - \widehat{V}_h(s, \sigma(\alpha), \sigma(\beta), \sigma(\gamma)) \right| \leq \frac{(H+1-h)\varepsilon}{2H^2};$$

(b) for any $s \in \mathcal{S}$ and $\tau_{h-1} \in \Gamma_{h-1}$

$$\max_{a \in \mathcal{A}} \mathbb{E}_{s' \sim \bar{\mathbb{P}}(\cdot|s,a)} \left[ \bar{V}_{h+1}^{\widehat{\theta}_{h+1:H}}(s', \{s, a, \tau_{h-1}\}) \right] - \bar{V}_h^{\widehat{\theta}_{h:H}}(s, \tau_{h-1}) \leq \frac{(H+1-h)\varepsilon}{H^2};$$

(c) given the tensor $\widehat{V}_{h+1}$ it is possible to find $\widehat{a}_h(s, i, j, k)$ and $\widehat{V}_h(s, i, j, k)$ for all quartets using poly $\left(|\mathcal{S}|, |\mathcal{A}|, H, \zeta, \frac{1}{\varepsilon}\right)$ time and memory.

Note that $\max_{a \in \mathcal{A}} \mathbb{E}_{s' \sim \bar{\mathbb{P}}(\cdot|s,a)} \left[ \bar{V}_{h+1}^{\widehat{\theta}_{h+1:H}}(s', \{s, a, \tau_{h-1}\}) \right]$ corresponds to the conditional-value (see the Definition of $\bar{V}$ in equation (59)) of taking the best action at step $h$ when the policy for the future steps is $\widehat{\theta}_{h+1:H}$.

**Base case:** The base case of the induction is at step $H$.

*Part (a):* Fix an $\alpha, \beta$ and $\gamma$ and define the shorthand $\widehat{a}_H := \widehat{a}_H(s, \sigma(\alpha), \sigma(\beta), \sigma(\gamma))$. By the definition of $\check{V}_H, \widehat{V}_H$ and the policy $\widehat{\theta}$ we have

$$\left| \check{V}_H^{\widehat{\theta}_H}(s, \alpha, \beta, \gamma) - \widehat{V}_H(s, \sigma(\alpha), \sigma(\beta), \sigma(\gamma)) \right|$$

$$= \left| \min \left\{ \mu \left( \alpha + w_H(s, \widehat{a}_h) \right) + \beta + v_H(s, \widehat{a}_H), 1 \right\} + \gamma + b_H(s, \widehat{a}_H) \right.$$

$$\left. - \min \left\{ \mu \left( \nu_{\sigma(\alpha)} + w_H(s, \widehat{a}_H) \right) + \nu_{\sigma(\beta)} + v_H(s, \widehat{a}_H), 1 \right\} + \nu_{\sigma(\gamma)} + b_H(s, \widehat{a}_H) \right|$$

$$\overset{(i)}{\leq} \left| \mu \left( \alpha + w_H(s, \widehat{a}_h) \right) - \mu \left( \nu_{\sigma(\alpha)} + w_H(s, \widehat{a}_H) \right) \right| + |\beta - \nu_{\sigma(\beta)}| + |\gamma - \nu_{\sigma(\gamma)}|$$

$$\overset{(ii)}{\leq} |\alpha - \nu_{\sigma(\alpha)}| + |\beta - \nu_{\sigma(\beta)}| + |\gamma - \nu_{\sigma(\gamma)}| \overset{(iii)}{\leq} 3 \times \frac{\varepsilon}{6H^2} = \frac{\varepsilon}{2H^2}$$

where $(i)$ follows since the function $z \mapsto \min(z, 1)$ is 1-Lipschitz and by the triangle inequality, and $(ii)$ follows since $\mu$ is 1-Lipschitz, and $(iii)$ follows by the definition of the function $\sigma$, that projects a number onto a grid with granularity $\varepsilon/(6H^2)$.

*Part (b):* An episode terminates at the end of step $H$, therefore we define $\bar{V}_{H+1}(s', \tau_H) := \min \left\{ \mu \left( \sum_{h=1}^H w_h(s_h, a_h) \right) + \sum_{h=1}^H v_h(s_h, a_h), 1 \right\} + \sum_{h=1}^H b_h(s_h, a_h)$. Thus, by the definition of the extended conditional value function $\check{V}_H$

$$\max_{a \in \mathcal{A}} \mathbb{E}_{s' \sim \bar{\mathbb{P}}(\cdot|s,a)} \left[ \bar{V}_{H+1}(s', \{s, a, \tau_{H-1}\}) \right] - \bar{V}_H^{\widehat{\theta}_H}(s, \tau_{H-1})$$

$$= \max_{a \in \mathcal{A}} \mathbb{E}_{s' \sim \bar{\mathbb{P}}(\cdot|s,a)} \left[ \bar{V}_{H+1}(s', \{s, a, \tau_{H-1}\}) \right] - \check{V}_H^{\widehat{\theta}_H} \left( s, \sum_{\ell=1}^{H-1} w_\ell(s_\ell, a_\ell), \sum_{\ell=1}^{H-1} v_\ell(s_\ell, a_\ell), \sum_{\ell=1}^{H-1} b_\ell(s_\ell, a_\ell) \right)$$

$$\overset{(i)}{=} \max_{a \in \mathcal{A}} \mathbb{E}_{s' \sim \bar{\mathbb{P}}(\cdot|s,a)} \left[ \bar{V}_{H+1}(s', \{s, a, \tau_{H-1}\}) \right] - \widehat{V}_H^{\widehat{\theta}_H}(s, i_H, j_H, k_H)$$

$$+ \widehat{V}_H^{\widehat{\theta}_H}(s, i_H, j_H, k_H) - \check{V}_H^{\widehat{\theta}_H} \left( s, \sum_{\ell=1}^{H-1} w_\ell(s_\ell, a_\ell), \sum_{\ell=1}^{H-1} v_\ell(s_\ell, a_\ell), \sum_{\ell=1}^{H-1} b_\ell(s_\ell, a_\ell) \right)$$

$$\overset{(ii)}{\leq} \max_{a \in \mathcal{A}} \mathbb{E}_{s' \sim \bar{\mathbb{P}}(\cdot|s,a)} \left[ \bar{V}_{H+1}(s', \{s, a, \tau_{H-1}\}) \right] - \widehat{V}_H^{\widehat{\theta}_H}(s, i_H, j_H, k_H) + \frac{\varepsilon}{2H^2}$$

where in $(i)$ recall the definitions of $i_H, j_H$ and $k_H$ from above in equations (60a)-(60c), and $(ii)$ follows by Part $(a)$ of the induction hypothesis. Continuing

$$\max_{a \in \mathcal{A}} \mathbb{E}_{s' \sim \bar{\mathbb{P}}(\cdot | s, a)} \left[ \bar{V}_{H+1}(s', \{s, a, \tau_{H-1}\}) \right] - \bar{V}_H^{\hat{\theta}_H}(s, \tau_{H-1})$$

$$\overset{(i)}{\leq} \max_{a \in \mathcal{A}} \left\{ \min \left\{ \mu \left( \sum_{h=1}^{H-1} w_h(s_h, a_h) + w_H(s, a) \right) + \sum_{h=1}^{H-1} v_h(s_h, a_h) + v_H(s, a), 1 \right\} \right.$$

$$\left. + \sum_{h=1}^{H-1} b_h(s_h, a_h) + b_H(s, a) \right\}$$

$$- \max_{a' \in \mathcal{A}} \left\{ \min \left\{ \mu \left( \nu_{i_H} + w_H(s, a') + \nu_{j_H} + v_H(s, a') \right), 1 \right\} + \nu_{k_H} + b_H(s, a') \right\} + \frac{\varepsilon}{2H^2}$$

$$\leq \max_{a \in \mathcal{A}} \left\{ \min \left\{ \mu \left( w_H(s, a) + \sum_{h=1}^{H-1} w_h(s_h, a_h) \right) + \sum_{h=1}^{H-1} v_h(s_h, a_h) + v_H(s, a), 1 \right\} \right.$$

$$+ \sum_{h=1}^{H-1} b_h(s_h, a_h) + b_H(s, a)$$

$$\left. - \min \left\{ \mu \left( w_H(s, a) + \nu_{i_H} \right) + v_H(s, a) + \nu_{j_H}, 1 \right\} - \nu_{k_H} - b_H(s, a) \right\} + \frac{\varepsilon}{2H^2}$$

$$\overset{(ii)}{\leq} \left| \sum_{h=1}^{H-1} w_h(s_h, a_h) - \nu_{i_H} \right| + \left| \sum_{h=1}^{H-1} v_h(s_h, a_h) - \nu_{j_H} \right| + \left| \sum_{h=1}^{H-1} b_h(s_h, a_h) - \nu_{k_H} \right| + \frac{\varepsilon}{2H^2} \overset{(iii)}{\leq} \frac{\varepsilon}{H^2}.$$

where $(i)$ follows by the definition of $\widehat{V}_H(s, i_H, j_H, k_H)$, $(ii)$ follows because the functions $z \mapsto \min\{z, 1\}$ and $z \mapsto \frac{1}{1+\exp(-z)}$ are both 1-Lipschitz, and $(iii)$ follows by the definition of the maps $i_H, j_H$ and $k_H$, and the intervals $\psi_j$. This proves the second part of the inductive hypothesis in the base case.

*Part (c):* Let's show $\widehat{a}_H(s, i, j, k)$ and $\widehat{V}_H(s, i, j, k)$ can be computed efficiently. Fix a quartet $(s, i, j, k) \in \mathcal{S} \times [m] \times [m] \times [m]$. Then the values

$$\widehat{a}_H(s, i, j, k) \in \underset{a \in \mathcal{A}}{\arg\max} \left\{ \min \left\{ \mu \left( \nu_i + w_H(s, a) \right) + \nu_j + v_H(s, a), 1 \right\} + \nu_k + b_H(s, a) \right\}$$

$$\widehat{V}_H(s, i, j, k) = \max_{a \in \mathcal{A}} \left\{ \min \left\{ \mu \left( \nu_i + w_H(s, a) \right) + \nu_j + v_H(s, a), 1 \right\} + \nu_k + b_H(s, a) \right\}$$

can be found using $\mathsf{poly}(|\mathcal{A}|)$ time and memory. Therefore, the entire tensor can be found using $|\mathcal{S}|m^3 \times \mathsf{poly}(|\mathcal{A}|) = \mathsf{poly}(|\mathcal{S}|, |\mathcal{A}|, H, \zeta, \frac{1}{\varepsilon})$ time and memory.

**Induction step:** Assume that the induction hypothesis holds at the steps $H, \ldots, h+1$. We will now prove that each part of the induction hypothesis also holds at the step $h \geq 1$.

*Part (a):* Fix an $\alpha, \beta$ and $\gamma$ and let's define the shorthand $\widehat{a}_h = \widehat{a}_h(s, \sigma(\alpha), \sigma(\beta), \sigma(\gamma))$. Hence[2],

$$\left| \check{V}_h^{\widehat{\theta}_{h:H}}(s, \alpha, \beta, \gamma) - \widehat{V}_h(s, \sigma(\alpha), \sigma(\beta), \sigma(\gamma)) \right|$$

$$= \left| \mathbb{E}_{s' \sim \check{\mathbb{P}}(\cdot|s, \widehat{a}_h(s, \tau_{h-1}))} \left[ \check{V}_{h+1}^{\widehat{\theta}_{h+1:H}}(s', \alpha + w_h(s, \widehat{a}_h), \beta + v_h(s, \widehat{a}_h), \gamma + b_h(s, \widehat{a}_h)) \right. \right.$$

$$\left. \left. - \widehat{V}_{h+1}\left( s', \sigma\left(w_h(s, \widehat{a}_h) + \nu_{\sigma(\alpha)}\right), \sigma\left(v_h(s, \widehat{a}_h) + \nu_{\sigma(\beta)}\right), \sigma\left(b_h(s, \widehat{a}_h) + \nu_{\sigma(\gamma)}\right)\right) \right] \right|$$

$$= \left| \mathbb{E}_{s' \sim \check{\mathbb{P}}(\cdot|s, \widehat{a}_h(s, \tau_{h-1}))} \left[ \check{V}_{h+1}^{\widehat{\theta}_{h+1:H}}(s', \alpha + w_h(s, \widehat{a}_h), \beta + v_h(s, \widehat{a}_h), \gamma + b_h(s, \widehat{a}_h)) \right. \right.$$

$$- \check{V}_{h+1}^{\widehat{\theta}_{h+1:H}}\left(s', w_h(s, \widehat{a}_h) + \nu_{\sigma(\alpha)}, v_h(s, \widehat{a}_h) + \nu_{\sigma(\beta)}, b_h(s, \widehat{a}_h) + \nu_{\sigma(\gamma)}\right)$$

$$+ \check{V}_{h+1}^{\widehat{\theta}_{h+1:H}}\left(s', w_h(s, \widehat{a}_h) + \nu_{\sigma(\alpha)}, v_h(s, \widehat{a}_h) + \nu_{\sigma(\beta)}, b_h(s, \widehat{a}_h) + \nu_{\sigma(\gamma)}\right)$$

$$\left. \left. - \widehat{V}_{h+1}\left( s', \sigma\left(w_h(s, \widehat{a}_h) + \nu_{\sigma(\alpha)}\right), \sigma\left(v_h(s, \widehat{a}_h) + \nu_{\sigma(\beta)}\right), \sigma\left(b_h(s, \widehat{a}_h) + \nu_{\sigma(\gamma)}\right)\right) \right] \right|$$

$$\leq \left| \mathbb{E}_{s' \sim \check{\mathbb{P}}(\cdot|s, \widehat{a}_h(s, \tau_{h-1}))} \left[ \check{V}_{h+1}^{\widehat{\theta}_{h+1:H}}(s', \alpha + w_h(s, \widehat{a}_h), \beta + v_h(s, \widehat{a}_h), \gamma + b_h(s, \widehat{a}_h)) \right. \right.$$

$$\left. \left. - \check{V}_{h+1}^{\widehat{\theta}_{h+1:H}}\left(s', w_h(s, \widehat{a}_h) + \nu_{\sigma(\alpha)}, v_h(s, \widehat{a}_h) + \nu_{\sigma(\beta)}, b_h(s, \widehat{a}_h) + \nu_{\sigma(\gamma)}\right) \right] \right| + \frac{(H-h)\varepsilon}{2H^2}$$

$$(61)$$

where the last inequality follows by Part (a) of the inductive hypothesis at step $h + 1$. Let us now bound

$$\left| \check{V}_{h+1}^{\widehat{\theta}_{h+1:H}}(s', \alpha + w_h(s, \widehat{a}_h), \beta + b_h(s, \widehat{a}_h)) - \check{V}_{h+1}^{\widehat{\theta}_{h+1:H}}(s', w_h(s, \widehat{a}_h) + \nu_{\sigma(\alpha)}, b_h(s, \widehat{a}_h) + \nu_{\sigma(\beta)}) \right|$$

$$= \left| \mathbb{E}_{\tau \sim \check{\mathbb{P}}^{\widehat{\theta}_{h+1:H}}} \left[ \min\left\{ \mu\left( \alpha + w_h(s, \widehat{a}_h) + \sum_{\ell=h+1}^{H} w_\ell(s_\ell, a_\ell) \right) + \beta + v_h(s, \widehat{a}_h) + \sum_{\ell=h+1}^{H} v_\ell(s_\ell, a_\ell), 1 \right\} \right. \right.$$

$$\left. + \gamma + b_h(s, \widehat{a}_h) + \sum_{\ell=h+1}^{H} b_\ell(s_\ell, a_\ell) \, \Big| \, s_{h+1} = s', \tau_h = \{s, \widehat{a}_h, \tau_{h-1}\} \right]$$

$$- \mathbb{E}_{\tau \sim \check{\mathbb{P}}^{\widehat{\theta}_{h+1:H}}} \left[ \min\left\{ \mu\left( \nu_{\sigma(\alpha)} + w_h(s, \widehat{a}_h) + \sum_{\ell=h+1}^{H} w_\ell(s_\ell, a_\ell) \right) + \nu_{\sigma(\beta)} + v_h(s, \widehat{a}_h) + \sum_{\ell=h+1}^{H} v_\ell(s_\ell, a_\ell), 1 \right\} \right.$$

$$\left. \left. + \nu_{\sigma(\gamma)} + b_h(s, \widehat{a}_h) + \sum_{\ell=h+1}^{H} b_\ell(s_\ell, a_\ell) \, \Big| \, s_{h+1} = s', \tau_h = \{s, \widehat{a}_h, \tau_{h-1}\} \right] \right|.$$

Since the functions $z \mapsto \min\{z, 1\}$ and $z \mapsto \frac{1}{1+\exp(-z)}$ are 1-Lipschitz, therefore

$$\left| \check{V}_{h+1}^{\widehat{\theta}_{h+1:H}}(s', \alpha + w_h(s, \widehat{a}_h), \beta + b_h(s, \widehat{a}_h)) - \check{V}_{h+1}^{\widehat{\theta}_{h+1:H}}(s', w_h(s, \widehat{a}_h) + \nu_{\sigma(\alpha)}, b_h(s, \widehat{a}_h) + \nu_{\sigma(\beta)}) \right|$$

$$\leq |\alpha - \nu_{\sigma(\alpha)}| + |\beta - \nu_{\sigma(\beta)}| + |\gamma - \nu_{\sigma(\gamma)}| \leq \frac{\varepsilon}{2H^2}. \tag{62}$$

This combined with inequality (61) shows that

$$\left| \check{V}_h^{\widehat{\theta}_{h:H}}(s, \alpha, \beta) - \widehat{V}_h(s, \sigma(\alpha), \sigma(\beta)) \right| \leq \frac{(H+1-h)\varepsilon}{2H^2}$$

and completes the proof of the first part of the induction step.

---

[2] In the arguments that follow when $h = 1$, the outer expectation $\mathbb{E}_{s' \sim \bar{P}(\cdot|s, \widehat{a}_h(s, \tau_{h-1}))}$ is replaced by $\mathbb{E}_{s_1 \sim \rho}$ however the same arguments remain unchanged.

*Part (b):* Here let $\widehat{a}_h$ be shorthand for $\widehat{a}_h(s, \sigma(\sum_{\ell=1}^{h-1} w_\ell(s_\ell, a_\ell)), \sigma(\sum_{\ell=1}^{h-1} v_\ell(s_\ell, a_\ell)), \sigma(\sum_{\ell=1}^{h-1} b_\ell(s_\ell, a_\ell)))$. Since the policy $\widehat{\theta}_h$ picks the action $\widehat{a}_h$

$$
\begin{aligned}
&\bar{V}_h^{\widehat{\theta}_{h:H}}(s, \tau_{h-1}) \\
&= \mathbb{E}_{s' \sim \bar{\mathbb{P}}(\cdot|s,\widehat{a}_h)} \left[ \bar{V}_{h+1}^{\widehat{\theta}_{h+1:H}}(s', \{s, \widehat{a}_h, \tau_{h-1}\}) \right] \\
&= \mathbb{E}_{s' \sim \bar{\mathbb{P}}(\cdot|s,\widehat{a}_h)} \left[ \bar{V}_{h+1}^{\widehat{\theta}_{h+1:H}}(s', \{s, \widehat{a}_h, \tau_{h-1}\}) \right. \\
&\qquad \left. - \check{V}_{h+1}^{\widehat{\theta}_{h+1:H}}(s', w_h(s, \widehat{a}_h) + \nu_{i_h}, v_h(s, \widehat{a}_h) + \nu_{j_h}, b_h(s, \widehat{a}_h) + \nu_{k_h}) \right] \\
&\quad + \mathbb{E}_{s' \sim \bar{\mathbb{P}}(\cdot|s,\widehat{a}_h)} \left[ \check{V}_{h+1}^{\widehat{\theta}_{h+1:H}}(s', w_h(s, \widehat{a}_h) + \nu_{i_h}, v_h(s, \widehat{a}_h) + \nu_{j_h}, b_h(s, \widehat{a}_h) + \nu_{k_h}) \right]. \quad (63)
\end{aligned}
$$

We know that the difference of the first two terms in the expectation above

$$
\begin{aligned}
&\bar{V}_{h+1}^{\widehat{\theta}_{h+1:H}}(s', \{s, \widehat{a}_h, \tau_{h-1}\}) - \check{V}_{h+1}^{\widehat{\theta}_{h+1:H}}(s', w_h(s, \widehat{a}_h) + \nu_{i_h}, v_h(s, \widehat{a}_h) + \nu_{j_h}, b_h(s, \widehat{a}_h) + \nu_{k_h}) \\
&= \mathbb{E}_{\tau \sim \bar{\mathbb{P}}^{\widehat{\theta}_{h+1:H}}} \left[ \min\left\{ \mu\left( \sum_{\ell=1}^{h-1} w_\ell(s_\ell, a_\ell) + w_h(s, \widehat{a}_h) + \sum_{\ell=h+1}^{H} w_\ell(s_\ell, a_\ell) \right) \right.\right. \\
&\qquad\qquad\qquad + \sum_{\ell=1}^{h-1} v_\ell(s_\ell, a_\ell) + v_h(s, \widehat{a}_h) + \sum_{\ell=h+1}^{H} v_\ell(s_\ell, a_\ell), 1 \Big\} \\
&\qquad\qquad + \sum_{\ell=1}^{h-1} b_\ell(s_\ell, a_\ell) + b_h(s, \widehat{a}_h) + \sum_{\ell=h+1}^{H} b_\ell(s_\ell, a_\ell) \\
&\qquad\qquad - \min\left\{ \mu\left( \nu_{i_h} + w_h(s, \widehat{a}_h) + \sum_{\ell=h+1}^{H} w_\ell(s_\ell, a_\ell) \right) \right. \\
&\qquad\qquad\qquad + \nu_{j_h} + v_h(s, \widehat{a}_h) + \sum_{\ell=h+1}^{H} v_\ell(s_\ell, a_\ell), 1 \Big\} \\
&\qquad\qquad \left.\left. - \nu_{k_h} - b_h(s, \widehat{a}_h) - \sum_{\ell=h+1}^{H} b_\ell(s_\ell, a_\ell) \;\Big|\; s, \widehat{a}_h, \tau_{h-1} \right] \right. \\
&\overset{(i)}{\geq} - \left| \sum_{\ell=1}^{h-1} w_\ell(s_\ell, a_\ell) - \nu_{i_h} \right| - \left| \sum_{\ell=1}^{h-1} v_\ell(s_\ell, a_\ell) - \nu_{j_h} \right| - \left| \sum_{\ell=1}^{h-1} b_\ell(s_\ell, a_\ell) - \nu_{k_h} \right| \overset{(ii)}{\geq} -\frac{\varepsilon}{2H^2}.
\end{aligned}
$$

where $(i)$ follows since the functions $z \mapsto \min\{z, 1\}$ and $z \mapsto \frac{1}{1+\exp(-z)}$ are 1-Lipschitz and $(ii)$ follows since $\nu_{i_h}, \nu_{j_h}$ and $\nu_{k_h}$ are the nearest neighbors of $\sum_{\ell=1}^{h-1} w_\ell(s_\ell, a_\ell), \sum_{\ell=1}^{h-1} v_\ell(s_\ell, a_\ell)$ and $\sum_{\ell=1}^{h-1} b_\ell(s_\ell, a_\ell)$ respectively in the $\frac{\varepsilon}{6H^2}$ grid. This previous inequality combined with equation (63) yields

$$
\begin{aligned}
&\bar{V}_h^{\widehat{\theta}_{h:H}}(s, \tau_{h-1}) \\
&\qquad \geq \mathbb{E}_{s' \sim \bar{\mathbb{P}}(\cdot|s,\widehat{a}_h)} \left[ \check{V}_{h+1}^{\widehat{\theta}_{h+1:H}}(s', w_h(s, \widehat{a}_h) + \nu_{i_h}, v_h(s, \widehat{a}_h) + \nu_{j_h}, b_h(s, \widehat{a}_h) + \nu_{k_h}) \right] - \frac{\varepsilon}{2H^2}.
\end{aligned}
$$

This relates the true conditional-value function to the extended value function $\check{V}$. We will now continue further to relate the true conditional-value function to the surrogate $\widehat{V}$ that we can compute

on the grid of histories. Continuing from the previous display above we get

$$
\bar{V}_h^{\widehat{\theta}_{h:H}}(s, \tau_{h-1})
$$

$$
\geq \mathbb{E}_{s' \sim \bar{\mathbb{P}}(\cdot | s, \widehat{a}_h)} \left[ \widecheck{V}_{h+1}^{\widehat{\theta}_{h+1:H}}\left(s', w_h(s, \widehat{a}_h) + \nu_{i_h}, v_h(s, \widehat{a}_h) + \nu_{j_h}, b_h(s, \widehat{a}_h) + \nu_{k_h}\right) \right.
$$

$$
\left. - \widehat{V}_{h+1}\left(s', \sigma\left(\nu_{i_h} + w_h(s, \widehat{a}_h)\right), \sigma\left(\nu_{j_h} + v_h(s, \widehat{a}_h)\right), \sigma\left(\nu_{k_h} + b_h(s, \widehat{a}_h)\right)\right) \right]
$$

$$
+ \mathbb{E}_{s' \sim \bar{\mathbb{P}}(\cdot | s, \widehat{a}_h)} \left[ \widehat{V}_{h+1}\left(s', \sigma\left(\nu_{i_h} + w_h(s, \widehat{a}_h)\right), \sigma\left(\nu_{j_h} + v_h(s, \widehat{a}_h)\right), \sigma\left(\nu_{k_h} + b_h(s, \widehat{a}_h)\right)\right) \right] - \frac{\varepsilon}{2H^2}
$$

$$
\stackrel{(i)}{\geq} \mathbb{E}_{s' \sim \bar{\mathbb{P}}(\cdot | s, \widehat{a}_h)} \left[ \widehat{V}_{h+1}\left(s', \sigma\left(\nu_{i_h} + w_h(s, \widehat{a}_h)\right), \sigma\left(\nu_{j_h} + v_h(s, \widehat{a}_h)\right), \sigma\left(\nu_{k_h} + b_h(s, \widehat{a}_h)\right)\right) \right]
$$

$$
- \frac{(H-h)\varepsilon}{2H^2} - \frac{\varepsilon}{2H^2}
$$

$$
\stackrel{(ii)}{=} \max_{a \in \mathcal{A}} \mathbb{E}_{s' \sim \bar{\mathbb{P}}(\cdot | s, a)} \left[ \widehat{V}_{h+1}\left(s', \sigma\left(\nu_{i_h} + w_h(s, a)\right), \sigma\left(\nu_{j_h} + v_h(s, a)\right), \sigma\left(\nu_{k_h} + b_h(s, a)\right)\right) \right]
$$

$$
- \frac{(H+1-h)\varepsilon}{2H^2} \tag{64}
$$

where $(i)$ follows by using the first part of the induction hypothesis at step $h+1$ and $(ii)$ follows by the definition of $\widehat{a}_h$. With this lower bound in place let us now establish a bound on the quantity of interest

$$
\max_{a \in \mathcal{A}} \mathbb{E}_{s' \sim \bar{\mathbb{P}}(\cdot | s, a)} \left[ \bar{V}_{h+1}^{\widehat{\theta}_{h+1:H}}(s', \{s, a, \tau_{h-1}\}) \right] - \bar{V}_h^{\widehat{\theta}_{h:H}}(s, \tau_{h-1})
$$

$$
\stackrel{(i)}{\leq} \max_{a \in \mathcal{A}} \left\{ \mathbb{E}_{s' \sim \bar{\mathbb{P}}(\cdot | s, a)} \left[ \bar{V}_{h+1}^{\widehat{\theta}_{h+1:H}}(s', \{s, a, \tau_{h-1}\}) \right] \right\}
$$

$$
- \max_{a \in \mathcal{A}} \left\{ \mathbb{E}_{s' \sim \bar{\mathbb{P}}(\cdot | s, a)} \left[ \widehat{V}_{h+1}\left(s', \sigma\left(\nu_{i_h} + w_h(s, a)\right), \sigma\left(\nu_{j_h} + v_h(s, a)\right), \sigma\left(\nu_{k_h} + b_h(s, a)\right)\right) \right] \right\}
$$

$$
+ \frac{(H+1-h)\varepsilon}{2H^2}
$$

$$
\leq \max_{a \in \mathcal{A}} \left\{ \mathbb{E}_{s' \sim \bar{\mathbb{P}}(\cdot | s, a)} \left[ \bar{V}_{h+1}^{\widehat{\theta}_{h+1:H}}(s', \{s, a, \tau_{h-1}\}) \right. \right.
$$

$$
\left. \left. - \widehat{V}_{h+1}\left(s', \sigma\left(\nu_{i_h} + w_h(s, a)\right), \sigma\left(\nu_{j_h} + v_h(s, a)\right), \sigma\left(\nu_{k_h} + b_h(s, a)\right)\right) \right] \right\}
$$

$$
+ \frac{(H+1-h)\varepsilon}{2H^2} \tag{65}
$$

where $(i)$ follows by invoking inequality (64). Note that by its definition

$$
\widecheck{V}_{h+1}^{\widehat{\theta}_{h+1:H}}\left(s', \sum_{\ell=1}^{h-1} w_\ell(s_\ell, a_\ell) + w_h(s, a), \sum_{\ell=1}^{h-1} v_\ell(s_\ell, a_\ell) + v_h(s, a), \sum_{\ell=1}^{h-1} b_\ell(s_\ell, a_\ell) + b_h(s, a)\right)
$$

$$
= \bar{V}_{h+1}^{\widehat{\theta}_{h+1:H}}(s', \{s, a, \tau_{h-1}\}),
$$

therefore continuing from inequality (65)

$$\max_{a\in\mathcal{A}} \mathbb{E}_{s'\sim\bar{\mathbb{P}}(\cdot|s,a)}\left[\bar{V}_{h+1}^{\hat{\theta}_{h+1:H}}(s',\{s,a,\tau_{h-1}\})\right] - \bar{V}_h^{\hat{\theta}_{h:H}}(s,\tau_{h-1})$$

$$\leq \max_{a\in\mathcal{A}}\left\{\mathbb{E}_{s'\sim\bar{\mathbb{P}}(\cdot|s,a)}\left[\right.\right.$$
$$\check{V}_{h+1}^{\hat{\theta}_{h+1:H}}\left(s', \sum_{\ell=1}^{h-1} w_\ell(s_\ell,a_\ell)+w_h(s,a), \sum_{\ell=1}^{h-1} v_\ell(s_\ell,a_\ell)+v_h(s,a), \sum_{\ell=1}^{h-1} b_\ell(s_\ell,a_\ell)+b_h(s,a)\right)$$
$$\left.\left. - \widehat{V}_{h+1}\left(s', \sigma\left(\nu_{i_h}+w_h(s,a)\right), \sigma\left(\nu_{j_h}+v_h(s,a)\right), \sigma\left(\nu_{k_h}+b_h(s,a)\right)\right)\right]\right\}$$
$$+ \frac{(H+1-h)\varepsilon}{2H^2}$$

$$\leq \max_{a\in\mathcal{A}}\left\{\mathbb{E}_{s'\sim\bar{\mathbb{P}}(\cdot|s,a)}\left[\right.\right.$$
$$\check{V}_{h+1}^{\hat{\theta}_{h+1:H}}\left(s', \sum_{\ell=1}^{h-1} w_\ell(s_\ell,a_\ell)+w_h(s,a), \sum_{\ell=1}^{h-1} v_\ell(s_\ell,a_\ell)+v_h(s,a), \sum_{\ell=1}^{h-1} b_\ell(s_\ell,a_\ell)+b_h(s,a)\right)$$
$$- \check{V}_{h+1}^{\hat{\theta}_{h+1:H}}\left(s', \nu_{i_h}+w_h(s,a), \nu_{j_h}+v_h(s,a), \nu_{k_h}+b_h(s,a)\right)$$
$$+ \check{V}_{h+1}^{\hat{\theta}_{h+1:H}}\left(s', \nu_{i_h}+w_h(s,a), \nu_{j_h}+v_h(s,a), \nu_{k_h}+b_h(s,a)\right)$$
$$\left.\left. - \widehat{V}_{h+1}\left(s', \sigma\left(\nu_{i_h}+w_h(s,a)\right), \sigma\left(\nu_{j_h}+v_h(s,a)\right), \sigma\left(\nu_{k_h}+b_h(s,a)\right)\right)\right]\right\}$$
$$+ \frac{(H+1-h)\varepsilon}{2H^2}$$

$$\overset{(i)}{\leq} \max_{a\in\mathcal{A}}\left\{\mathbb{E}_{s'\sim\bar{\mathbb{P}}(\cdot|s,a)}\left[\check{V}_{h+1}^{\hat{\theta}_{h+1:H}}\left(s', \nu_{i_h}+w_h(s,a), \nu_{j_h}+v_h(s,a), \nu_{k_h}+b_h(s,a)\right)\right.\right.$$
$$\left.\left. - \widehat{V}_{h+1}\left(s', \sigma\left(\nu_{i_h}+w_h(s,a)\right), \sigma\left(\nu_{j_h}+v_h(s,a)\right), \sigma\left(\nu_{k_h}+b_h(s,a)\right)\right)\right]\right\}$$
$$+ \frac{(H+1-h)\varepsilon}{2H^2} + \frac{\varepsilon}{2H^2}$$
$$\overset{(ii)}{\leq} \frac{(H-h)\varepsilon}{2H^2} + \frac{(H+2-h)\varepsilon}{2H^2} = \frac{(H+1-h)\varepsilon}{H^2}.$$

where $(i)$ follows by bounding

$$\check{V}_{h+1}^{\hat{\theta}_{h+1:H}}\left(s', \sum_{\ell=1}^{h-1} w_\ell(s_\ell,a_\ell)+w_h(s,a), \sum_{\ell=1}^{h-1} v_\ell(s_\ell,a_\ell)+v_h(s,a), \sum_{\ell=1}^{h-1} b_\ell(s_\ell,a_\ell)+b_h(s,a)\right)$$
$$- \check{V}_{h+1}^{\hat{\theta}_{h+1:H}}\left(s', \nu_{i_h}+w_h(s,a), \nu_{j_h}+v_h(s,a), \nu_{k_h}+b_h(s,a)\right) \leq \frac{\varepsilon}{2H^2}$$

using the same logic as we used above to arrive at inequality (62), and $(ii)$ follows by using the Part (a) of the inductive hypothesis at step $h+1$. This proves the second part of the inductive hypothesis.

*Part (c):* Recall the definition of

$$\widehat{a}_h(s,i,j,k) \in \arg\max_{a\in\mathcal{A}} \mathbb{E}_{s'\sim\bar{\mathbb{P}}(\cdot|s,a)}\left[\widehat{V}_{h+1}(s',\sigma(w_h(s,a)+\nu_i),\sigma(v_h(s,a)+\nu_j),\sigma(b_h(s,a)+\nu_k))\right],$$
$$\widehat{V}_h(s,i,j,k) := \max_{a\in\mathcal{A}} \mathbb{E}_{s'\sim\bar{\mathbb{P}}(\cdot|s,a)}\left[\widehat{V}_{h+1}(s',\sigma(w_h(s,a)+\nu_i),\sigma(v_h(s,a)+\nu_j),\sigma(b_h(s,a)+\nu_k))\right].$$

For a fixed quartet $(s,i,j,k)$ to calculate $\widehat{a}_h(s,i,j,k)$ it is possible to first calculate

$$\widehat{V}_{h+1}(s',\sigma(w_h(s,a)+\nu_i),\sigma(w_h(s,a)+\nu_j),\sigma(b_h(s,a)+\nu_k))$$

for all $s' \in \mathcal{S}$ and all $a \in \mathcal{A}$. Since we already have access to the tensor the entire $\widehat{V}_{h+1}$ this takes poly$(|\mathcal{S}||\mathcal{A}|)$ time and memory. Once we have calculated this it is possible to use this to calculate

$$\mathbb{E}_{s' \sim \bar{\mathbb{P}}(\cdot|s,a)} \left[ \widehat{V}_{h+1}(s', \sigma(w_h(s,a) + \nu_i), \sigma(b_h(s,a) + \nu_j)) \right]$$

for all choices of $a$ (we can do this since we have access to the distribution $\bar{\mathbb{P}}$) using poly$(|\mathcal{S}|, |\mathcal{A}|)$ time and memory. After we have enumerated this value for all $a \in \mathcal{A}$ we can identify $\widehat{a}_h(s, i, j, k)$ and $\widehat{V}_h(s, i, j, k)$ for this quartet. There are $|\mathcal{S}|m^3 = |\mathcal{S}| \left( \left\lceil \frac{12H^2\zeta}{\varepsilon} \right\rceil \right)^3$ quartets. Therefore it is possible to calculate both these tensors using poly$\left(|\mathcal{S}|, |\mathcal{A}|, H, \zeta, \frac{1}{\varepsilon}\right)$ time and memory, which proves our claim.

This completes the proof of all parts of the induction hypothesis.

**Part II: Using the induction hypothesis to prove the lemma.** We begin by proving that the policy $\widehat{\theta}$ can be found efficiently. To see this, notice that at every step the policy $\widehat{\theta}$ only requires to know the tensor of actions $\widehat{a}_h$. Starting from $h = H$, we have shown that each $\widehat{a}_h$ can be computed using poly$\left(|\mathcal{S}|, |\mathcal{A}|, H, \zeta, \frac{1}{\varepsilon}\right)$ time and memory. Thus, all $H$ of these tensors can be found using poly$\left(|\mathcal{S}|, |\mathcal{A}|, H, \zeta, \frac{1}{\varepsilon}\right)$ time and memory.

Now let's prove that

$$\bar{V}^{\theta_\star} - \bar{V}^{\widehat{\theta}} \leq \varepsilon.$$

Define a policy $\theta_{\star h} := \left( \theta_{\star 1}, \ldots, \theta_{\star h}, \widehat{\theta}_{h+1}, \widehat{\theta}_H \right)$ for $h \in \{0, \ldots, H\}$. Therefore,

$$\bar{V}^{\theta_\star} - \bar{V}^{\widehat{\theta}} = \sum_{h=H}^{1} \bar{V}^{\theta_{\star h}} - \bar{V}^{\theta_{\star h-1}}. \tag{66}$$

Consider any term in this decomposition above,
$\bar{V}^{\theta_{\star h}} - \bar{V}^{\theta_{\star h-1}}$

$$= \mathbb{E}_{s_1 \sim \rho, \, \tau_{h-1} \sim \bar{\mathbb{P}}^{\theta_\star 1:h-1}} \left[ \mathbb{E}_{s_h \sim \bar{\mathbb{P}}(\cdot|s_{h-1}, a_{h-1})} \left[ \max_{a \in \mathcal{A}} \mathbb{E}_{s' \sim \bar{\mathbb{P}}(\cdot|s_h, a)} \left[ \bar{V}_{h+1}^{\widehat{\theta}_{h+1:H}} (s', \{s_h, a, \tau_{h-1}\}) \right] \right.\right.$$

$$\left.\left. - \bar{V}_h^{\widehat{\theta}_{h:H}} (s_h, \tau_{h-1}) \right] \right]$$

where the outer expectation $\mathbb{E}_{\tau_{h-1} \sim \bar{\mathbb{P}}^{\theta_\star 1:h-1}}$ is over the randomness in the first $h - 1$ round where the policy is $(\theta_{\star 1}, \ldots, \theta_{\star h-1})$ and the initial state is $s_1$. Now by invoking the second part of the induction hypothesis to bound the RHS in the display above we get

$$\bar{V}^{\theta_{\star h}} - \bar{V}^{\theta_{\star h-1}} \leq \frac{(H + 1 - h)\varepsilon}{H^2}.$$

Plugging this into equation (66) we conclude that

$$\bar{V}^{\theta_\star} - \bar{V}^{\widehat{\theta}} \leq \frac{\varepsilon}{2H^2} \sum_{h=1}^{H} (H + 1 - h) < \frac{\varepsilon}{2H^2} \sum_{h=1}^{H} (H + 1) \leq \varepsilon$$

completing our proof. ∎

## E.2 Proof of Proposition 3.7

Recall the statement of the proposition from above.

**Proposition 3.7.** *For any $t \in [N]$ define $\widetilde{V}_t^{\mathsf{sd}}(\pi) := \mathbb{E}_{s_1 \sim \rho, \, \tau \sim \widehat{\mathbb{P}}_t^\pi(\cdot|s_1)} \left[ \widetilde{\mu}_t^{\mathsf{sd}}(\widehat{\mathbf{w}}_t, \tau) \right]$. Given any $\varepsilon > 0$, under Assumptions 2.2, 3.3 and 3.4 it is possible to find a policy $\widehat{\pi}^{(t)}$ that satisfies*

$$\widetilde{V}_t^{\mathsf{sd}}(\pi^{(t)}) - \widetilde{V}_t^{\mathsf{sd}}(\widehat{\pi}^{(t)}) \leq \varepsilon,$$

*using at most* poly$\left(|\mathcal{S}|, |\mathcal{A}|, H, d, B, \|\widehat{\mathbf{w}}_t\|_2, \frac{1}{\varepsilon}, \log\left(\frac{N}{\delta}\right)\right)$ *time and memory.*

**Proof** The proof shall follow by simply invoking Lemma E.1. Recall from equation (D.5) that

$$\widetilde{\mu}_t^{\mathsf{sd}}(\widehat{\mathbf{w}}_t, \tau) := \min\left\{\mu\left(\mathbf{w}^\top \phi(\tau)\right) + \sqrt{\kappa}\beta_t(\delta)\sum_{h=1}^H \|\phi_h(s_h, a_h)\|_{\mathbf{\Sigma}_t^{-1}}, 1\right\} + \sum_{h=1}^{H-1} \xi_{s_h, a_h}^{(t)}.$$

First notice that since $\|\phi(\tau)\|_2 \leq 1$ we have that

$$|\widehat{\mathbf{w}}_t, \phi(\tau)| \leq \|\widehat{\mathbf{w}}_t\|_2 \|\phi(\tau)\|_2 \leq \|\widehat{\mathbf{w}}_t\|_2. \tag{67}$$

Next observe that

$$\sqrt{\kappa}\beta_t(\delta)\sum_{h=1}^H \|\phi_h(s_h, a_h)\|_{\mathbf{\Sigma}_t^{-1}} \leq \sqrt{\kappa}\beta_t(\delta)\sqrt{\lambda_{\max}(\mathbf{\Sigma}_t^{-1})}\sum_{h=1}^H \|\phi_h(s_h, a_h)\|_2 \tag{68}$$

$$\overset{(i)}{\leq} \frac{\sqrt{\kappa}\beta_t(\delta)}{\sqrt{\lambda_{\min}(\mathbf{\Sigma}_t)}}\sqrt{H}\sqrt{\sum_{h=1}^H \|\phi_h(s_h, a_h)\|_2^2} \tag{69}$$

$$\overset{(ii)}{\leq} \frac{\sqrt{\kappa}\beta_t(\delta)}{\sqrt{\lambda_{\min}(\mathbf{\Sigma}_t)}}\sqrt{H}\|\phi(\tau)\|_2 \tag{70}$$

$$\overset{(iii)}{\leq} \frac{\sqrt{\kappa}\beta_t(\delta)}{\sqrt{\kappa}}\sqrt{H}\|\phi(\tau)\|_2 \tag{71}$$

$$\leq \sqrt{H}\beta_t(\delta) \overset{(iv)}{\leq} \sqrt{H} \times \mathsf{poly}\left(d, B, \log\left(\frac{N}{\delta}\right)\right), \tag{72}$$

where $(i)$ follows since for any $z \in \mathbb{R}^H$, $\|z\|_1 \leq \sqrt{H}\|z\|_2$, $(ii)$ follows since by Assumption 3.4 for any $h \neq h' \in [H]$, the features $\phi_h$ and $\phi_{h'}$ are orthogonal and by Assumption 3.3 the feature map $\phi$ is sum-decomposable, $(iii)$ follows since $\mathbf{\Sigma}_t \succeq \kappa\mathbf{I}$, and $(iv)$ follows by the definition of $\beta_t(\delta)$ in equation (4). Finally the definition of $\xi^{(t)}$ in equation (7) we know that

$$\left|\sum_{h=1}^{H-1} \xi_{s_h, a_h}^{(t)}\right| \leq 2H. \tag{73}$$

In light of inequalities (67), (72) and (73) we can conclude that if we invoke Lemma E.1 with a $\zeta$ that is a large enough polynomial in $\|\widehat{\mathbf{w}}_t\|_2, d, B, \log(N/\delta), H$ then the claim follows. ∎

# F   Experiments

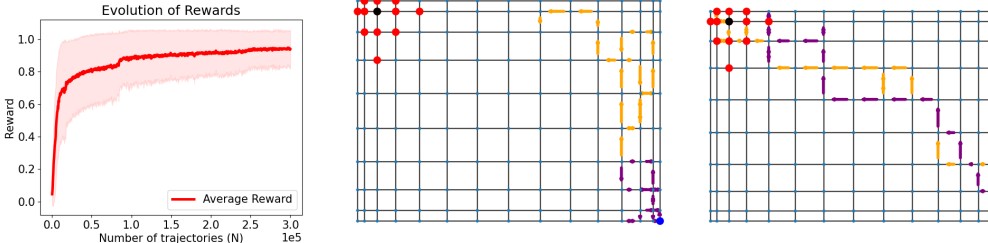

**Figure 1. Left:** Reward learning curve averaged over 40 independent runs. The shaded region represents a confidence interval which is ±standard deviation. **Middle:** The purple and yellow paths represent two sample paths taken by an initial random policy. **Right:** The purple and yellow paths represent two sample paths taken by a trained policy.

In this section we experimentally show that it is possible to learn a good policy in a simple non-Markovian domain with binary rewards—received once per episode—using a policy gradient

algorithm. We parameterize each policy $\pi_\theta$ by $\theta \in \mathbb{R}^k$. The gradients of the value function can be computed using the REINFORCE [36] algorithm as follows

$$\nabla_\theta V^{\pi_\theta} = \mathbb{E}_{y_\tau, \tau \sim \mathbb{P}^\pi} \left[ y_\tau \left( \sum_{h=1}^{H} \nabla_\theta \log \left( \pi_\theta(a_h | s_h) \right) \right) \right].$$

We approximate this expectation empirically by using $30$ sample trajectories, and use the Adam optimizer [21] with a default step size of one to update the policy. We studied the behavior of this algorithm on a custom $10 \times 15$ grid environment. The agent is initialized at a random location on the grid denoted by the large blue dot. Then the agent is allowed to take one of the actions $\{\mathrm{UP}, \mathrm{DOWN}, \mathrm{LEFT}, \mathrm{RIGHT}\}$, and move to an adjacent node (if permitted). During the last three steps of an episode, with $H = 30$, if the agent stays at either the black dot ('goal') or at any adjacent nodes marked by the red dots, then the agent receives a reward of $1$, while if the agent is not at one of these nodes during the last three steps then it receives a reward of $0$. The location of the 'goal' node is also randomly chosen at each episode. We parametrize the policy using a fully connected neural network with 10 hidden layers and with width $4$. The state representation that is fed to this policy is of the form $(x^{\mathrm{current}}, y^{\mathrm{current}}, x^{\mathrm{goal}}, y^{\mathrm{goal}})$, where $(x^{\mathrm{current}}, y^{\mathrm{current}})$ represents the current coordinates of the agent and $(x^{\mathrm{goal}}, y^{\mathrm{goal}})$ denotes the coordinates of the 'goal' node.