# OpenReview forum: "On the Theory of Reinforcement Learning with Once-per-Episode Feedback"
_NeurIPS.cc/2021/Conference — NeurIPS 2021 Poster_

### Official Review · Reviewer_NZK6 · 2021-07-07

**Rating:** 6
**Confidence:** 3

**Summary:**


The authors studied an RL setting in which the feedback is received once per-episode. Unlike previous works, the authors assumed the value of a policy is given by E_{\tau~\pi}[ logistic(w * \phi(\eta))] where \tau is a trajectory sampled by following the policy \pi. That is, the transition model is Markovian, but the model is not an MDP, since the value is a non-linear function of the cumulative reward.

The authors then studied two algorithms for this setting.
1) A computationally hard algorithm, based on GLM-bandits analysis.
2) Under additional assumptions -- 3.3 and 3.4 -- a computationally  tractable algorithm for this setting (nevertheless, the computational complexity of this approach scales poly(N), where N is the number of episodes, which makes this algorithm problematic from practical perspective).


**Ethics Review Area:**

["I don’t know"]

**Limitations And Societal Impact:**

Yes

**Main Review:**

Pros.
1) The setting is of interest and the results extend what is currently known for the setting in feedback is received only once per episode.

2) The planning algorithm (presented in Appendix E) which the authors suggested to solve line 9 of algorithm 1 is novel to the best of my knowledge.

Cons.
1) I understand why the suggested model is natural from a theoretical perspective. However, it seems not very natural from practical point of view (and the authors do not motivate this model from the practical perspective). In which practical scenarios we can expect the value of a policy to be given as E_{\tau~\pi}[ logistic(w * \phi(\eta))]?

2) The writing needs to be improved in several critical aspects in my opinion.

a) The assumptions made are quite restrictive (both the modelling assumption and the assumptions needed for the analysis). The authors do not elaborate on these assumptions - why these assumptions (especially assumption 3.4) are needed? why the analysis fails without them? It seems that assumption 3.4 is quite harsh on its own, let alone knowing the value of \omega. Can the suggested algorithm be adaptive to an unknown value of \omega?

b) The relation to GLM is not explain well enough - what is special about the logistic mapping as oppose to GLM? in section 2 can't the result be established for a general GLM function instead of the logistic mapping? if not, why is it challenging? I believe that these questions are much important for a reader. Furthermore, does the parameter \kappa arises in bandits with GLM parametrization? or is it a new parameter the authors introduce.

c) In the experimental part a different algorithm is being implemented. I believe this can create confusion and harms the readability of this work.

Reference to previous works.
1) Lemma C.1. can be found in "Improved Algorithms for Linear Stochastic Bandits". I believe the authors should cite it.

-) Minor questions.
-) What does equation 66 mean? why the LHS equal to the RHS?

-) Minor
1) Stating that this work deals with non-Markovian reward is confusing. Is there a reason not to explicitly mention the assumption 2.1 in the name? for example, logistic-reward function / trajectory logistic-reward function. This makes the parametric assumption much more apparent.
2) I found the notation (s_h,a_h) in definition 8b confusing. Possibly it would be easier to follow if it would be stated explicitly after the equation that \tau = ( s_1,a_1,.., s_H,a_H).
3) As the authors stated, solving line 9 in algorithm 1 seems hard. Then, under additional assumptions, a computationally efficient algorithm is suggested. I believe that writing in line 10 in the abstract that the algorithm is computationally efficient is a bit misleading (further assumptions should be made to make the algorithm computationally efficient).
4) The table of content in appendix contains the sections of the main paper.

-) Typos.
Line 853, "and" -> "and any".

**Time Spent Reviewing:**

6

---

> ### Author Response · Authors · 2021-08-10
> **Thank you for your review**
>
> We want to thank the reviewer for their comments. We are glad the reviewer liked the paper. We would like to address the reviewer’s concerns one by one.
>
> - Applications of our setting:
>
> A guiding example for our setting was the following. Consider a subjective task such as making a robot dance in an aesthetically pleasant way. In this case there is no success state. A dance sequence is aesthetically pleasant when the combination of movements across the timesteps of the trajectory form a coherent and beautiful whole. The stochasticity of the label (which is modelled by the logistic map) may come from asking this subjective judgement (was the dance pleasant to the eye) to a single random member of a target audience. Our setting can capture this and other related settings. We would be happy to discuss this further in our introduction.
>
> - Regarding Assumption 3.4:
>
> This is required in our analysis to show that the sum-decomposable bonus that we add in Eq. 11a is optimistic. This is key to developing a computational tractable algorithm. The sum-decomposability of the bonus is not guaranteed unless the features satisfy this orthogonality assumption. We would be happy to add additional discussion regarding this in the paper.
>
> - Regarding knowledge of omega:
>
> Thanks for this great question! If omega is not known in advance then one cannot set N_{EUL} and N_{EVAL} directly, which affects Algorithm 2 and its analysis Lemma 3.5. However, a slightly more complicated algorithm based on an adaptive stopping schedule for N_{EUL} and N_{EVAL} can be used to bypass this issue.
> In short, if we didn't know omega, the idea is to estimate a confidence interval radius around the value function of the optimal policy “pi_star” in each EULER round (in line 5 of the algorithm). This confidence interval will depend on both the number of timesteps played and the theoretical regret guarantee of the EULER algorithm. The EULER algorithm is stopped when the empirical average of the collected rewards is at least twice this confidence radius. This ensures the current policy has a reward of at least omega/2. A similar adaptive procedure can be used instead of fixing the value of N_{EVAL}.
>
> - Relation to GLMs:
>
> We study the specific setting with the logistic map as a first step. When the rewards are dictated by the logistic map we could rely on the vast literature on logistic bandits to adapt Lemma 3.1 which provides guarantees for the quality of hat{w}. We believe that other GLM models could also be used, and similar guarantees could be established if one establishes analogs of Lemma 3.1 for that GLM model.
>
> The parameter kappa also appears in the past study of logistic bandits and is not introduced by us for the first time.  We would be happy to add both clarifications in the paper.
>
> - Regarding the experiments:
>
> The objective of this section is to show that despite the potential intractability of Algorithm 1 in the worst case, in some cases it is possible to use policy gradients to find an optimal policy even if the rewards are non-Markovian. This means that in practice step (2) of Algorithm 1 can be executed via a simple policy gradient algorithm, possibly using a RNN policy network.
>
> - Regarding Equation 66: The equality follows by the definition of the policies \theta_{\star h} and because the sum telescopes.
>
>
> We would be happy to add the missing citation, and fix the typos, minor points raised by the reviewer.

---

> > ### Author Response · Authors · 2021-08-23
> > **Follow-up**
> >
> > We just wanted to check in and ask if the rebuttal clarified and answered the questions raised your review. We would be very happy engage further if there are additional questions!

---

### Official Review · Reviewer_ViC9 · 2021-07-11

**Rating:** 5
**Confidence:** 4

**Summary:**

The paper tackles RL problems where feedback is received only at the end of each episode. Specifically, they extend the trajectory-feedback model of Efroni et al.[10] in two aspects: first, the reward is logistic, and not the sum of state-action rewards, and second, they allow per-trajectory feature representation, instead of working with state-action-visitations. In this setting, they first combine UCBVI with a UCB algorithm for GLMs to derive $\sqrt{N}$ regret bounds (where $N$ is the number of episodes). However, the resulting algorithms is computationally intractable, Then, under specific assumptions on the feature representation (‘sum-decomposable features’ and ‘explorability’) they provide an additional algorithm with $N^{2/3}$ regret and polynomial per-episode computational complexity.

**Ethical Concerns:**

None.

**Limitations And Societal Impact:**

In general, most limitations are clear. I would only add explicit bounds on the computational complexity (as mentioned in the main review) and a comment about the requirement to know $\omega$.

**Main Review:**

I think that the setting of once-per-episode feedback (or trajectory feedback) is interesting for two reasons. First, as states in the paper, per-state feedback is not always available, and per-episode feedback is more natural in many applications. Then, this model actually bypasses the need for reward shaping. In addition, it is a good starting point for learning RL problems with non-Markovian rewards.

However, I think that there is a major computational issue with the results of this work. First, although Algorithm 1 achieves an optimal regret rate of $\sqrt{N}$ (up to problem-dependent constants of $S, A, H$), the algorithm is computationally intractable. Second, I have some concerns regarding the UCBVI algorithm with the added exploration:
- The two parts of assumption 3.4 are somewhat contradicting each other. The most natural way to meet the condition of $\phi_h(s, a)^T\phi_{h’}(s, a)=0$ is to allocate different features (or orthogonal subspaces) to each timestep. However, this means that the second requirement of the assumption does not hold. Is there any reasonable problem instance with natural features for which both parts of the assumption simultaneously hold?
- The main reason for this algorithm is its computation tractability, but the computational requirements are hidden in $O$-notations, and I found it hard to understand them even when looking at the appendix. Concretely, what is the computational complexity for any given $\epsilon$, and specifically, for $\epsilon$ such that the regret will not degrade any further? Even if it is polynomial but badly depends on $S, A,1/\epsilon$, the algorithm is not necessarily tractable.
- I want to ignore the logistic function for a moment. Then, if I understand correctly, the sum-decomposable assumption essentially results with a (structured) Markovian reward ($r_h(s,a)=w^T\phi_h(s,a)$), and then, one could probably just use a variant of the algorithm in [10]. When adding the logistic function, one can then replace the linear TS with a GLM TS, and since GLMs are monotone, maximizing the internal linear function is equivalent to maximizing the GLM. Notably, doing so might lead to an algorithm that only requires GLM TS and standard planning. Of course, there are numerous small details to sort out, but if you agree that this should work (and I would love to hear your opinion on the matter), then this would result in a computationally efficient $\sqrt{N}$ algorithm, and not $N^{2/3}$. Then I would expect this paper to prove comparable regret bounds and/or extend to feature maps that are not so easily covered.
- Finally, another issue with assumption 3.4 is that the orthogonality requirement $\phi_h(s, a)^T\phi_{h’}(s', a')=0$ leads to a dimension constraint of $d\ge SA(H-1)$ (assuming meaningful features $\phi_h(s, a)\ne 0$) - this is almost the same number of features as the direct parameterization, so there's no dimensionality reduction. If the sum-decomposable assumption indeed leads to a Markovian reward and there's no dimensionality reduction, is there any reason not to use the direct parameterization?

***************************

Other questions/comments:
- Can you further explain why you specifically require the logistic function and whether the results could be extended to other GLMs?
- In general, it would have been nice to see some examples of problems with natural feature maps that are not the direct parameterization.
- Another issue that should be mentioned (but is arguably unavoidable) – the minimization of the regularized cross-entropy loss – maybe it can be mitigated by doing rare updates (e.g., according to some doubling trick)?
- Algorithm 2: EULER is, in general, a regret minimization algorithm, so it usually does not output a policy – what is $U_n$?
- Another comment/limitation – for lemma 3.5, one must know $\omega$, which seems nontrivial, especially since it depends on the dynamics of the MDP.
- Can you add an algorithm block (even in the appendix) for the planning algorithm of Proposition 3.7?
- Simulations – what is the added value of simulating REINFORCE? In my opinion, simulations are a ‘nice-to-have’ in this work, but if you perform them, the interesting thing is to simulate your algorithm (and maybe measure its runtime and show that it is a no-regret algorithm, namely, converges to the optimal policy at a reasonable rate).

***************************

Minor issues:
- Algorithm 3, step 2 – the if-else seems unnecessary.

***************************

I thank the authors for their response. Even after the clarifications, I still find the assumptions and the resulting bounds a bit too weak, so I decided to leave my score unchanged.

**Time Spent Reviewing:**

6

---

> ### Author Response · Authors · 2021-08-10
> **Thank you for your review!**
>
> We want to thank the reviewer for their comments. We are glad the reviewer liked the paper. We would like to address the reviewer’s concerns one by one.
>
> - “The two parts of assumption 3.4 are somewhat contradicting each other.”
> We thank the reviewer for this astute observation. There is a typo in the text. The second half of the assumption should read:
>
> There exists $\omega \in (0,1)$ such that for any unit vector $\mathbf{v} \in \mathbb{R}^d$ we have that
> \begin{equation*}
> \sup_{\pi \in \Pi} \mathbb{E}_{s_1 \sim \rho, \tau \sim \mathbb{P}^{\pi}} \left[  \sum_h \mathbf{v}^\top \phi_h(s_h,a_h)  \right] \geq \omega
> \end{equation*}
>
> That is, instead of the assumption holding for each h, it only needs to holds for the sum over all h. We note that this corrected assumption holds in the case of a tabular RL representation with indicator features such that, for any state there exists a policy that visits said state with a minimum probability of at least \omega.
>
> - “The main reason for this algorithm is its computation tractability… ”:
>
> We would like to note that all the dependencies on the quantities S, A and 1/\epsilon are small polynomials. We will make sure these are exhibited more explicitly in the final version of the manuscript. We consider our contribution to be theoretical in nature, as such we are applying the definition of tractability to mean that there exists an algorithm that runs in polynomial time and memory.
>
> - “I want to ignore the logistic function for a moment….”:
>
> We would like to clarify that it is not possible to build intuition for our setting by ignoring the logistic function. The presence of the logistic map, even under Assumption 3.3, renders the rewards, optimal policies and optimal value functions to be non-Markovian.
>
> - “Finally, another issue with assumption 3.4 …”:
>
> We would like to clarify and respectfully point out that for the explorability assumption to hold it suffices if  d \geq H. To see why this is true, consider the example where each phi_h maps an orthogonal 1-dimensional subspace. Here phi_h(s,a)^T phi_{h’}(s’,a’) = 0.

---

> > ### Author Response · Authors · 2021-08-23
> > **Follow-up**
> >
> > We just wanted to check in and ask if the rebuttal clarified and answered the questions raised your review. We would be very happy engage further if there are additional questions!

---

### Official Review · Reviewer_s9t9 · 2021-07-16

**Rating:** 5
**Confidence:** 3

**Summary:**

The paper studies a variant of the classical RL problem in which, instead of having a reward function depending on the state-action pairs, a reward function depending on the whole trajectory is considered. The rationale behind this choice is that, in some problems, the reward of the whole trajectory is revealed at the end only. After having formalized the setting, considering a binary reward governed by a logistic model, a first algorithm is proposed achieving sublinear regret for the case of finite states and actions. To overcome some limitations of the first algorithm (in particular the complexity in computing the optimal policy at every iteration), a second algorithm is presented, working under additional assumptions and achieving sublinear (although higher) regret. Finally, a small experiment is provided.

**Limitations And Societal Impact:**

The paper is mainly theoretical, thus I do not forsee any societal impact.

**Main Review:**

- Assumption 2.2: This assumption requires that the weights w_{\star} are bounded in L2-norm by a finite constant B. As a consequence, from Assumption 2.1 and given that the features \phi are bounded, the reward function must be a stochastic mapping from trajectories to {0,1} (that is, there always exists a non-zero probability that every trajectory is rewarded with either 0 or 1). What if the mapping between trajectories and reward were deterministic?
- Section 3.1 and 3.2: The authors claim in Section 3.2 that Algorithm 1 could not be computationally efficient since finding the optimistic policy might be difficult. The optimistic policy is, in general, a non-Markovian policy. Did the author provide a (maybe intractable) algorithm to compute it?
- Assumption 3.4: I couldn't grasp the intuition behind this assumption. The first condition seems a form of "orthogonality" between the features at different steps h and h'. Can the authors elaborate more on this assumption, possibly adding some explanation in the paper?
- If I understood well, the problem with Algorithm 1 is the possible intractability in the computation of the optimistic (non-Markovian) policy. Requiring that the rewards are Markovian, as in Assumption 3.3, wouldn't already solve the problem? With Markovian rewards, the optimal policy is Markovian and I guess that the computation of the optimistic policy becomes tractable (you can use extended value iteration, for instance). I haven't understood the intuition behind the need for computing the exploration mixture policy. I would appreciate if the authors could explain this point in more detail.
- The experiment seems detached from the rest of the paper. The only contact point is that the considered environment is non-Markovian. Indeed, the algorithms presented in the paper are not employed in the experiment (that uses policy gradient methods). What is the point of the experiment?
- Concerning the related works, in [1], a similar setting is considered in which an optimization over the parameters of the policy is carried out, also achieving sublinear guarantees on the reget. The formulation is trajectory-based and actually does not need to observe the reward for the individual steps but just the trajectory return.
- I haven't checked the appendix.

***Minor***
- Line 56: Section E - > Appendix E
- Section 2.2: Since a finite-horizon MDP is considered, for the general case, the transition model P(\cdot|s,a) should depend on the step h too. The fact that it is independent of h is a requirement for the presented algorithm? Considering P_h(\cdot|s,a) would require changes in the analysis? In particular, would this affect the regret in Theorem 3.2?
- Section 2.2: The paper assumes that the initial state distribution \rho is known to the learner. What if it were unknown?
- Equation 4: The confidence bound \rho_t makes confusion with the initial state distribution \rho. I suggest changing the symbol.
- Line 145: hasn't been - > has not been
- Algorithm 1 and 3, line 2: Shouldn't \hat{\mathbb{P}}_1 be initialized as uniform?

***Overall***
I think that the paper contains interesting ideas. However, I have some concerns about some of the presented assumptions. I will opt for a borderline score, although I am willing to adjust my score based on the author's response.

[1] Papini, Matteo, Alberto Maria Metelli, Lorenzo Lupo, and Marcello Restelli. "Optimistic policy optimization via multiple importance sampling." In International Conference on Machine Learning, pp. 4989-4999. PMLR, 2019.

***Post Discussion***
I thank the authors for the feedback. I read the reviews and the author's feedback. Although the once-per-episode feedback is an interesting setting, I found some choices of the formulation unmotivated; in particular, the fact that the episode reward can be just 0 or 1 and that cannot be deterministic. These assumptions are motivated by the analysis but seem not to be related to the nature of the problem. So, I remain inclined towards rejection.

**Time Spent Reviewing:**

3

---

> ### Author Response · Authors · 2021-08-10
> **Thank you for your review**
>
> We want to thank the reviewer for their comments. We would like to address the reviewer’s concerns one by one.
>
> - “What if the mapping between trajectories and reward were deterministic?”:
>
> This is a great question! Our model can handle near deterministic rewards (think of a large norm bound on both w_star and phi) at the cost of an increase in the dependence on \kappa. As this parameter goes to infinity, our regret guarantee becomes looser. Intuitively, this makes sense as it should get harder and harder to get any valid regret guarantees if the feedback is too sparse and only obtained when certain specific trajectories are encountered.
>
> - “Did the author provide a (maybe intractable) algorithm to compute it?”:
>
> A naive intractable algorithm would be an exhaustive search over an epsilon net of policies (this can be done since the dynamics \hat{P} is known to the algorithm).
>
> - Regarding Assumption 3.4:
>
> This is required in our analysis to show that the sum-decomposable bonus that we add in Eq. 11a is optimistic. This is key to developing a computational tractable algorithm. The sum-decomposability of the bonus is not guaranteed unless the features satisfy this orthogonality assumption. We would be happy to add additional discussion regarding this in the paper.
>
> - “Requiring that the rewards ... solve the problem?” “With Markovian rewards, …. for instance).”:
>
> We would like to clarify Assumption 3.3 is NOT requiring the rewards to be markovian. This is because the value function of a policy equals an expectation of the logistic function evaluated on the dot product of these features and the unknown reward vector, the resulting value function is not linear on these features, and thus is non Markovian. Again, we would be very happy to add a clarification in the paper after we introduce this assumption.
>
> - Regarding Exploration mixture policy:
>
> The result of the EULER algorithm is to produce an ensemble of policies, the average value of which is close to be optimal.
>
> - Regarding the experiment
>
> The objective of this section is to show that despite the potential intractability of Algorithm 1 in the worst case, in some cases it is possible to use policy gradients to find an optimal policy even if the rewards are non-Markovian. This means that in practice step (2) of Algorithm 1 can be executed via a simple policy gradient algorithm, possibly using a RNN policy network.
>
> - “Concerning the related works, in [1]...”:
>
> This work considers the case where a linear reward is only evaluated at the end of a trajectory. This setting is markedly different from ours since in [1] the reward signal is ultimately Markovian in nature whereas here it is not.
>
> - “Considering P_h(\cdot|s,a) would require changes in the analysis?”:
>
> Our results hold for that case as well. There would be an extra dependence on H in the regret corresponding to the estimation problem for the transition dynamics over all steps in the trajectory.
>
> - Minor comments and Typos:
> We will make sure there are no typos present in the camera ready version of the manuscript and also address the minor comments identified.

---

> > ### Author Response · Authors · 2021-08-23
> > **Follow-up**
> >
> > We just wanted to check in and ask if the rebuttal clarified and answered the questions raised your review. We would be very happy engage further if there are additional questions!

---

### Official Review · Reviewer_LW3y · 2021-07-16

**Rating:** 4
**Confidence:** 3

**Summary:**

The authors consider the setting of reinforcement learning where the agent is provided with binary rewards with reward-probabilities determined by a logistic linear model. Unlike usual, the reward applies to the entire trajectory. The authors assume features to be available to the agent. The first algorithm proposed by the authors combines bounds on logistic bandits and bounds on model learning to learn an optimistic approximation of the reward which, if optimized with an unspecified algorithm, provides a sqrt(N) regret bound. In the second algorithm, the authors propose an approach to optimize said reward under additional assumptions on the structure of the trajectory features to obtain a N^2/3 regret bound.


**Limitations And Societal Impact:**

-

**Main Review:**

The paper is well-written and easy to read. The problem studied in this work is relevant to a variety of practical applications.

In its most general form, as specified in Section 3, the algorithm and its accompanying regret bound depends entirely on finding the optimal trajectory given the learned reward function. In a sense, this simply defers the problem: the algorithm as written provides the framework for incorporating optimism in a way that is necessary to learn from learned rewards, but the solution of the reinforcement learning problem with once-per-episode feedback is left open. The best use-case of this framework is perhaps practical: learn a reward function with uncertainty bonus as defined in the algorithm, then use the learned bonus to solve the problem with episodic algorithms such as reward-weighted regression or ES; however, with the strong assumptions on the structure of the state-action space and the true reward function, it is unclear how much of a boon the derived bonus would give in practice and without specifying the algorithm fully, the given bound cannot actually be obtained in any scenario.

In Section 4, the authors aim to address this problem by providing an approach to iteratively decompose the reward into a per-step reward, to be optimized by an RL algorithm with theoretical guarantees. The issue here is that the additional assumptions made are highly restrictive: Assumption 3.3 says the features decompose additively, similar to the way in which successor states/features operate (a citation would be appropriate here). Assumption 3.4 goes further and defines features to be orthogonal to each other. With this assumption, the features cannot do any summarization of the trajectories anymore and instead fully characterize the state-action-distribution. This allows the authors to maximize the trajectory features in a desired direction and to specify the remainder of the algorithm.

The experimental section of this paper is puzzling, no comparison between algorithms is made and no empirical evaluation of the proposed algorithms seems to be present. It is unclear if the proposed algorithm is even used. Instead, it seems that the experiment serves only as a demonstration that REINFORCE is able to solve some arbitrary MDP with per-episode rewards. Clearly this is known.


**Time Spent Reviewing:**

3

---

> ### Author Response · Authors · 2021-08-10
> **Thank you for your review!**
>
> We want to thank the reviewer for their comments. We would like to address the reviewer’s concerns one by one.
>
> - “the algorithm and its accompanying regret bound…. reinforcement learning problem with once-per-episode feedback is left open.”:
>
> We believe that this is an incorrect assessment of our results. If the environment is not deterministic the optimal policy may induce a distribution over trajectories with support larger than one. We are also not sure what the reviewer meant here by “the solution of the reinforcement learning problem with once-per-episode feedback is left open.” Algorithm 1 and Theorem 3.2 do exactly this.
>
> - “In Section 4, the authors aim to address this problem by providing an approach to iteratively decompose the reward into a per-step reward, to be optimized by an RL algorithm with theoretical guarantees”:
>
> Again we would again like to clarify and correct the reviewer here. We do not decompose the reward into per-step rewards in Section 4. The model we assume for the purpose of computational tractability is that, there is a sum decomposable surrogate reward on top which the logistic map acts to generate binary rewards that depend on the entire trajectory. We would like to reiterate that, under this assumption, the optimal policy and value function are still non-Markovian and outside the scope of past results.
>
> - Regarding the experiment:
>
> The objective of this section is to show that despite the potential intractability of Algorithm 1 in the worst case, in some cases it is possible to use policy gradients to find an optimal policy even if the rewards are non-Markovian. This means that in practice step (2) of Algorithm 1 can be executed via a simple policy gradient algorithm, possibly using a RNN policy network.

---

> > ### Comment · Reviewer_LW3y · 2021-08-11
> > **Clarification**
> >
> > To clarify on the first point, I understand that Algorithm 1 would solve the problem if it were tractable; however, it is not. This issue has been raised by other reviewers as well. Only in Section 3.2 (not 4, apologies) do we see a tractable version of the algorithm. I understand that the algorithm requires us to find a policy, not a trajectory as I wrote.
> >
> > The second point is a follow-up: additional assumptions are necessary to find a per-step reward that allows for tractable computation. I understand that the per-step reward is a surrogate reward. My concerns here were entirely about the additional assumptions needed to arrive at such a per-step reward and especially assumption 3.4.

---

> > > ### Author Response · Authors · 2021-08-12
> > > **Per step rewards**
> > >
> > > "I understand that the per-step reward is a surrogate reward"
> > >
> > > This is wrong. We cannot stress enough that there is never a per step reward in our setting. All dot products between trajectory features and the unknown vector $\mathbf{w}$ are processed through the logistic map $\mu$. In our setting the objective is to find a policy that maximizes the expectation $E_\pi[\mu(\phi(\tau)^\top \mathbf{w})]$. Notice the expectation is outside the $\mu$ and that $\mathbf{w}$ is not known by the learner. This renders the problem highly non-linear even if $\phi(\tau)^\top \mathbf{w}$ can be decomposed as a sum. The reason why Assumption 3.4 allows us to develop a tractable algorithm is because (as we have explained in section 3.2) it leads to sum decomposable optimistic bonuses.
> > >
> > > When the reviewer says "My concerns here were entirely about the additional assumptions needed to arrive at such a per-step reward and especially assumption 3.4." there is mention of a 'pe-step reward'. This is again wrong. There is never a per step reward in our setting. The whole point of our paper is precisely the introduction of a framework that goes beyond assuming per-step rewards as it is done in traditional treatment of the RL problem.

---

> > > > ### Comment · Reviewer_LW3y · 2021-08-12
> > > > **Clarification**
> > > >
> > > > I am referring to r^v_h which is a reward function that is defined for every step. Again, I understand it is a surrogate reward and not a decomposition of the true reward which cannot be decomposed. Could you expand on the practical implications and reasonableness of the additional assumptions?

---

> > > > > ### Author Response · Authors · 2021-08-23
> > > > > **Response**
> > > > >
> > > > > We thank the reviewer for clarifying their question. While we aren’t sure of the direct practical implications of the additional assumptions we note that similar explorability assumptions have been past work studying theoretical reinforcement learning to provide guarantees in the cases where these assumptions might hold. For example see:
> > > > >
> > > > > 1. Definition 2 and Theorem 4.1 in Provably Efficient Reward-Agnostic Navigation with Linear Value Iteration by Zanette et al., 2020;
> > > > > 2. Assumption 3.1 and 3.2 in Provably efficient RL with Rich Observations via Latent State Decoding by Du et al. 2019;
> > > > >
> > > > > among others.

---

### Decision · Program_Chairs · 2021-09-28

**Decision:**

Accept (Poster)

**Comment:**

The reviewer were overall ambivalous about this paper, even after rebuttal and discussion.

The reviewers agreed that the proposed model is new, interesting, and that the authors provide the first regret bound for this formulation.

However, they were also concerned by a number of aspects of this work:
* several assumptions seem unnatural and are hard to verify without extensive discussion as of their meaning (e.g., the kappa parameter or the decomposability assumption)
* some choices of the formulation unmotivated (beyond being required in the analysis) and not properly explained in the rebuttal (eg, the nature of the reward)
* Alg 1 is intractable, the complexity of Alg 2 is still unclear after rebuttal (while the regret dependency to the number of episode is quite bad), and the Alg in the experimental section has little to see to the previous ones.

I think that a revision addressing these points would require another round of review, and therefore recommend rejecting this paper.

**Consistency Experiment:**

NeurIPS has a long history of experimentation. In 2014, NeurIPS ran an experiment in which 10% of submissions were reviewed by two independent committees to quantify the randomness in the review process. This year, we repeated a variant of this experiment to see how the quality of the review process has changed over time.  This paper was part of the experiment and was therefore assigned to two committees (consisting of reviewers, an Area Chair, and a Senior Area Chair) that reached independent decisions.  If both committees made the same recommendation, this recommendation was followed. If a single committee recommended acceptance, the paper was accepted (with the exception of a few cases in which the other committee identified what we considered a fatal flaw, e.g., an error in a key result).

This copy’s committee reached the following decision: **Reject**

The other committee assigned to the paper recommended **Accept (Poster)**.  You can find the other set of reviews, along with any follow up discussion with the authors here:
https://openreview.net/forum?id=c-CD0Yu1ew1